# RIFINs displayed on malaria-infected erythrocytes bind KIR2DL1 and KIR2DS1

Akihito Sakoguchi[1,10], Samuel G. Chamberlain[2,3,10], Alexander M. Mørch[4], Marcus Widdess[4], Thomas E. Harrison[2], Michael L. Dustin[4,5], Hisashi Arase[6,7,8,9 ✉], Matthew K. Higgins[2,3 ✉] & Shiroh Iwanaga[1,8,9 ✉]

Natural killer (NK) cells use inhibitory and activating immune receptors to differentiate between human cells and pathogens. Signalling by these receptors determines whether an NK cell becomes activated and destroys a target cell. In some cases, such as killer immunoglobulin-like receptors, immune receptors are found in pairs, with inhibitory and activating receptors containing nearly identical extracellular ligand-binding domains coupled to different intracellular signalling domains[1]. Previous studies showed that repetitive interspersed family (RIFIN) proteins, displayed on the surfaces of *Plasmodium falciparum*-infected erythrocytes, can bind to inhibitory immune receptors and dampen NK cell activation[2,3], reducing parasite killing. However, no pathogen-derived ligand has been identified for any human activating receptor. Here we identified a clade of RIFINs that bind to inhibitory immune receptor KIR2DL1 more strongly than KIR2DL1 binds to the human ligand (MHC class I). This interaction mediates inhibitory signalling and suppresses the activation of KIR2DL1-expressing NK cells. We show that KIR2DL1-binding RIFINs are abundant in field-isolated strains from both Africa and Asia and reveal how the two RIFINs bind to KIR2DL1. The RIFIN binding surface of KIR2DL1 is conserved in the cognate activating immune receptor KIR2DS1. We find that KIR2DL1-binding RIFINs can also bind to KIR2DS1, resulting in the activation of KIR2DS1-expressing NK cells. This study demonstrates that activating killer immunoglobulin-like receptors can recruit NK cells to target a pathogen and reveals a potential role for activating immune receptors in controlling malaria infection.

The deadliest human-infective malaria parasite, *Plasmodium falciparum*, places protein molecules on the surfaces of infected red blood cells (iRBCs). These are often encoded by multigene families, of which repetitive interspersed family (RIFIN) proteins are the largest. Clades of RIFINs have been shown to bind inhibitory receptors, such as leukocyte immunoglobulin-like receptor B1 (LILRB1) and leukocyte-associated immunoglobulin-like receptor 1 (LAIR1)[4]. LILRB1-binding RIFINs suppress natural killer (NK) cell function and are likely to reduce clearance of the parasite by host immunity[2]. A correlation between high anti-RIFIN antibody titres and less severe disease has been observed in infected children in Gabon[5]. Moreover, people living in malaria-endemic areas across Africa have unique antibodies, containing exons of LILRB1 and LAIR1, which can bind multiple RIFINs[4,6,7]. Therefore, RIFIN-mediated suppression of the host immune response is likely to facilitate parasite survival, and recognition of RIFINs by the host may aid in protection against malaria.

Killer immunoglobulin-like receptors (KIRs) are expressed on NK cells and consist of pairs of inhibitory and activating immune receptors.

Within each pair, these KIRs possess highly similar extracellular domains but contain differences in their intracellular domains, which results in opposite outcomes from signalling. There are seven types of inhibitory KIRs (KIR2DL1–3 and 5 and KIR3DL1–3) and six types of activating KIRs (KIR2DS1–5 and KIR3DS1)[8]. Inhibitory KIRs play a crucial role in mediating the 'missing-self' theory of NK cells[9]. Here reduced concentrations of the inhibitory KIR ligand (human leukocyte antigen (HLA) class I) in virus-infected cells result in reduced KIR-mediated inhibitory signalling and increased NK cell-mediated elimination of infected cells[10]. Activating KIRs have been proposed to have evolved from inhibitory KIRs through truncation of their cytoplasmic tails, addition of a charged residue in their transmembrane regions to facilitate pairing with immunoreceptor tyrosine-based activation motif-containing adaptor proteins (such as DAP12) and mutations in the extracellular domains that attenuate recognition of HLA-encoded ligands[11]. Activating KIRs have also been proposed to recognize surface-displayed molecules on pathogens, thereby activating NK cells to facilitate the clearance of pathogens[12]. However, no pathogen-derived protein ligands for activating KIRs have yet been identified, leaving their biological roles during infection hypothetical.

[1]Department of Protozoology, Research Institute for Microbial Diseases, The University of Osaka, Suita, Japan. [2]Department of Biochemistry, University of Oxford, Oxford, UK. [3]Kavli Institute for Nanoscience Discovery, University of Oxford, Oxford, UK. [4]Kennedy Institute of Rheumatology, University of Oxford, Oxford, UK. [5]Chinese Academy of Medical Sciences Oxford Institute, University of Oxford, Oxford, UK. [6]Department of Immunochemistry, Research Institute for Microbial Diseases, The University of Osaka, Suita, Japan. [7]Laboratory of Immunochemistry, World Premier International Research Center, Immunology Frontier Research Center, The University of Osaka, Suita, Japan. [8]Center for Infectious Disease Education and Research, The University of Osaka, Suita, Japan. [9]Center for Advanced Modalities and DDS, The University of Osaka, Suita, Japan. [10]These authors contributed equally: Akihito Sakoguchi, Samuel G. Chamberlain. ✉e-mail: arase@biken.osaka-u.ac.jp; matthew.higgins@bioch.ox.ac.uk; iwanaga@biken.osaka-u.ac.jp

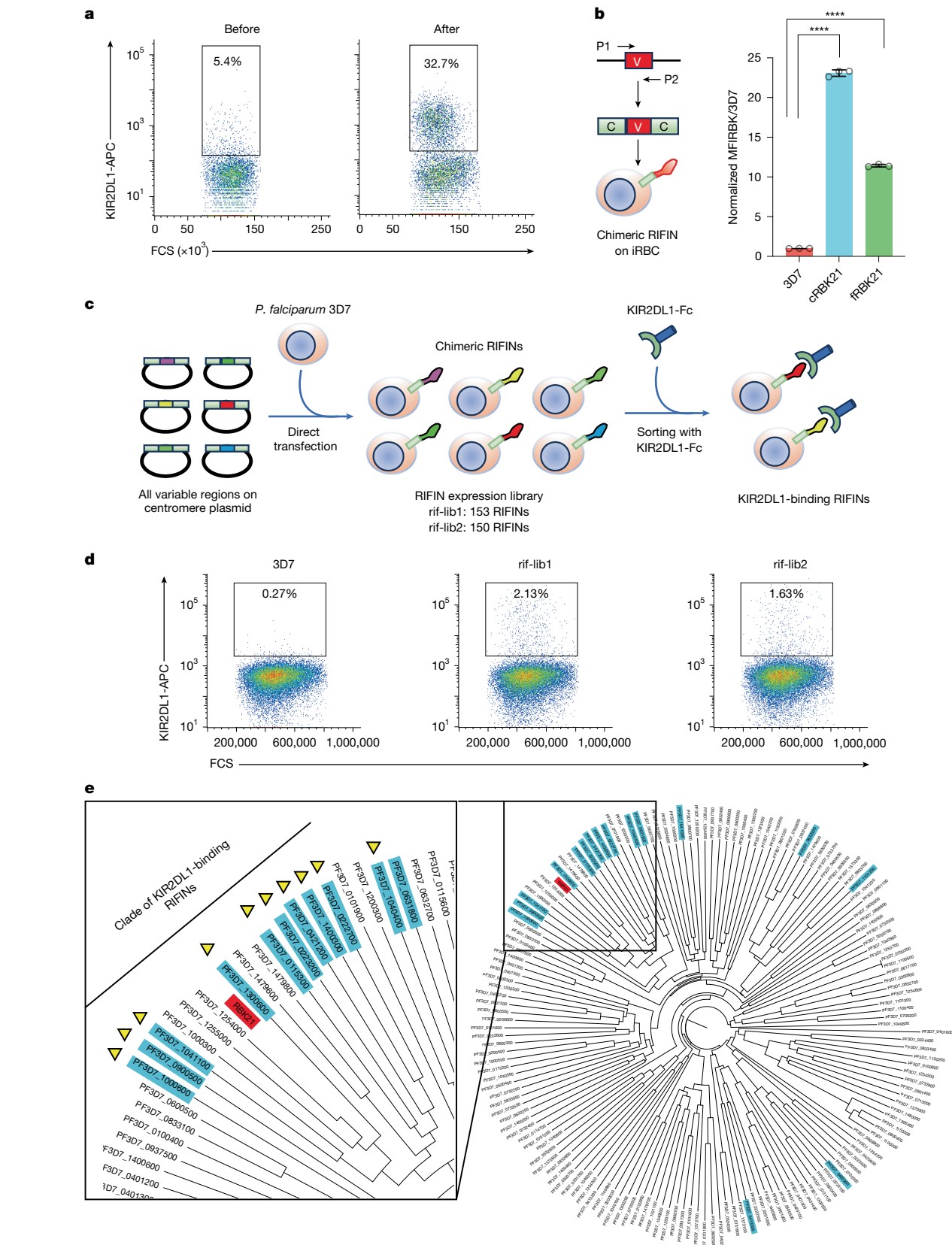

**Fig. 1** | See next page for caption.

## Identification of KIR2DL1-binding RIFINs

The discovery of RIFINs that bind to inhibitory immune receptors, coupled with the structural similarity of KIRs to LILRs[13] and LAIRs[14], led us to investigate whether some RIFINs bind inhibitory KIRs. We first assessed whether four field-isolated strains of *P. falciparum*, collected in Thailand, interacted with seven fluorescently labelled Fc-tagged inhibitory KIRs by flow cytometry (Extended Data Fig. 1). A small fraction (5.4%) of

Fig. 1 | Identification of KIR2DL1-binding RIFINs. a, Flow sorting of Lek174-iRBCs, before (left) and after (right) selection using fluorescent KIR2DL1-Fc. b, A chimaeric KIR2DL1-binding RIFIN containing the variable domain of RBK21 was expressed on the surface of iRBCs. The normalized mean fluorescence intensity (MFI) of binding of iRBCs expressing chimaeric RBK21 (cRBK21) and full-length RBK21 (fRBK21) to KIR2DL1-Fc was calculated by dividing with the MFI for parental 3D7 iRBCs. Data represent the mean ± s.d.; $n = 3$ biologically independent samples, with ****$P < 0.0001$ (two-sided Student's $t$-test; $P$ values in source data). c, Schematic overview of the generation of the RIFIN expression library and its screening for binding to KIR2DL1-Fc. d, Flow sorting of iRBC expressing KIR2DL1-binding RIFINs from the 3D7 strain, rif-lib1 or rif-lib2 using fluorescent KIR2DL1-Fc. These iRBCs were further sorted and used to identify KIR2DL1-binding RIFINs with next-generation sequencing. e, Phylogenetic tree of the variable regions of RIFIN from the *P. falciparum* 3D7 strain. The KIR2DL1-binding RIFINs commonly identified from rif-lib1 and rif-lib2 are highlighted in blue, whereas those classified into the same clade are marked with triangles. RBK21 is indicated in pink.

iRBCs that had been infected with strain Lek174 bound to allophycocyanin (APC)-labelled KIR2DL1-Fc (Fig. 1a). By contrast, we did not detect significant binding between other inhibitory KIRs and iRBC (Lek174) or binding of any inhibitory KIR-Fc to the remaining three parasite strains tested, despite these KIR-Fc proteins all binding to antibodies with conformational epitopes by flow sorting (Extended Data Fig. 1a). As not all RIFINs are expressed at any time in a population of iRBCs, this does not exclude that these lines also contain KIR2DL1-binding RIFINs.

Next, we assessed whether a RIFIN mediates the binding of Lek174-infected iRBCs to KIR2DL1. Because other RIFINs use their variable domains for ligand binding, we enriched KIR2DL1-binding cells from iRBCs infected with parental Lek174 by cell sorting (Fig. 1a). We then produced complementary DNA (cDNA) for the RIFIN variable domains. This amplified a single cDNA, which we named 'RIFIN that binds to KIR2DL1' (RBK21). We generated a transgenic parasite line that expressed a chimaeric RIFIN consisting of the variable domain of RBK21 fused to the conserved region of a LILRB1-binding RIFIN (encoded by PF3D7_1254800) (Supplementary Fig. 1) and showed that this line bound KIR2DL1-Fc (Fig. 1b). We cloned the full-length cDNA of this RIFIN, and iRBCs infected with a parasite line expressing this RIFIN also bound to KIR2DL1-Fc (Fig. 1b).

The *P. falciparum* genome contains around 150 RIFINs, with transcription of most of these suppressed by heterochromatin[15]. Therefore, we tested whether this includes multiple KIR2DL1-binding RIFINs by producing and screening a comprehensive RIFIN expression library that includes iRBCs expressing each of the RIFINs from the laboratory-adapted 3D7 strain. The RIFIN variable regions were amplified from genomic DNA using degenerate primers, which amplified all RIFINs and were ligated into the centromere plasmid, pFCEN_rif, between two conserved regions from PF3D7_1254800 (Fig. 1c). These chimaeric RIFINs were electroporated into the recipient 3D7 strain. Library construction was performed in duplicate, generating two biologically independent RIFIN expression libraries (rif-lib1 and rif-lib2). To determine the RIFIN coverage, we recovered variable regions and sequenced them using amplicon-seq. This showed high coverage, identifying 150 (ref-lib1) and 153 (rif-lib2) of the 157 possible variable regions from 3D7 (Supplementary Table 1).

We then assessed the binding of iRBCs expressing these libraries to KIR2DL1-Fc by flow sorting and identified 2.13% of positive iRBCs in rif-lib1 and 1.63% in rif-lib2 (Fig. 1d and Supplementary Table 2). Enriched iRBC libraries were cultured, and amplicon-seq of the recovered pFCEN_rif plasmids identified KIR2DL1-binding RIFIN candidates. Notably, 19 and 24 RIFINs were identified as candidates from rif-lib1 and rif-lib2, respectively, with 16 of these identified in both biological replicates (Extended Data Table 1). To investigate the relationship between these KIR2DL1-binding RIFINs, we conducted a phylogenetic analysis on the basis of amino acid sequences of variable regions. Ten of the 16 KIR2DL1-binding candidate RIFINs were classified into the same clade of this tree, indicating that they probably evolved from a common ancestral RIFIN (Fig. 1e and Extended Data Fig. 2a). RBK21 from the Lek174 strain fell within the same clade, indicating that this evolutionary relationship is probably conserved among *P. falciparum* strains (Fig. 1e). This clade consisted of 17 RIFINs, of which ten were identified by library screening. To determine whether the entire clade binds to KIR2DL1, we generated transgenic parasites expressing each

of the 17 RIFINs and discovered that 14 of the 17 RIFINs could bind to KIR2DL1-Fc (Extended Data Fig. 2b). The two RIFINs that did not bind to KIR2DL1 were not identified in the library screen and may have lost KIR2DL1 binding during evolution.

## KIR2DL1-binding RIFINs are common

We investigated whether KIR2DL1-binding RIFINs are widely conserved in *P. falciparum* strains from different geographical locations. Two biologically independent RIFIN libraries were generated from each of the two Southeast Asian strains (Lek174 and Lek79) and were screened for KIR2DL1-Fc binding. From both strains, we detected iRBCs that bind KIR2DL1-Fc (Extended Data Fig. 2c). Our previous observation that Lek79-iRBCs did not bind KIR2DL1-Fc (Extended Data Fig. 1b) was attributable to the transcriptional silencing of KIR2DL1-binding RIFINs[15] rather than a lack of such RIFINs in the genome. KIR2DL1-binding iRBCs were enriched by two rounds of cell sorting, followed by next-generation sequencing analysis using the assembled contigs of Lek174 and Lek79 as reference genomic sequences. We selected the top 20 candidates from each screen on the basis of mapped read counts (Extended Data Table 2). The number of RIFINs found in the top 20 in both replicates was 16 for Lek174 and 17 for Lek79. We extracted variable region sequences of the identified candidates and conducted phylogenetic analysis (Fig. 2a and Extended Data Fig. 2d), obtaining 12 variable regions from Lek174 and 16 from Lek79. Among these, 6 and 12 RIFINs were classified into the same clade as the KIR2DL1-binding RIFINs from the 3D7 strain. Therefore, these three strains all contain equivalent clades of RIFINs that bind to KIR2DL1.

We investigated whether we could identify KIR2DL1-binding RIFINs in the genomes of two African strains: pfKE01 from Kenya and pfSN01 from Senegal[16]. Phylogenetic analysis was conducted for a panel of all RIFINs from each of these strains, together with the KIR2DL1-binding RIFINs from 3D7. In both cases, the KIR2DL1-binding RIFINs from 3D7 clustered into one major clade of RIFINs from the field-isolated strain, which contained 14 RIFINs from pfKE01 and eight from pfSN01 (Fig. 2b and Extended Data Fig. 2e,f). To determine whether these clades contained KIR2DL1 binders, six representative RIFINs from each field isolate, selected to represent sequence diversity across the clade, were produced to test for KIR2DL1 binding by surface plasmon resonance (SPR). Of the six Kenyan RIFINs, five were expressed in sufficient quantities for SPR analysis, four bound to KIR2DL1 and one unexpectedly bound to LILRB1 (Fig. 2c). Of the six Senegalese RIFINs, four were expressed, three bound to KIR2DL1 and one bound to both KIR2DL1 and LILRB1. Therefore, we can predict, although imperfectly, which RIFINs bind to KIR2DL1. Given that each *P. falciparum* strain contains approximately 150 RIFINs[16], KIR2DL1-binding RIFINs make up approximately 10% of the RIFIN repertoire in all four field-isolated strains tested from geographically distinct regions of both Southeast Asia and Africa. Therefore, KIR2DL1-binding RIFINs are a conserved and evolutionarily retained feature of *P. falciparum*.

## Structures of RIFIN–KIR2DL1 complexes

Previous structural studies have shown how RIFINs bind to the inhibitory immune receptors LAIR1 and LILRB1 (refs. 3,17,18). RIFINs can

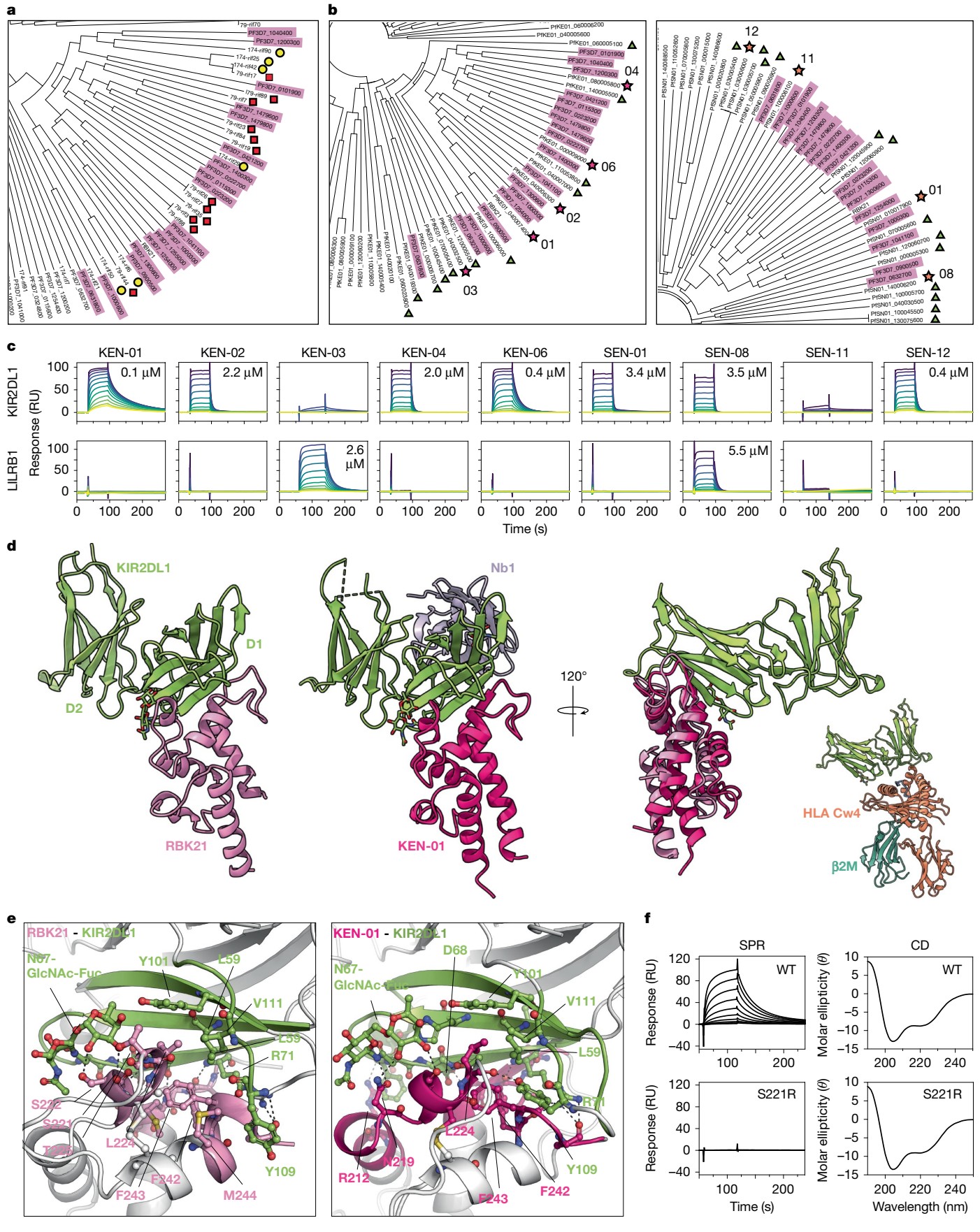

**Fig. 2** | See next page for caption.

**Fig. 2 | Structural basis of RIFIN binding to KIR2DL1. a**, Phylogenetic analysis of KIR2DL1-binding RIFINs from two Thai field-isolated strains (red squares, Lek79; yellow circles, Lek174) and the 3D7 KIR2DL1 binders (pink highlight). **b**, Phylogenetic analyses of RIFINs from pfKE01 (left) or pfSE01 (right) and the KIR2DL1-binding RIFINs from 3D7 (pink highlight). Triangles indicate RIFINs predicted to bind to KIR2DL1, and stars indicate the candidates tested by SPR analysis in **c. c**, SPR analysis of the binding of KIR2DL1-binding RIFIN from pfKE01 and pfSN01 to KIR2DL1 and LILRB1. Affinities calculated by equilibrium fitting from at least three independent experiments (Prism 10) are denoted within each box, if applicable. **d**, Crystal structures of complexes of KIR2DL1 (green) with RBK21 (light pink) or KEN-01 (dark pink). The right-hand panel shows the overlay of these two complexes aligned on KIR2DL1. The lower right inset shows KIR2DL1 (green) bound to MHC class I, HLA-Cw4 (orange and teal) (Protein Data Bank (PDB) 1IM9). **e**, Interfaces between RBK21 (left) or KEN-01 (right) and KIR2DL1, showing interfacial residues. **f**, Experiments comparing binding (SPR, left) and folding (circular dichroism (CD), right) of wild-type (WT) RBK21 and its S221R mutant. For SPR, twofold dilutions from 20 μM to 39 nM of either WT or S221R RBK21 were flowed over the immobilized KIR2DL1. The same RBK21 variants were tested for correct folding by comparison by means of CD spectroscopy.

mimic human HLA class I, thereby signalling through LILRB1 to reduce NK cell activation[2,3]. HLA class I molecules are also ligands of KIR2DL1. To show how RIFINs bind to KIR2DL1, we determined the structures of RBK21 and KEN-01, each bound to KIR2DL1. For RBK21, the variable region (residues 148–299) was mixed with the KIR2DL1 ecto-domain and crystallized (Extended Data Fig. 3a). Data were collected at 2.89 Å, and molecular replacement using KIR2DL1 (ref. 19) as a search model allowed for structure determination, revealing eight copies of the KIR2DL1:RBK21 complex in the asymmetric unit (Extended Data Table 3). For KEN-01, we expressed the variable domain (159–288) and crystallized it in complex with KIR2DL1 and a KIR2DL1-binding nanobody. The nanobody was included as a crystallization chaperone (Extended Data Fig. 3b). Data were collected at 2.17 Å, and molecular replacement allowed structure determination, this time with a single copy of the complex in the asymmetric unit (Extended Data Table 3).

The variable regions of both RBK21 and KEN-01 consisted of a helical bundle that presented extended loops and short helices at the KIR-proximal end (Fig. 2d and Extended Data Figs. 3 and 4). These RIFINs share 87% sequence identity, and structural alignment showed a root-mean-square deviation of 2.0 Å. In both cases, the KIR2DL1 binding site was formed from two helices and two loops, which contacted the D1 domain of KIR2DL1, with a conserved binding pose despite only 53% of the KIR2DL1-contacting residues being shared (Fig. 2d,e and Extended Data Table 4). The binding site was centred around a conserved hydrophobic core, with L224 and F242 from each RIFIN occupying a shallow hydrophobic pocket containing residues L59, Y101 and V111 of KIR2DL1. This was stabilized by an intermolecular β-sheet, which contained loop 4 of RBK21 or KEN-01 and ended with a hydrogen bond linking S258 of RBK21 or KEN-01 and D68 of KIR2DL1. Both RIFINs bound the *N*-acetylglucosamine–fucose core of an N-linked glycan on N67 of KIR2DL1 but mediated this interaction using different residues (Fig. 2e and Extended Data Fig. 3c). Other differences included a variable network of hydrogen bonds and interactions involving R212 of KEN-01, which formed a salt bridge with D93, and a cation–π interaction with F66 of KIR2DL1, both of which were absent in RBK21. In RBK21, residue S221 was at the centre of the interface. To develop a non-binding mutant of RBK21 for future studies, we produced S221R. This abolished binding to KIR2DL1, as shown by SPR analysis, despite circular dichroism indicating that it adopted the correct fold (Fig. 2f). Notably, RIFIN and MHC class I bound to non-overlapping sites on KIR2DL1, although they approached from the same direction (Fig. 2d).

We next investigated conservation in both KIR2DL1 and KIR2DL1-binding RIFINs because KIR2DL1 has several allotypes and KIR2DL1-binding RIFINs can share sequence identity as low as 35% while still retaining binding. We first selected the validated 3D7 KIR2DL1-binding RIFINs, calculated the sequence entropy for each amino acid and plotted this onto the RBK21 structure (Extended Data Fig. 4c). Conserved residues were predominantly found within the domain core, indicating the maintenance of the overall architecture (Extended Data Fig. 4d). By contrast, the surface, except for the two exposed serine residues, was highly variable. This pattern of conservation of critical hydrophobic residues that maintain a core fold, coupled with high surface sequence

diversity, is also observed in other iRBC surface protein families, such as the DBL and CIDRα domains of PfEMP1 (refs. 20,21), CIR proteins[22] and LILRB1-binding RIFINs[3]. With low sequence identity between KIR2DL1-binding RIFINs, we investigated whether they compete for KIR2DL1 binding. Indeed, LILRB1-binding RIFINs can bind to two different non-competing sites on LILRB1 (refs. 3,4). Therefore, we coupled KIR2DL1 to an SPR chip and flowed over KEN-01, mixed with each of the Kenyan and Senegalese KIR2DL1-binding RIFINs. In each case, we observed RIFIN binding at levels similar to that observed for KEN-01 alone, indicating that they cannot bind simultaneously (Extended Data Fig. 4e). Therefore, although there may be subtle differences in binding site, KIR2DL1-binding RIFINs share a similar binding site on KIR2DL1. Finally, 177 allotypes of KIR2DL1 have been observed in different global populations, as recorded in the IPD-KIR database[23]. We plotted all polymorphisms of KIR2DL1 found in this database onto the RBK21:KIR2DL1 structure, and only four of 38 were in contact with RIFIN (Extended Data Fig. 4f). Of these, only V111, which is found within the RIFIN-binding hydrophobic pocket of KIR2DL1, is reported to be common in malaria-endemic regions, with V111F found in Baka populations in northern Gabon/southwest Cameroon[24]. However, the V111F mutation in KIR2DL1 did not alter RBK21 binding affinity (Extended Data Fig. 4g). Therefore, despite various KIR2DL1 allotypes and natural sequence variation in RIFINs, the KIR2DL1-binding phenotype is conserved in this group of RIFINs.

## RIFINs can signal through KIR2DL1

The different binding sites for RBK21 and MHC class I on KIR2DL1 raised the question of whether RIFINs can transduce inhibitory signals, analogous to HLA class I. To test this, we used an assay in which iRBCs infected with RBK21-expressing transgenic parasites were incubated with an activated T cell reporter line (NFAT–GFP). In this system, the extracellular domain of KIR2DL1 was fused to the transmembrane and intracellular domains of paired immunoglobulin-like receptor β (PILRβ). Upon KIR2DL1 activation, this interacted with a signalling adaptor (DAP12), resulting in GFP expression. Co-culture of this line with iRBCs infected with the parental 3D7 strain showed GFP expression in less than 0.41% of the KIR2DL1-reporter cells (Fig. 3a). By contrast, iRBCs infected with RBK21-expressing parasites induced strong GFP expression of 41.5% in the reporter line (Fig. 3a).

Next, we investigated whether RBK21-mediated signalling suppressed NK cell activity. We examined the effect of RBK21 on the activation of NK cells by measuring the expression of CD107a, a marker of degranulation, and the production of the cytokines interferon-γ (IFNγ) and tumour necrosis factor (TNF). Primary KIR2DL1-positive NK cells were purified from peripheral blood mononuclear cells (PBMCs) obtained from donors (Fig. 3b), and the effect of co-culturing these KIR2DL1-positive NK cells with K562 cells, which lack HLA class I expression, was monitored. When purified NK cells were co-cultured with normal K562, CD107a expression and production of IFNγ and TNF were detected in 72.6 ± 0.3%, 40.1 ± 0.4% and 24.2 ± 0.4% of the KIR2DL1-positive NK subset, respectively, suggesting a substantial fraction of activated cells (Fig. 3c–e and Extended Data Fig. 5a). Similar CD107a

expression and production of IFNγ and TNF (69.6 ± 0.2%, 29.8 ± 0.4% and 17.6 ± 0.2% of the KIR2DL1-positive NK subset) were detected when co-cultured with K562 expressing a RIFIN that does not bind to KIR2DL1 (PF3D7_1254200). By contrast, when K562 expressing RBK21 was co-cultured with NK cells, lower CD107a expression (30.5 ± 1.0%) and lower production of IFNγ (5.4 ± 0.2%) and TNF (3.8 ± 0.3%) were detected in the KIR2DL1-positive subset. Similar suppression of CD107a expression and production of IFNγ and TNF was reproducibly detected in the KIR2DL1-positive subset obtained from another donor (Extended Data Figs. 5b and 6a). By contrast, suppression by RBK21 was not detected in the KIR2DL1-negative subset, and its effect on these NK cells was comparable to that of the negative control, RIFIN (PF3D7_1254200), which did not bind to KIR2DL1 (Extended Data Figs. 5c and 6b). Furthermore, RBK21-expressing K562 cells significantly suppressed the cytotoxic activities of KIR2DL1-positive NKL cells (Fig. 3f). In contrast, K562 and K562 expressing PF3D7_1254200 were killed by activated KIR2DL1-positive NKL cells (Fig. 3f), and K562 expressing RBK21 was also killed by NKL and KIR2DL3-positive NKL (Extended Data Fig. 6d). Therefore, RBK21 specifically inhibited the activation of NK cells by means of inhibitory signalling through KIR2DL1.

Next, we used a supported lipid bilayer (SLB) assay[3] to image the effects of RBK21 and KEN-01 on NK cell activation (Fig. 3g–i). Primary NK cells were purified from three independent donors and incubated with SLBs, which display ICAM-1 to mediate adhesion, and PfRH5, together with a human PfRH5-binding antibody, to trigger antibody-dependent cellular cytotoxicity (ADCC). Some bilayers were also coated with RBK21, the non-binding RBK21[S221R] mutant, KEN-01 or the LILRB1-binding RIFIN (Pf3D7_1254800) as a positive control. Immune synapses were imaged using total internal reflection fluorescence microscopy, allowing visualization of RIFIN accumulation and perforin deposition (Fig. 3g–i)[25]. Synapses on bilayers that lacked RIFINs showed perforin deposition, whereas bilayers decorated with Pf3D7_1254800 showed RIFIN accumulation and reduced perforin deposition, as previously observed[3]. When RBK21 and KEN-01 were present in the bilayer, they were recruited to the immune synapse (Fig. 3i) and caused a significant reduction in perforin deposition (Fig. 3h), indicating KIR2DL1-mediated signalling. By contrast, the non-binding mutant RBK21[S221R] was not significantly recruited to the synapse and did not inhibit perforin deposition. These data confirmed that RIFIN-mediated engagement of KIR2DL1 results in inhibitory signalling that suppresses NK cell activation.

## RIFINs bind activating receptor KIR2DS1

The KIR family contains both inhibitory and activating receptors, which have largely conserved extracellular domains. Inspection of the RBK21–KIR2DL1 structure showed that the KIR2DL1 residues, which interact with RBK21, were almost completely conserved in KIR2DS1 (Fig. 4a,b), except for the conservative substitution of V111L. Therefore, we used SPR analysis to quantify the binding of KIR2DL1 and KIR2DS1 to RBK21 and the MHC class I molecule (HLA-Cw4). KIR2DS1 bound with a 1.7 μM affinity to RBK21, close to the 0.5 μM affinity of the RBK21–KIR2DL1 complex, whereas its affinity for HLA-Cw4 is more than 10-fold weaker (Fig. 4c). We also measured the binding of the Kenyan and Senegalese field isolates to KIR2DS1 and found that all KIR2DL1-binding RIFINs also bound to KIR2DS1 with low micromolar affinities (Extended Data Fig. 7a). Flow cytometry analysis demonstrated that transgenic parasites expressing RBK21 also bound to KIR2DS1 (Fig. 4d). This indicates that KIR2DL1-binding RIFINs are new pathogen-displayed ligands for activating KIR receptors, supporting a model in which RIFIN recognition might activate KIR2DS1-expressing NK cells to destroy infected erythrocytes.

Using the NFAT–GFP reporter system modified to express KIR2DS1, we next examined whether binding of RBK21 to KIR2DS1 resulted in signal transduction (Fig. 4e). Increased expression of GFP was found in KIR2DS1-expressing reporter cells co-cultured with RBK21-expressing parasites (31.9%) compared to the control (0.4%). We also co-cultured

RBCs infected with RBK21-expressing transgenic parasites with primary NK cells containing the KIR2DS1-positive NK subset, which was obtained from a donor by flow sorting (Fig. 4f), and monitored the markers CD107a, TNF and IFNγ. The presence of RBK21 resulted in increased CD107a expression (82.1 ± 1.1%) and in the production of IFNγ (29.4 ± 1.5%) and TNF (27.8 ± 0.8%) in the KIR2DS1-positive NK cell subset (Fig. 4f–i and Extended Data Fig. 8a). By contrast, RBCs infected with wild-type parasite (CD107a, 11.2 ± 0.6%; IFNγ, 1.7 ± 0.2%; and TNF, 1.5 ± 0.4%) or with a transgenic parasite expressing PF3D7_1254200 (CD107a, 11.2 ± 0.7%; IFNγ, 0.9 ± 0.1%; and TNF, 0.9 ± 0.2%) did not activate the KIR2DS1-expressing NK subset. Similar RBK21-induced increases in CD107a expression (90.1 ± 0.3%) and in production of IFNγ (61.1 ± 1.4%) and TNF (48.9 ± 1.1%) were also observed in the KIR2DS1-positive NK subset obtained from a second donor (Extended Data Figs. 7b and 8b). Although activation of KIR2DS1-positive NK subset by RBCs infected with wild-type parasite (CD107a, 68.6 ± 0.8%; IFNγ, 30.9 ± 0.8%; and TNF, 17.4 ± 0.4%) or with a transgenic parasite expressing PF3D7_1254200 (CD107a, 71.2 ± 0.3%; IFNγ, 29.1 ± 0.4%; and TNF, 15.4 ± 0.4%) was detected, the activation by iRBC with RBK21 was significantly greater than either (Extended Data Fig. 7b). Therefore, KIR2DL1-binding RIFIN can engage both the inhibitory receptor KIR2DL1 and activating receptor KIR2DS1. Although the engagement of KIR2DL1 suppresses KIR2DL1-expressing NK cells, KIR2DS1 binding has the opposite effect, activating KIR2DS1-expressing NK cells by triggering both cytotoxic and cytokine responses. Because different NK cells present within different humans express different KIRs, this will equip the subset of NK cells expressing KIR2DS1 with the ability to clear iRBCs expressing these RIFINs.

## Discussion

Paired immune receptors consist of an inhibitory receptor and an activating receptor with similar extracellular domains but opposite signalling outcomes on ligand binding. KIRs are a family of paired receptors found in humans. Although both human and pathogen-derived ligands have been identified for inhibitory KIRs, no ligand has been found for any activating KIR. An analogous mouse system involves inhibitory and activating C-type lectin Ly49 receptors[26]. Inhibitory Ly49 receptors, such as Ly49I, recognize MHC class I molecules as endogenous ligands and regulate NK cell function, similar to inhibitory KIRs[27]. In addition, an activating Ly49 receptor (Ly49H) binds to the murine cytomegalovirus-derived molecule (m157), which is expressed on infected cell surfaces[28], triggering cytotoxic activity and viral clearance. We now provide experimental evidence that activating KIRs have a similar role, recognizing pathogen-derived molecules in the form of RIFINs, allowing NK cells to contribute to parasite killing.

Recent in vitro studies have shown that NK cells can destroy malaria parasite iRBCs[29]. This operates through the antibody-dependent mechanism (ADCC), in which the CD16 receptor (FCγRIII) on an NK cell recognizes the Fc portion of a pathogen-bound antibody[30]. Indeed, epidemiological studies conducted in Ugandan children have shown that repeated infection with *P. falciparum* alters the composition of NK cell populations, resulting in an increase in CD56neg NK cells, which exhibit stronger ADCC activity than CD56dim NK cells, perhaps increasing NK cell-mediated iRBC clearance[31]. However, RBCs infected with *P. falciparum* express RIFINs that bind to inhibitory immune receptors. In this study, we have shown that in addition to RIFINs that bind to LAIR1 and LILRB1, KIR2DL1-binding RIFINs are commonly found in *P. falciparum* genomes and signalling through KIR2DL1 to suppress NK cell activity. Therefore, the parasite genome contains different families of RIFINs that can suppress ADCC by signalling through different inhibitory receptors, most probably protecting the parasite from NK cell-mediated killing.

Activating immune receptors evolved under selection pressure that resulted in the retention of extracellular homology between the

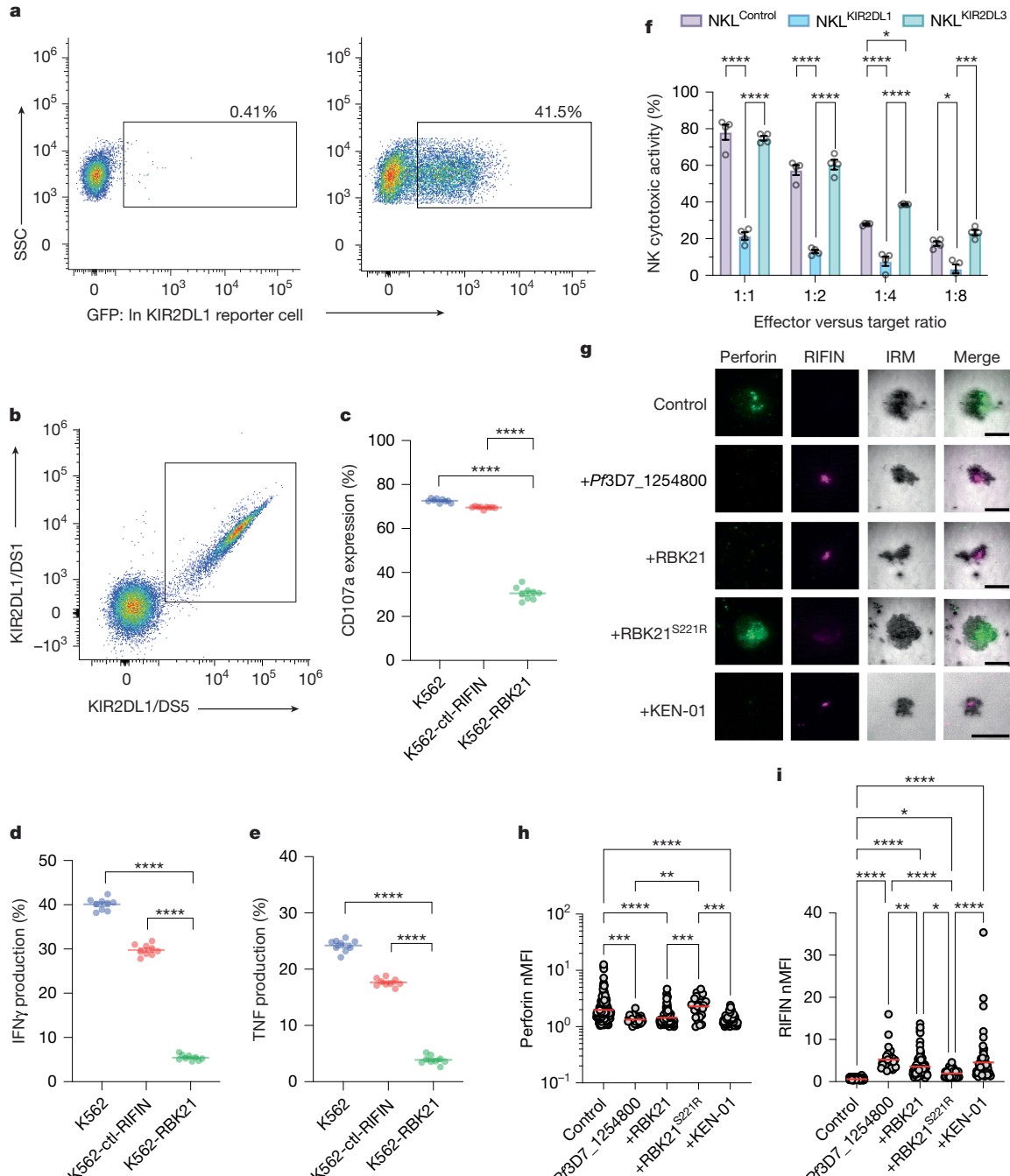

**Fig. 3 | Effect of RBK21 on NK cell function. a**, GFP expression in KIR2DL1-reporter cells upon stimulation with iRBCs infected with parental 3D7 cells (left) or the RBK21-expressing parasite line (right). The percentage of GFP-positive cells is shown. **b**, KIR2DL1-positive NK cells gated using anti-KIR2DL1/DS1 and anti-KIR2DL1/DS5 antibodies for **c**–**e**. **c**–**e**, Suppression of CD107a expression (**c**), IFNγ production (**d**) and TNF production in KIR2DL1+ NK cells by K562 cells, which express RBK21, with K562 parental cells or K562 expressing a non-KIR2DL1-binding RIFIN (ctl-RIFIN; PF3D7_1254200) as negative controls (**e**). Lines show the mean (*n* = 9 independent measurements from one donor) with ****P < 0.0001 (two-sided Student's *t*-test). Equivalent analysis of cells from a different donor is shown in Extended Data Fig. 6a. **f**, Effect of K562 cells expressing RBK21 on the cytotoxic activity of NKL cells (purple) or NKL cells expressing either KIR2DL1 (blue) or KIR2DL3 (green), assessed at four ratios

of target to effector cells. Data represent the mean ± s.d. (*n* = 3 biologically independent samples) with ****P < 0.0001, ***P < 0.001 and **P < 0.01 (two-sided Student's *t*-test). **g**, Analysis of localization of RIFINs (pink) or perforin (green) in contact areas for NK cells on SLBs coated with RIFIN Pf3D7_1254800, RBK21 or RBK21^S221R, showing representative images. **h,i**, Quantification of perforin (**h**) and RIFIN (**i**) in contact areas from **g**. Measurements from three independent donors with control (*n* = 43 cells), Pf3D7_1254800 (*n* = 28 cells), RBK21 (*n* = 31 cells), RBK21^S221R (*n* = 29 cells) and KEN-01 (*n* = 63 cells). Each data point represents a measurement from one cell. Dunn's multiple comparison test (**h**) and Tukey's multiple comparison test (**i**) were performed. *P < 0.05, **P < 0.01, ***P < 0.001 and ****P < 0.0001. Exact *P* values in source data. Scale bar, 10 µm.

members of the activating/inhibitory pair[32]. This is true for KIR2DL1/S1, with the RIFIN-binding interface remaining almost perfectly conserved between the pair. This led us to investigate whether KIR2DS1 can

also bind RIFINs, and we found that all characterized KIR2DL1-binding RIFINs also bind KIR2DS1. By contrast, KIR2DS1 contains a threonine-to-lysine mutation that reduces binding to the MHC class I molecule

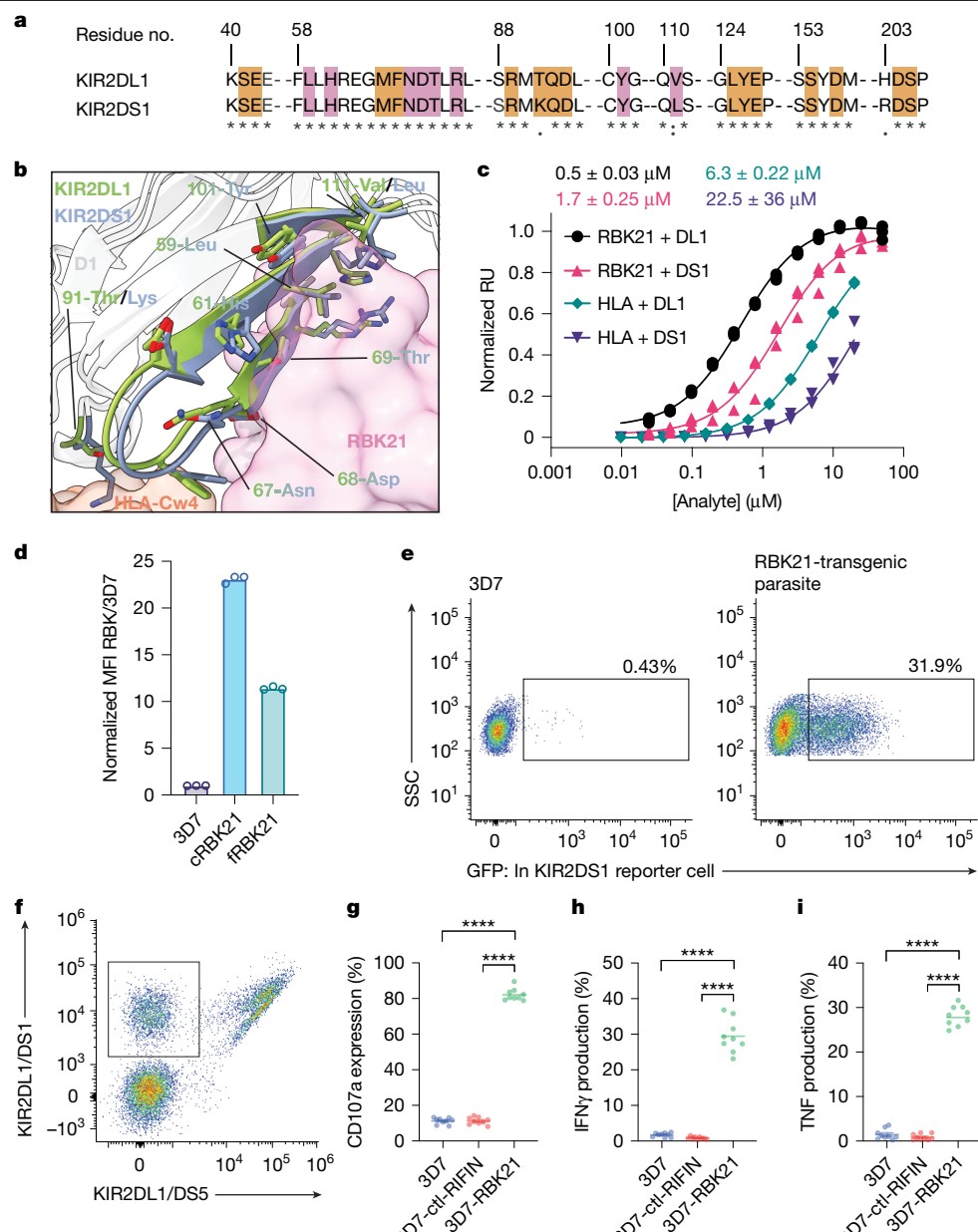

**Fig. 4 | KIR2DL1-binding RIFINs also bind and signal through activating immune receptor KIR2DS1. a**, Sequence alignment of the regions of KIR2DL1 and KIR2DS1, which bind RBK21 and HLA-Cw4. Residues interacting with RBK21 (pink) and HLA-Cw4 (orange) are highlighted. *Indicates identical residues, : indicates similar residues and . indicates residues of weakly similar properties. **b**, The structure of KIR2DL1 (green) overlaid with a homology model of KIR2DS1 (blue). Residues that contact RBK21 (pink surface) or HLA-Cw4 (orange surface) are shown as sticks. Polymorphisms at positions 91 and 111 are highlighted. **c**, SPR analysis of the binding of KIR2DL1 and KIR2DS1 to RBK21 and HLA-Cw4. Each point represents the mean of three independent measurements, and the error bars represent the s.d. Dissociation constants, shown above the graph, were calculated using 'one-site total' fitting model in Prism 10, including all three measurements in fitting. **d**, Normalized MFI of binding of iRBC,

expressing chimaeric RBK21 (cRBK21) and full-length RBK21 (fRBK21), to KIR2DS1-Fc was calculated by dividing by the MFI of 3D7. Data represent the mean ± s.d. of three biologically independent samples, with ****$P < 0.0001$ (two-sided Student's $t$-test). **e**, GFP expression in KIR2DS1-reporter cells upon stimulation with *P. falciparum* 3D7 (left) or with a transgenic parasite expressing RBK21 (right). The percentage of GFP-positive cells is shown. **f**, KIR2DS1-positive NK cells were gated, as indicated in the box. **g–i**, Effect of RBK21 on CD107a expression (**g**) and IFNγ (**h**) and TNF (**i**) production in the gated KIR2DS1+ subset from **f**, assessed as shown in Fig. 3c–e. The 3D7 strain and iRBC expressing PF3D7_1254200 (ctl-RIFIN) were used as negative controls. Data represent the mean ($n = 9$ independent measurements from one donor) with ****$P < 0.0001$ (two-sided Student's $t$-test). A similar assay using a different donor is shown in Extended Data Fig. 7c. Exact $P$ values in source data.

(HLA-Cw4), preventing the activation of KIR2DS1-expressing NK cells when co-cultured with host cells expressing HLA class I[33] (Extended Data Fig. 7c). The presence of KIR2DS1-binding RIFINs in parasites from diverse geographic locations and the conservation of the RIFIN binding surface of KIR2DS1 support a model in which activating KIRs retain the ability to recognize pathogens, such as *P. falciparum*, while reducing binding to their canonical ligands (MHC class I).

Although the host has evolved paired receptors, the parasite has been shaped by evolutionary pressure to avoid immune detection, with antibodies to RIFINs commonly found in individuals who have experienced natural malaria infections, including atypical antibodies that contain the ectodomains of LAIR1 and LILRB1 (refs. 4,5,34). The result of this pressure to diversify is evident in KIR2DL1-binding RIFINs, which conserve their core fold but diversify their surfaces, most probably

generating antigenically distinct molecules. Despite this diversity, phylogenetic analysis allowed us to correctly identify KIR2DL1-binding RIFINs in the African field isolates. These RIFINs were found in isolates from both Southeast Asia and Africa and clustered into the same clade. This indicates that KIR2DL1 binding is likely to be a very common property of RIFINs, which have evolved through a common ancestor and are now found in isolates from across the globe. It also indicates that as we identify the binding partners of other RIFINs, we are likely to be able to predict RIFIN function solely from sequence information, allowing future studies that link the binding phenotype of RIFINs expressed in an individual with the disease outcomes that they experience. KIRs are also members of a complex multigene family, consisting of 14 types of activating, inhibitory and unknown receptors, with the frequency of each activating and inhibitory receptor varying across different ethnicities and in different individuals. Coupled with the complexity of RIFINs and our finding that activating receptors can recognize RIFINs and suppress NK activation, it is clear that continued evolution of both RIFINs and immune receptors plays a fascinating part in the battle between the host and parasite.

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

## Methods

### Ethical statement

Erythrocytes and human serum were obtained from the Japanese Red Cross (research ID 25J0143) with written informed consent. Parasites were collected from Thai patients according to the ethical approval of Chiang Mai University (permission number: 187/2554) and Mie University (permission number: 1312), and their use was approved by the University of Osaka (permission number: 149-003).

### Cells and antibodies

The NKL cells were generously provided by L. L. Lanier at the University of California, San Francisco. The human erythroleukemia cell line (K562) was sourced from the Cell Resource Center for Biomedical Research, Institute of Development, Aging and Cancer, Tohoku University. Anti-KIR2DL1/DS1 (Miltenyi Biotec; 130-118-973) and anti-KIR2DL1/DS5 (R&D Systems; MAB1844-SP) antibodies were purchased and used to detect KIR2DL1$^+$ NK and KIR2DS1$^+$ NK cells in PBMCs. The PBMCs were obtained from healthy donors. Anti-FLAG antibody (Sigma-Aldrich; F1804) and APC-conjugated anti-human IgG Fc antibody (Jackson ImmunoResearch; 109-136-098) were used to detect transgenic K562 and for binding assay involving Fc fusion proteins, respectively. Pacific Blue-conjugated anti-human CD107a (BioLegend; 328623), fluorescein isothiocyanate (FITC)-conjugated CD56 (BioLegend; 318303), PerCP/Cy5.5-conjugated anti-human IFNγ (BioLegend; 506527), APC/Cy7-conjugated anti-human TNF (BioLegend; 502943) and APC-conjugated anti-mouse-CD45 (BioLegend; 103111) antibodies were purchased and used for functional assay of RBK21.

### Parasite culture

*P. falciparum* was cultured with human RBCs (type O blood; haematocrit 2%) in complete medium. This medium consisted of RPMI 1640 medium containing 10% human serum and 10% AlbuMAX I (Invitrogen), along with 25 mM HEPES, 0.225% sodium bicarbonate and 0.38 mM hypoxanthine. The medium was supplemented with 10 mg ml$^{-1}$ of gentamicin. The cultures were maintained in an atmosphere containing 90% $N_2$, 5% $CO_2$ and 5% $O_2$. Parasites were routinely synchronized every 4 days using a 5% sorbitol solution and subsequently used for analyses.

### Cloning of RBK21 cDNA

The iRBCs that bound to KIR2DL1-Fc were enriched using the SH800 cell sorter (Sony). Enriched iRBCs were cultured, and total RNA was purified using TRIzol (Thermo Fisher Scientific). Subsequently, cDNA was synthesized using SuperScript III Reverse Transcriptase (Thermo Fisher Scientific) and random hexamers according to the manufacturer's instructions. The variable regions of RIFIN expressed in enriched iRBCs were amplified using the primers listed in Supplementary Table 3. The obtained fragment, which encoded a part of the *RBK21* gene, was cloned into the centromere plasmid[35] (pFCEN-rif), resulting in a chimaeric RIFIN. Expression was controlled by the promoter of elongation factor α of *Plasmodium berghei*. Before cloning the full-length RBK21 cDNA, the 5′-end and 3′-end of RBK21 cDNA were analysed using 5′-Full and 3′-Full RACE Core Sets (Takara) with specific primers (Supplementary Table 3). Following sequencing of the amplified product, a DNA fragment encompassing the entire coding region of RBK21 was obtained by means of polymerase chain reaction (PCR). The amplified fragment was cloned in the pFCEN-rif and used for further analysis.

### Transfection of *P. falciparum*

Transfection of *P. falciparum* was performed, as described previously[36]. In brief, schizonts of the *P. falciparum* 3D7 strain were purified using a Percoll/sorbitol gradient solution and cultured with fresh RBCs for 4 h. Subsequently, the parasites were synchronized within a 4-h window by treatment with 5% sorbitol. Full mature schizonts were purified from highly synchronized parasites and then transfected with pFCEN-rif

plasmids using an Amaxa Nucleofector 2 with Parasite Nucleofector Kit 2 (Lonza). Because pFCEN-rif has human dihydrofolate reductase as a drug-selectable marker, transgenic parasites could be selected by drug treatment with pyrimethamine. Transfection experiments for each plasmid were carried out in duplicate throughout this study, resulting in biologically independent transgenic parasites.

### Production of KIR-Fc proteins for library screening

The coding regions of KIR receptors KIR2DL1*00302 (Gene ID 3802; amino acid residues 22–245), KIR2DS1*001 (Gene ID 3806; amino acid residues 22–245), KIR2DL2*001 (Gene ID 3803; amino acid residues 22–245), KIR2DL3*001 (Gene ID 3804; amino acid residues 22–245), KIR2DL5*001 (Gene ID Q14953; amino acid residues 22–239), KIR3DL1*001 (Gene ID 3811; amino acid residues 20–339), KIR3DL2*001 (Gene ID 3812; amino acid residues 22–341) and KIR3DL3*001 (UniProt ID Q8N743; amino acid residues 25–320) were cloned into the vector pCAGSS. The resultant plasmids were introduced into HEK293T cell using TransIT-293 (Mirus Bio), and these KIRs were expressed as secreted human-Fc fusion proteins. These KIRs-Fc fusion proteins were obtained from supernatants of transfected HEK293 cells. To validate the correct folding of KIR-Fc fusion proteins, we used an established flow cytometry-based bead assay[37]. In brief, anti-KIR3DL2 antibody (BioLegend; 389602) or KIR-Fc fusion proteins were conjugated to Dynabeads M-450 Tosylactivated (Invitrogen; 14013) according to the manufacturer's instructions. KIR-Fc beads were then blocked with PBS + 1% w/v bovine serum albumin (BSA) for 10 min under agitation before staining with the following antibodies: FITC anti-KIR2DL1/DL5 (R&D Systems; FAB1844F), PE anti-KIR2DL2/DL3/DS2 (BioLegend; 312605), PE anti-KIR2DL5 (Miltenyi Biotec; 130-096-199), FITC anti-KIR3DL1 (BioLegend; 312705), anti-KIR3DL2 (BioLegend; 389602) and anti-KIR3DL3 (R&D Systems; FAB8919P). Anti-KIR3DL2 beads were blocked with PBS + 1% w/v BSA for 10 min before staining with KIR3DL2-hFc for 20 min under agitation. If secondary detection was required, the beads were washed twice with cold PBS before staining with either 2 µg ml$^{-1}$ of AF647-labelled goat anti-human F(ab')2 (Jackson ImmunoResearch; 709-605-098) or AF647-labelled donkey anti-mouse F(ab')$_2$ (Jackson ImmunoResearch; 715-606-150) for 20 min under agitation. The beads were then washed twice in cold PBS before analysis on BD LSR II using BD FACSDiva software.

### Flow cytometry

The binding of KIRs-Fc fusion proteins to iRBCs was assessed by flow cytometric analysis using an Attune NxT (Thermo Fisher Scientific). Before the assay, the KIRs-Fc fusion proteins were mixed with an APC-conjugated anti-human IgG Fc antibody. The iRBCs were mixed with the APC-labelled KIRs-Fc binding proteins, and parasite nuclei were stained with Hoechst 33342. The iRBC populations were gated on the basis of fluorescence from Hoechst 33342, and KIR-Fc-bound iRBCs were selected using APC fluorescence (Supplementary Fig. 2). All assays were performed in triplicate, and all data were analysed using FlowJo software (Becton Dickinson).

The iRBCs binding to KIR2DL1 were selectively sorted using the SH800 cell sorter (Sony). Before sorting, iRBCs containing late trophozoites and schizonts were separated from uninfected RBCs using the Percoll/sorbitol gradient solution, followed by mixing with APC-labelled KIR2DL1-Fc fusion protein. After binding KIR2DL1-Fc to iRBC, they were suspended in complete medium and subjected to SH800. The sorted iRBCs were immediately recovered in complete medium and cultured with fresh RBCs.

### Preparation of RIFIN expression library

DNA fragments that encoded all RIFIN variable regions were amplified from the genomic DNA of the 3D7 strain using degenerated primers designed at the internal sites of the N-terminal and C-terminal conserved regions of RIFINs (Supplementary Table 3). The amplified

fragments representing the variable regions of all RIFINs of the 3D7 strain were cloned into the *Bsm*BI sites of the pFCEN-rif plasmid (Supplementary Fig. 1). The cloned fragments were flanked with the N-terminal and C-terminal conserved regions of pFCEN-rif, resulting in chimaeric *RIFIN* genes. The pFCEN-rif plasmids containing chimaeric *RIFIN* genes were introduced into the *P. falciparum* 3D7 strain through single electroporation. Transgenic parasites were subsequently obtained by means of treatment with pyrimethamine, resulting in the creation of a RIFIN expression library. Each parasite within the RIFIN expression library expressed distinct chimaeric RIFINs. Library construction was performed twice, resulting in the establishment of two biologically independent RIFIN expression libraries, designated rif-lib1 and rif-lib2.

The RIFIN expression libraries from Lek174 and Lek79 were generated in a manner similar to that of the 3D7 strain. Briefly, DNA fragments encoding the RIFIN variable region were amplified using the degenerated primers (Supplementary Table 3), cloned into the pFCEN-rif plasmid and introduced into the 3D7 strain. Two biologically independent libraries were generated for each of the two field-isolated parasite strains and were used for screening using KIR2DL1-Fc.

## Screening for KIR-binding RIFINs from expression libraries

The iRBCs of rif-lib1 and rif-lib2 were incubated with the KIR2DL1-Fc fusion protein and screened using the cell sorter. The cells were cultured in a complete medium immediately after sorting. To identify the selected RIFINs, the pFCEN-rif plasmids containing chimaeric *RIFIN* genes were recovered from the sorted iRBCs. The variable regions integrated into the plasmids were subsequently amplified by PCR using primers designed on the basis of the plasmid backbone sequence and recovered plasmid as a template (Supplementary Table 3). Following this, sequence tags were introduced at the ends of the amplified DNA fragments by means of PCR, and the resultant tagged DNA fragments were sequenced using MiSeq (Illumina). The reads containing the variable regions were isolated from the obtained FASTQ files using the SeqKit program[38] with the 'grep' option. Subsequently, the plasmid backbone sequence was trimmed using the Trimmomatic 0.39-2 program[39]. After confirming the sequence quality of the trimmed reads, they were mapped onto the reference genome sequence of the *P. falciparum* 3D7 strain, which was downloaded from PlasmoDB 62 using the bowtie2 tool[40]. Reads aligned to more than two different genomic loci were eliminated from the mapping data. The number of reads for each RIFIN gene was tallied using featureCounts[41] and normalized by dividing it by the total number of mapped reads. The normalized read count for each RIFIN gene was further divided by the normalized read count obtained before screening. Candidates for KIR2DL1-binding RIFINs were identified on the basis of values exceeding 2. Each variable region of a candidate RIFIN was amplified from the genomic DNA of the *P. falciparum* 3D7 strain and individually cloned into pFCEN-rif. Transgenic parasites were generated by transfecting the resultant plasmids into the 3D7 strain, and binding to KIR2DL1 was assessed using Attune NxT and Fc fusion protein.

The expression libraries for RIFINs from Lek174 and Lek79 were selected using a method similar to that used for 3D7. The variable regions of the candidate RIFINs were amplified from the recovered pFCEN-rif and tagged, followed by sequencing using MiSeq. The obtained sequence reads were processed using the SeqKit and Trimmomatic 0.39-2 program in a similar manner to those from rif-lib1 and rif-lib2. The genomes of Lek174 and Lek79 were sequenced using MinION (Oxford Nanopore Technologies) and MiSeq, and their genomic contigs were generated from the obtained long reads and short reads using Flye[42], BWA[43,44] and Pilon[45] programs. The processed sequenced reads of variable regions were then mapped onto the generated contig sequences, and the sequence information for the region of the contig to which the reads were mapped was manually obtained, together with 1,000 bp upstream and downstream. The general feature format files were generated using the sequence information of the region to which reads were mapped and contig ID, and the number of mapped reads

was counted using featureCounts 2.0.1 program, followed by ranking the regions containing mapped reads on the basis of the number of reads. The top 20 regions were selected, and ORF Finder (https://www.ncbi.nlm.nih.gov/orffinder/) was used to analyse these regions and identify the variable regions of the KIR2DL1-binding RIFN candidates. All the experiments were conducted using two biologically independent libraries for each of two strains.

## Phylogenetic analysis of RIFINs

The amino acid sequences of all RIFINs of the *P. falciparum* 3D7 strain were obtained from PlasmoDB (https://plasmodb.org/plasmo/app/) and aligned using the Clustal Omega program (https://www.ebi.ac.uk/jdispatcher/msa/clustalo). Regions equivalent to the variable region of the LILRB1-binding RIFN were selected (PF3D7_1254800)[3]. These were aligned using Clustal Omega, followed by the generation of a phylogenetic tree on the basis of the newly aligned sequences. All analyses were performed with default setting. The resultant data from the phylogenetic analysis were visualized using the TreeView program[46]. The amino acid sequences of the variable regions of KIR2DL1-binding RIFIN candidates from Lek174 and Lek79 were obtained as described above and analysed together with the variable regions of all RIFINs from the 3D7 strain. For prediction of KIR2DL1-binding RIFIN clades in field isolate genomes, 3D7 KIR2DL1-binding RIFIN sequences were combined with the RIFIN repertoire of either *Pf*KE01 or *Pf*SN01, as obtained using PlasmoDB, and the same multiple sequence alignment approach was used, as described above.

## Transfection of mammalian cells

Stable transfectants of K562–RBK21, K562-negative control rifin (PF3D7_1254200), NKL–KIR2DL1 and NKL–KIR2DL3 were generated using retrovirus-mediated transduction with the pMXs retroviral expression vector and PLAT-E retroviral packaging cells transfected with the amphotropic envelope, as described previously[2,47] (Cell Biolabs). Briefly, the variable regions of these RIFINs were fused with the transmembrane and cytoplasmic regions of PILRα (amino acid residues 196–256) and cloned into the pMXs plasmid. The resultant plasmid containing the fusion gene of RBK21 and PILRα was introduced into PLAT-E cells using TransIT-293 (Mirus Bio), and the recombinant retrovirus was collected from the supernatant 3 days after transfection. The full-length cDNA of KIR2DL1 and KIR2DL3 was chemically synthesized (GenScript) and subsequently cloned into the pMXs plasmid. Recombinant retrovirus was then produced using a method similar to that described above. The target cells (K562 and NKL) were seeded in a 24-well culture plate and infected with the produced recombinant virus, resulting in K562–RBK21, K562-negative control rifin, NKL–KIR2DL1 and NKL–KIR2DL3.

## Reporter assay

The extracellular domains of KIR2DL1 and KIR2DS1 were fused with PILRβ and expressed on the surface of mouse T cell hybridomas that were stably transfected with NFAT–GFP and FLAG-tagged DAP12, as described previously[48]. The transmembrane domain of PILRβ can transduce the signal through the DAP12 adaptor molecule in the reporter cell, resulting in the expression of GFP. The reporter cell lines, which expressed fusion proteins of KIR2DL1 and KIR2DS1 with PILRβ, were stimulated by iRBCs, which were infected with parasites expressing RBK21 for 16 h, and their GFP expression, which was selected using anti-CD45 antibody, was monitored using Attune NxT. The *P. falciparum* 3D7 strain was used as a negative control.

## Measurement of CD107a, IFNγ and TNF in NK cells

NK cells were purified using an NK cell isolation kit (Miltenyi Biotec), according to the manufacturer's instructions, from PBMCs obtained from two donors positive for KIR2DL1 and two donors positive for KIR2DS1. Following purification, the NK cells were cultured in the NK

cell growth medium, which consisted of RPMI 1640 supplemented with 15% FCS (Gibco), 5% human serum, 1× minimum essential medium non-essential amino acids, 1 mM sodium pyruvate, 100 U ml$^{-1}$ of penicillin, 100 μg ml$^{-1}$ of streptomycin, 500 U ml$^{-1}$ of human interleukin (hIL)-2, 5 ng ml$^{-1}$ of hIL-15, 10 ng ml$^{-1}$ of hIL-12, 40 ng ml$^{-1}$ of hIL-18 and 20 ng ml$^{-1}$ of hIL-21. To evaluate the inhibitory effect of RBK21 on NK cell function, purified NK cells ($1.0 \times 10^5$ cells) containing KIR2DL1-positive NK cells were co-cultured with K562–RBK21 cells ($1.0 \times 10^5$ cells) in a 96-well plate at 37 °C for 4 h. Pacific Blue-conjugated anti-human CD107a antibody was added to the medium at the beginning of co-culturing for the measurement of CD107a expression. After 1 h of co-culture, the NK cells were treated with BD GolgiStop reagent, which is included in the BD Cytofix/Cytoperm Fixation/Permeabilization Kit containing monensin (BD Biosciences). The NK cells were collected by centrifugation, and dead cells were stained at room temperature for 30 min using a LIVE/DEAD Fixable Yellow Dead Cell Stain Kit (Invitrogen). The NK cells were further stained on ice for 30 min with FITC-conjugated anti-CD56 antibody, PE-labelled anti-KIR2DL1/DS1 antibody and APC-labelled anti-KIR2DL1/DS5 antibody. The cells were fixed and permeabilized at 4 °C for 20 min with a fixation/permeabilization solution, which was also included in the kit. The permeabilized NK cells were then stained at 4 °C for 30 min in the dark with PerCP/Cy5.5-conjugated anti-human IFNγ antibody and APC/Cy7-conjugated anti-human TNF antibody using the BD Perm/Wash buffer. CD107a expression and production of IFNγ and TNF were assessed using Attune NxT. The K562 and K562-negative control RIFIN (PF3D7_1254200) were used as negative controls. To assess the activation of NK cells through KIR2DS1 by RBK21, NK cells containing KIR2DS1-positive NK cells were co-cultured with RBCs infected with RBK21-expressing transgenic parasites. The 3D7 strain and transgenic parasites expressing a negative control RIFIN were used as negative controls. CD107a expression and IFNγ and TNF production were assessed, as described above. All assays were performed in six or nine replicates, and representative raw flow cytometry data are shown in Extended Data Figs. 5 and 8.

## Cytotoxicity assay

Suppression of the cytotoxic activity of NKL–KIR2DL1 cells by RBK21 was examined, as described previously[2]. The viabilities of K562–RBK21 and NKL–KIR2DL1 were assessed before the assay. If the viability was lower than 85%, the cells were purified using Ficoll and cultured for 3 days. The highly viable K562–RBK21 was labelled with 15 μM calcein acetoxymethyl ester (calcein-AM) in assay medium (RPMI 1640 without phenol red, supplemented with 1% FCS) for 30 min at 37 °C, followed by washing twice with the assay medium. The NKL–KIR2DL1 cells were washed twice with the assay medium and mixed with K562–RBK21 in a 96-well plate with effector (NKL–KIR2DL1) to target (K562–RBK21) ratios ranging from 1:1 to 1:8 in triplicate. These cells were centrifuged at 250$g$ for 5 min and then co-cultured for 4 h at 37 °C. To detect the maximal release of calcein-AM from K562–RBK21, cells were dispensed without NKL–KIR2DL1 in the plate, followed by lysis with 1% Triton X-100 for 30 min before the end of co-incubation. In addition, the cells were cultured without NKL_KIR2DL1, and the spontaneous release of calcein-AM in the supernatant was measured. After co-incubation, the cells were centrifuged at 1,500 rpm for 2 min, and the fluorescence of the released calcein-AM was measured. Specific cytotoxicity $C$ (%) was calculated using the following formula: $C = 100 \times$ (mean fluorescence in co-culture − mean fluorescence in spontaneous lysis)/(mean fluorescence in maximal lysis − mean fluorescence in spontaneous lysis). K562 and K562-negative control RIFIN (PF3D7_1254200) were used as positive controls. In addition, NKL and NKL–KIR2DL3 were tested. All assays were performed in quadruplicate.

## Nanobody identification and screening

To identify KIR2DL1-binding nanobodies, a complex of KIR2DL1 (residues 27–221) and RBK21 variable domain (residues 148–299)

was produced and purified by size exclusion chromatography using a Superdex 75 10/300 column (Cytiva). After three rounds of llama immunization, a VHH cDNA library was generated, and nanobodies were screened in a phage–enzyme-linked immunosorbent assay approach using a biotinylated complex, adapted from a previous study[49]. Hits from the enzyme-linked immunosorbent assay screen were sequenced, and non-redundant nanobody sequences were expressed using periplasmic expression in a WK6 *Escherichia coli* strain. The VHH domain was flanked by an N-terminal PelB leader sequence and a C-terminal His$_6$-tag in a pET15b vector, and the nanobodies were expressed and purified, as detailed in a previous study[49]. The nanobodies were then screened by means of SPR, and the nanobody with the most favourable kinetic profile was selected for grafting onto a NabFab nanobody scaffold[50]. The resultant nanobody (Nb1) was expressed as previously but with the addition of a final polishing step using a Superdex 75 10/300 column (Cytiva).

## Protein expression and purification

To produce proteins for structural biology, the coding sequence for KIR2DL1 (residues 27–221) was cloned into the vector pHL-sec, including a C-terminal His$_6$-tag, and transfected into *Mycoplasma*-screened Expi293F GnTI- cells using ExpiFectamine 293 (Thermo Fisher Scientific). Mutagenesis of RBK21 constructs was carried out using the Q5 Site-Directed Mutagenesis Kit (New England Biolabs), following the manufacturer's instructions. Five days after transfection, the supernatant was collected by centrifugation at 5,000$g$ to pellet the cells before filtering with a bottle-top 0.45 μM filter. The pH was adjusted to 7.5 through the addition of Tris to a final concentration of 50 mM before passing over the Ni Sepharose excel resin (Cytiva). The captured KIR2DL1 protein was washed with 50 mM Tris (pH 7.5), 150 mM NaCl, 20 mM imidazole before elution with 50 mM Tris (pH 7.5), 150 mM NaCl and 500 mM imidazole. The protein was incubated with Endo Hf overnight at 37 °C to cleave glycans.

The RIFIN variable domain constructs (RBK21 residues 148–299 and KEN-01 residues 159–288 with A165C and A282C mutations) were cloned into a modified pENTR4LP vector, including a cleavable N-terminal monomeric Fc domain tag and a C-terminal C-tag before transfection into Expi293F GnTI- cells, as described above. The supernatant was processed, as described above, and flowed over the CaptureSelect C-tagXL Affinity Matrix (Thermo Fisher Scientific) to isolate the C-tagged protein. The captured RBK21-mFc was washed with 50 mM Tris (pH 7.5) and 150 mM NaCl and eluted with 2 M MgCl$_2$ and 20 mM Tris (pH 7.5) before buffer exchange with 50 mM Tris (pH 7.5) and 150 mM NaCl. The protein was incubated with Endo Hf and tobacco etch virus protease overnight at 37 °C to cleave glycans while simultaneously releasing the variable domain from the mFc tag. The RIFIN was flowed over Pierce Protein G Agarose to remove the mFc fusion tag, and the flow-through was collected before further purification by size exclusion chromatography using a Superdex 75 10/300 column (Cytiva).

The HLA-Cw4–β2m complex used in the SPR experiments was produced using a refolding protocol, as described previously[51]. In brief, HLA (accession no. MH254935) and β-2-microglobulin (β2M; accession no. NM_004048) were expressed separately in *E. coli* BL21(DE3) pLysS and purified as inclusion bodies. These were solubilized in a buffer containing 6 M guanidine HCl, 100 mM Tris-HCl (pH 8.0), 2 mM EDTA and 0.1 mM dithiothreitol. Solubilized HLA and β2m were diluted to 10 mg ml$^{-1}$ using solubilization buffer, with the addition of 1 M dithiothreitol to a final concentration of 10 mM, and the solutions were incubated at room temperature for 1 h. Refolding buffer (100 mM Tris (pH 8.0), 400 mM L-Arg, 2 mM EDTA, 3.73 mM cystamine and 6.73 mM cysteamine) was chilled to 4 °C before β2m was refolded by rapid dilution, to a final concentration of 2 μM. Refolding was allowed to proceed for 2 h before the addition of 10 mg l$^{-1}$ of peptide (QYDDAVYKL), and then HLA C was added dropwise to a final concentration of 2 μM. The heterotrimer was allowed to refold for 72 h at 4° before size exclusion

chromatography (HiLoad 26/600 Superdex 75 pg) in 20 mM Tris-HCl (pH 8.0) and 100 mM NaCl.

## Crystallization, data collection and structure determination

KIR2DL1 and RBK21 were combined to a 1:1 molar ratio, and the resulting complex was purified using a Superdex 75 10/300 column (Cytiva) in 20 mM HEPES and 150 mM NaCl. Initial crystallization trials were carried out using vapour diffusion in sitting drops by mixing 100 nl of protein solution (10 mg ml$^{-1}$) with 100 nl of well solution using commercial screens. For RBK21–KIR2DL1, initial hits were obtained after 10 days at 18 °C in condition H12 (0.1 M Tris-bicine (pH 8.5), 0.1 M amino acid mix and 37.5% v/v precipitant mix 4) of the Morpheus HT-96 screen (Molecular Dimensions). Using these initial crystals as seeds, an optimization screen was set up using the Morpheus stock solutions, screening a pH range from 8.0 to 9.0 and a final precipitant concentration between 21% and 28% by mixing 100 nl of protein solution with 100 nl of well solution and 25 nl of seed stock. The best crystals were obtained after 12 days with a well solution of 0.1 M Tris-bicine, 0.1 M amino acid mix and 23% precipitant mix 4 and were collected then cryo-cooled for data collection in liquid nitrogen.

Data were collected on beamline ID30A-3 at ESRF at a wavelength of 0.97625 Å, indexed using DIALS and scaled using AIMLESS, resulting in a full dataset with a final resolution of 2.89 Å. The structure was solved by means of molecular replacement using KIR2DL1 (PDB 1IM9) as a search model using Phaser-MR (v.2.8.3)[52]. Building and refinement cycles were carried out using Coot (v.0.8.9.2)[53] and BUSTER (v.2.10)[54].

For KEN-01, a complex was formed using a 1:1.2:1.5 ratio of KEN-01:KIR2DL1:Nb1, and an equimolar complex was obtained through purification on a Superdex 75 10/300 column (Cytiva) in 10 mM HEPES and 75 mM NaCl. The same preliminary screening strategy was implemented as described above, and crystals were obtained after 12 days at 18 °C in condition E3 of the PEG/ION-HT screen (Hampton Research; 0.1 M sodium malonate (pH 5.0) and 12% PEG 3350). These crystals were collected in mother liquor + 25% glycerol before cryo-cooling in liquid nitrogen. Data were collected on beamline i03 using a Diamond Light Source at a wavelength of 0.976246 Å. Indexing and data reduction were performed using xia2 DIALS, resulting in a full dataset with a final resolution of 2.17 Å. The previously solved KIR2DL1 and RBK21 structures were used as molecular replacement search models to solve the new dataset (Phaser-MR v.2.8.3 (ref. 52)) and preceded cycles of building and refinement using Coot and BUSTER.

## Surface plasmon resonance

All experiments were conducted on a Biacore T200 instrument (Cytiva) using a running buffer of 20 mM HEPES (pH 7.5), 300 mM NaCl and 0.005% v/v Tween-20. Proteins were desalted in this buffer using PD-10 columns (Cytiva). Sensorgrams were double referenced by subtraction of the response measured from a blank flow path with no protein immobilized, in addition to subtraction of the response attributable to buffer from the protein flow path. Kinetic values were obtained using the BIAevaluation software (Cytiva) by fitting data to a global 1:1 interaction model, allowing for the determination of the association rate constant ($k_{on}$), dissociation rate constant ($k_{off}$) and affinity ($K_D$). Equilibrium fits of the multicycle experiments were obtained using a 1:1 interaction model in BIAevaluation (Cytiva). All kinetic and equilibrium fits are contained within the Source Data.

For experiments comparing its binding affinity for KIR2DL1 and KIR2DS1, RBK21 was coupled to a CM5 sensor (Cytiva) through amine chemistry at approximately 150 response units (RU). A twofold dilution series of KIR2DL1 and KIR2DS1 was injected over the chip for 240 s at a flow rate of 30 µl min$^{-1}$, followed by a 300-s dissociation time and regeneration between cycles with 10 mM glycine (pH 2.5) for 10 s. Affinity values were derived from both equilibrium and kinetic fits to the data.

To measure HLA-Cw4 binding, approximately 300 RU of each Fc-tagged KIR2DL1 and KIR2DS1 was captured through Protein A/G

(Thermo Fisher Scientific) pre-immobilized on a CM5 sensor (approximately 1,500 RU on each flow cell). The same association and dissociation times as described above were used. Regeneration of the surface was carried out with 10 mM glycine (pH 2.5) before more KIR2DL1 and DS1 Fc-tagged protein were re-immobilized on the chip, and the next cycle was started. Affinity values for these data could only be fit using equilibrium measurements because of fast on-rate and off-rate, which make kinetic fits unreliable.

Comparison of the binding of wild-type and S221R RBK21 to KIR2DL1 was achieved through immobilization of approximately 300 RU of KIR2DL1 onto the surface of a CM5 sensor followed by injection of a twofold dilution series of RBK21 variants starting from 20 µM over the sensor surface at a flow rate of 30 µl min$^{-1}$. The association and dissociation times were 60 and 120 s, respectively, with regeneration with 10 mM glycine (pH 2.5) for 10 s between cycles.

RIFINs from field-isolated strains were screened for KIR2DL1/DS1 binding through immobilization of KIR2DL1 and KIR2DS1 and a negative control of LILRB1 to a CM5 chip sensor (Cytiva) using amine chemistry (500, 500 and 1,000 RU immobilized, respectively). The mFc-tagged RIFINs were exchanged into a running buffer of 20 mM HEPES (pH 7.5), 300 mM NaCl and 0.005% v/v Tween-20 and flowed over the sensor in multicycle experiments, as described above.

For competition experiments, a paired test was performed under saturating conditions. KEN-01 was flowed over the sensor either alone (5 µM) or mixed with another tested protein (both proteins at 5 µM). Response was measured 5 s before the dissociation phase, and all responses were expressed as a percentage increase relative to 5 µM of KEN-01 alone. A positive control of KEN-01 and Nb1 demonstrated how a known non-competitive binder resulted in a marked increase in response, whereas RBK21, a known competitive binder, showed no significant increase. Paired tests were performed with all field-isolated RIFINs in triplicate.

## Circular dichroism

Proteins were desalted into 10 mM sodium phosphate (pH 7.5) and 100 mM NaF using PD-10 columns (Cytiva). Measurements were made on a JASCO J815 CD spectrophotometer. Experiments were recorded at 20 °C between 190 and 260 nm at 0.5-nm intervals with a protein concentration of 0.1 mg ml$^{-1}$ using a 1-mm path-length cuvette (Hellma Macro Cell 110-QS). Three equivalent protein spectra were recorded and averaged after subtraction of a buffer-only blank measurement. Data were processed using the CAPITO online web server[55].

## Shannon entropy calculation

The variable region from each KIR2DL1-binding RIFIN identified in the clade was used to generate a multiple sequence alignment using MUSCLE[56]. The multiple sequence alignment was then provided to the Protein Variability Server[57] to calculate entropies. Per-residue entropy values were then binned into three categories: absolute conservation, high conservation and medium conservation with entropy values of 0, 0–0.5 and 0.5–0.8, respectively. These three bins were mapped onto the structure in PyMOL.

## Supported lipid bilayer assay

SLB experiments were performed, as described previously[3]. Briefly, 1,2-dioleoyl-*sn*-glycero-3-phosphocholine (Avanti Polar Lipids) micelles were supplemented with 12.5% 1,2-dioleoyl-*sn*-glycero-3-[(*N*-(5-amino-1-carboxypentyl) iminodiacetic acid) succinyl]-Ni (Avanti Polar Lipids) and infused into plasma-cleaned glass coverslips affixed within a six-lane adhesive chamber (ibidi) for 20 min. SLBs were washed three times with HEPES buffered saline + 0.1% BSA + 1 mM CaCl$_2$ + 2 mM MgCl$_2$ and blocked with 100 µM NiSO$_4$ in 5% BSA/PBS. After another washing step, protein dilutions were added to achieve 600 molecules µm$^{-2}$ ICAM-1 and 100 molecules µm$^{-2}$ PfRH5, with or without 100 molecules µm$^{-2}$ of the indicated RIFIN, as determined by flow cytometry

with bead-supported bilayers. The protein mixtures were incubated for 20 min to allow for attachment and then washed. Monoclonal antibody R5.016 was added at 2 µg ml$^{-1}$ for 20 min followed by washing. Then, $10^6$ NK cells, isolated from fresh blood samples using a RosetteSep Human NK Cell Enrichment Cocktail (STEMCELL Technologies), were infused into each lane, followed by incubation for 30 min at 37 °C. The bilayers were then fixed for 5 min in 4% paraformaldehyde/Hank's balanced salt solution, followed by washing. Perforin staining was performed using monoclonal anti-perforin Alexa Fluor 488 (clone dG9; BioLegend) at a concentration of 10 µg ml$^{-1}$ for at least 1 h. The bilayers were washed three times before image acquisition.

Imaging was performed, as described previously[3], using an Olympus cell TIRF-4Line system with a ×150 (numerical aperture: 1.45) oil objective at room temperature. We analysed images using ImageJ (v.1.54b; National Institutes of Health). Cell boundaries were defined on the basis of segmented ('default' algorithm in ImageJ) interference reflection images[58].

## Homology modelling of KIR2DS1

The SWISS-MODEL[59] web interface was used to generate a template homology model of KIR2DS1 for structural comparisons. The protein sequence coding for domains D1 and D2 of KIR2DS1 was inputted into the tool with a PDB template supplied by KIR2DL1 from our crystallographic data.

## Reporting summary

Further information on research design is available in the Nature Portfolio Reporting Summary linked to this article.

## Data availability

Crystallographic data were deposited in the Protein Data Bank with accession codes 9F2D and 9HML. Sequence data related to rif-lib1 and rif-lib2 were deposited at NCBI Gene Expression Omnibus with accession no. GSE286478. All materials are available from the authors. Source data are provided with this paper.

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

**Acknowledgements** This study was funded by the Wellcome Trust Collaborative Award (224343/Z/21/Z), Japan Agency for Medical Research and Development (223fa627002h, 24gm1810006h and 24ek0410124h), Japan Science and Technology (CREST; JPMJCR23B5), Japan Society for the Promotion of Science (22H04989) and Kennedy Trust for Rheumatology Research Cell Dynamics Platform. This study was supported by the Nippon Foundation. We thank R. Owens and J. Huo from the Rosalind Franklin Institute for support with llama immunization and nanobody cloning. We would like to thank E. Lowe and the beamline scientists at Diamond Beamline I03 and the European Synchrotron Radiation Facility Beamline ID30A-3 for their help with the crystallographic data collection.

**Author contributions** A.S., S.G.C., H.A., M.K.H. and S.I. designed and conceived the study. A.S., H.A. and S.I. identified RBK21 and conducted the functional analysis. A.S. and S.I. created RIFIN expression libraries and identified the KIR2DL1-binding RIFINs from the libraries. T.E.H. produced protein. S.G.C. produced and tested RIFINs from African strains and conducted all other structural and biophysical analyses. A.M.M. and M.W. performed the SLB assay. A.S., S.G.C., H.A., M.K.H. and S.I. wrote the paper. All authors contributed to the revision of the paper.

**Competing interests** The authors declare no competing interests.

**Additional information**
**Correspondence and requests for materials** should be addressed to Hisashi Arase, Matthew K. Higgins or Shiroh Iwanaga.

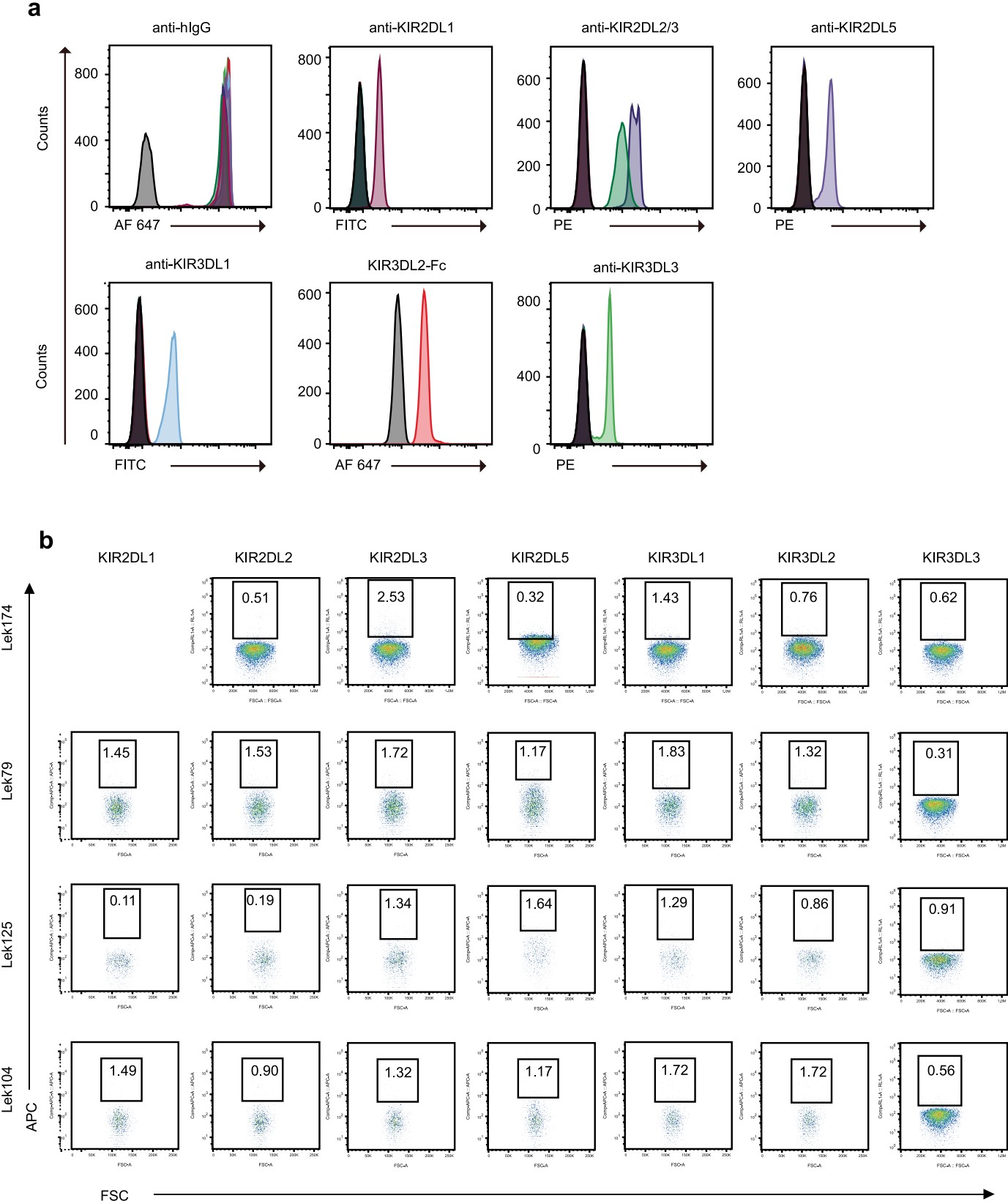

**Extended Data Fig. 1 | Identification of KIR2DL1-binding iRBCs. a)** The integrity of each KIR-Fc protein was validated using conformational antibodies targeting each KIR[37]. Beads coated with KIR-Fc proteins are probed with anti-human-IgG and labelled-antibodies specific for KIR2DL1, DL2/DL3, DL5, 3DL1, and 3DL3. In the case of KIR3DL2, the assay was reversed such that the antibody was immobilized on the bead and probed with KIR3DL2-Fc, prior to secondary staining. **b)** The binding of 4 field-isolated parasites to each fluorescently labelled inhibitory KIR-Fc protein, assessed by flow sorting. The binding of Lek174 strain to KIR2DL1-Fc is shown in Fig. 1a. The positive iRBCs are highlighted in rectangles with the percentage of positive cells indicated.

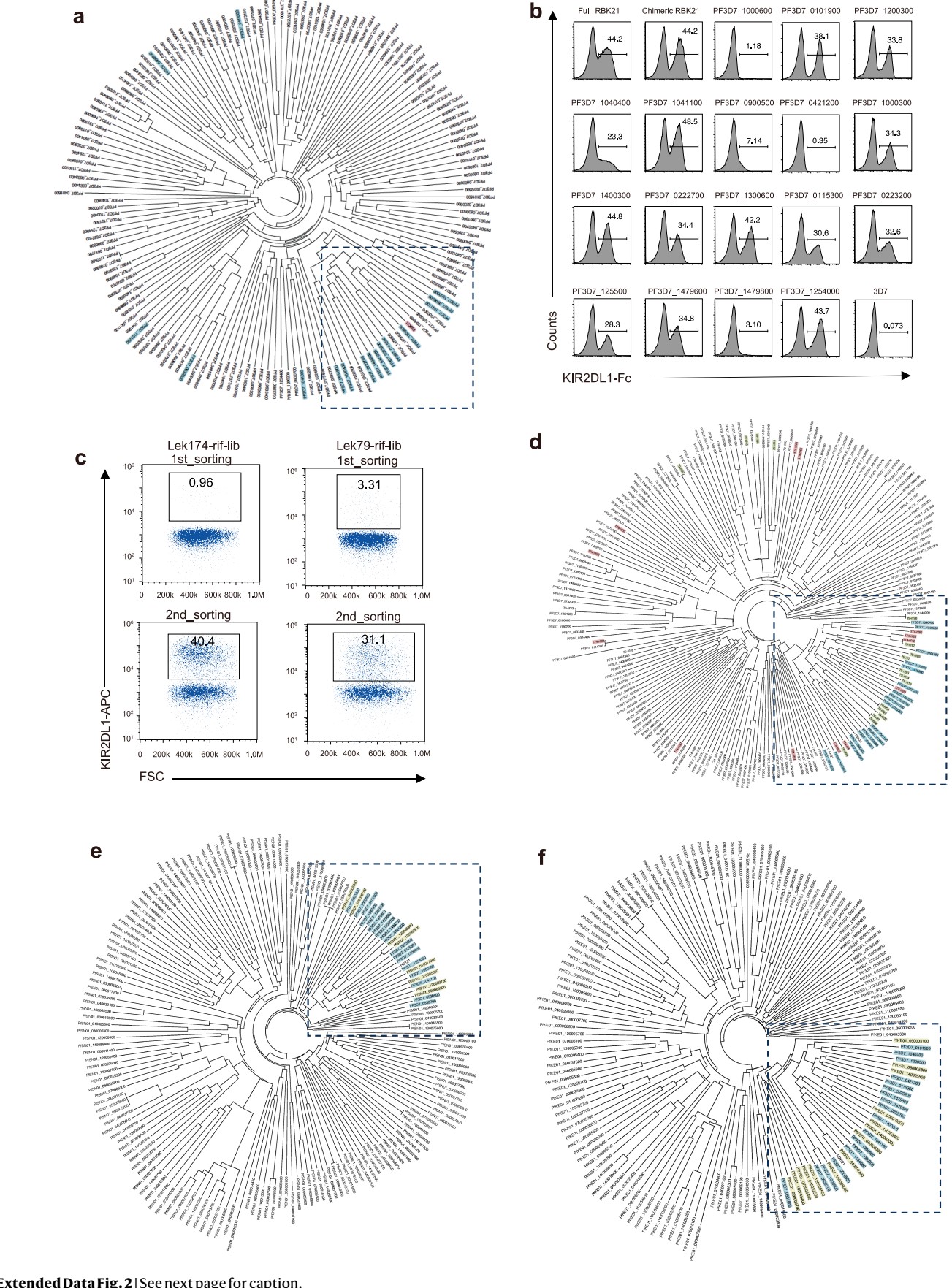

**a**

**b**

| Full_RBK21 | Chimeric RBK21 | PF3D7_1000600 | PF3D7_0101900 | PF3D7_1200300 |

44.2 | 44.2 | 1.18 | 38.1 | 33.8

| PF3D7_1040400 | PF3D7_1041100 | PF3D7_0900500 | PF3D7_0421200 | PF3D7_1000300 |

23.3 | 48.5 | 7.14 | 0.35 | 34.3

| PF3D7_1400300 | PF3D7_0222700 | PF3D7_1300600 | PF3D7_0115300 | PF3D7_0223200 |

44.8 | 34.4 | 42.2 | 30.6 | 32.6

| PF3D7_125500 | PF3D7_1479600 | PF3D7_1479800 | PF3D7_1254000 | 3D7 |

28.3 | 34.8 | 3.10 | 43.7 | 0.073

Counts

KIR2DL1-Fc

**c**

Lek174-rif-lib 1st_sorting — 0.96

Lek79-rif-lib 1st_sorting — 3.31

2nd_sorting — 40.4

2nd_sorting — 31.1

KIR2DL1-APC

FSC

**d**

**e**

**f**

**Extended Data Fig. 2** | See next page for caption.

**Extended Data Fig. 2 | Phylogenetic trees of RIFINs from 3D7 and field-isolated strains. a)** Phylogenetic tree of all RIFINs from the 3D7 strain with KIR2DL1-binding RIFINs, which are identified from rif-lib1 and -2, highlighted in blue. The dashed box indicates the clade of KIR2DL1-binding RIFINs. **b)** Assessment of the binding of iRBCs which express predicted KIR2DL1-binding RIFINs to fluorescent KIR2DL1-Fc by flow sorting with parental 3D7 as a negative control. All tested RIFINs are classified in the same clade in phylogenetic tree (Fig. 1e and Extended Data Fig. 3). The percentage of positive iRBC is indicated. **c)** Flow sorting plots showing two rounds of sorting of iRBCs expressing KIR2DL1-binding RIFINs, from Lek174-rif-lib and Lek79-rif-lib, using fluorescent KIR2DL1-Fc. Positive cells are shown in the rectangle and the percentage of positive cells is indicated. **d)** Phylogenetic analysis of KIR2DL1-binding RIFIN candidates identified from Lek174-rif-lib (red) and Lek79-rif-lib (green), analysed together with KIR2DL1-binding RIFINs the 3D7 (blue). The dashed box indicates the clade of KIR2DL1-binding RIFINs of 3D7. **e), f)** Phylogenetic trees of **e)** pfSN01and **f)** pfKE01, analysed together with KIR2DL1-binding RIFINs of 3D7 (blue). Yellow indicates the KIR2DL1-binding RIFINs candidates in each case and the dashed box indicates the clade of KIR2DL1-binding RIFINs of 3D7.

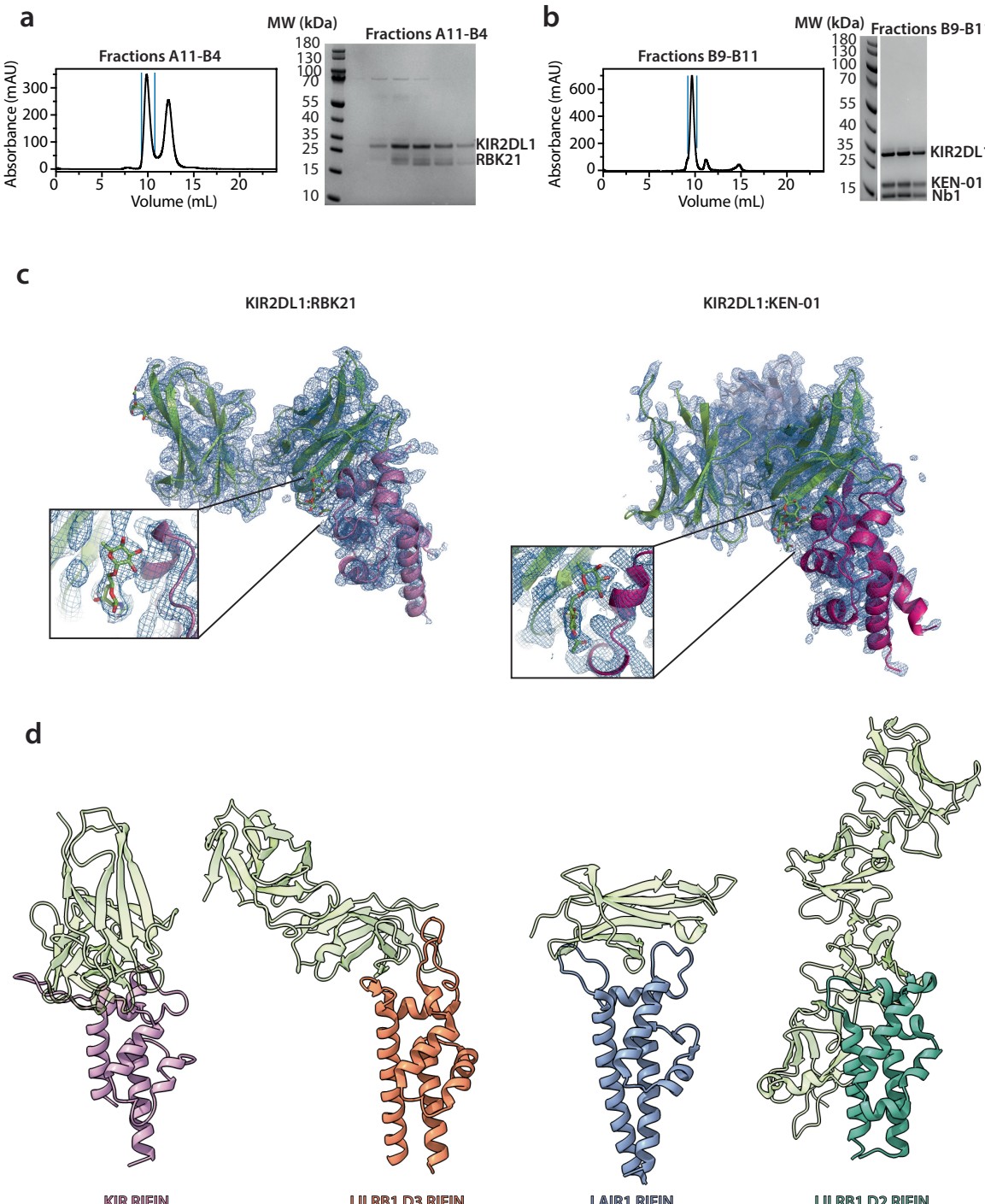

**Extended Data Fig. 3 | Structural analysis of binding of RIFINs to KIR2DL1.**
**a)** Purification of a complex of KIR2DL1 bound to RIFIN RBK21 with the left-hand panel showing the size exclusion chromatography plot on a Superdex 75 column, while the right-hand panel shows a Coomassie-stained gel, showing representative of n = 3 independent purifications. **b)** Purification of a complex of KIR2DL1 bound to RIFIN KEN-01, showing representative of n = 3 independent purifications. **c)** Structures of KIR2DL1 bound to RBK21 (left) and KEN-01

(right), showing electron density at contour level σ = 1.0 with grid spacing determined by PyMOL. The insets focus on a region of the electron density containing an ordered N-linked glycan attached to residue N67 of KIR2DL1. **d)** Comparison of the structure of RBK21 (KIR RIFIN, pink), bound to KIR2DL1 with RIFINs bound to LILRB1 (orange, PDB code 7KFK), LAIR1 (blue, PDB code 7JZK) or LILRB1 (green, PDB code 6ZDX). In each case, the immune receptor is shown in green.

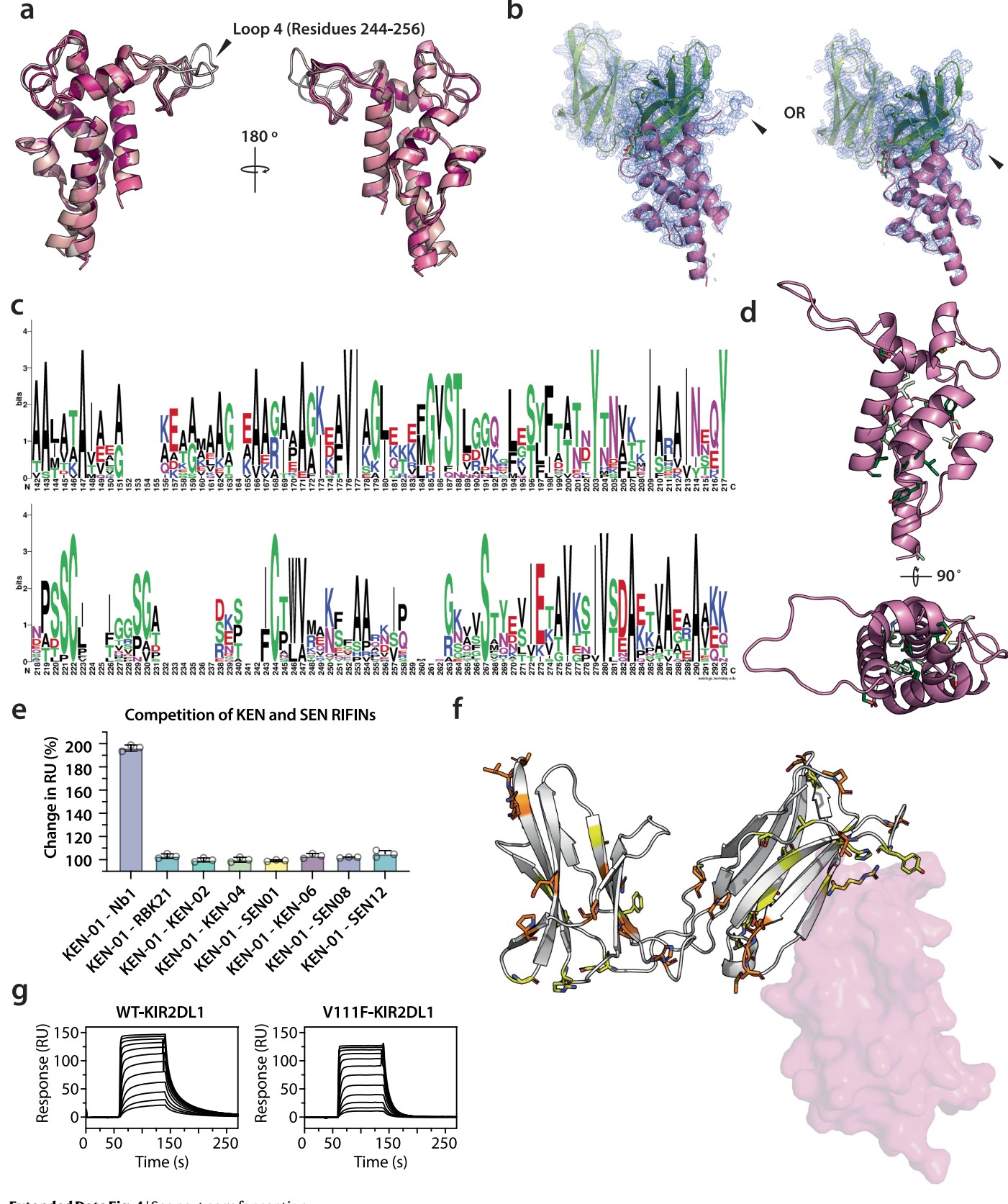

**a** Loop 4 (Residues 244-256)

180°

**b** OR

**c**

bits

bits

**d** 90°

**e** Competition of KEN and SEN RIFINs

Change in RU (%)

KEN-01 - Nb1
KEN-01 - RBK21
KEN-01 - KEN-02
KEN-01 - KEN-04
KEN-01 - SEN01
KEN-01 - KEN-06
KEN-01 - SEN08
KEN-01 - SEN12

**f**

**g**

WT-KIR2DL1

Response (RU)

Time (s)

V111F-KIR2DL1

Response (RU)

Time (s)

**Extended Data Fig. 4** | See next page for caption.

**Extended Data Fig. 4 | Analysis of KIR2DL1-binding RIFIN diversity and KIR2DL1 allotypes. a)** Overlay of each copy of RBK21 in the unit cell of the crystal lattice, highlighting the loop which adopts different conformations (residues 244–256) and a change in angle of the helix from 166–176 of 11°. **b)** Electron density for representatives of the two main conformers of RBK21 with the loop 256 in either an upward or downward conformation (black arrow). **c)** Sequence Logo of the clade of 3D7 KIR2DL1-binding RIFINs from, generated using WebLOGO[60] with numbering from 142 of RBK21. **d)** Shannon-entropy of sequence conservation was calculated using the Protein Variability Server (PVS[57]). Residues with entropy <0.8 are shown as sticks and coloured with dark green for 0, mid-green for 0–0.5 and light green for 0.5–0.8. **e)** Competition of RIFINs was measured by SPR analysis. 100% represents the binding of KEN-01 at 10 μM to a KIK2DL1 coated surface. This was repeated in the presence of a non-inhibitory nanobody (Nb1) or by other RIFINs indicated. Error bars show ± SD from n = 3 technical replicates. **f)** Plotting all polymorphisms found in the D1 and D2 domains of KIR2DL1, as found in the IPD-KIR database. Orange sticks denote the residue is mutated in more than one of the 177 recorded allotypes, yellow sticks denote a polymorphism found in only one recorded allotype. **g)** The effect of the V111F allotype of KIR2DL1 on RBK21 binding was measured by SPR analysis, in both cases with 500 RU on the chip and a concentration series of 12 doubling dilutions starting at 50 μM.

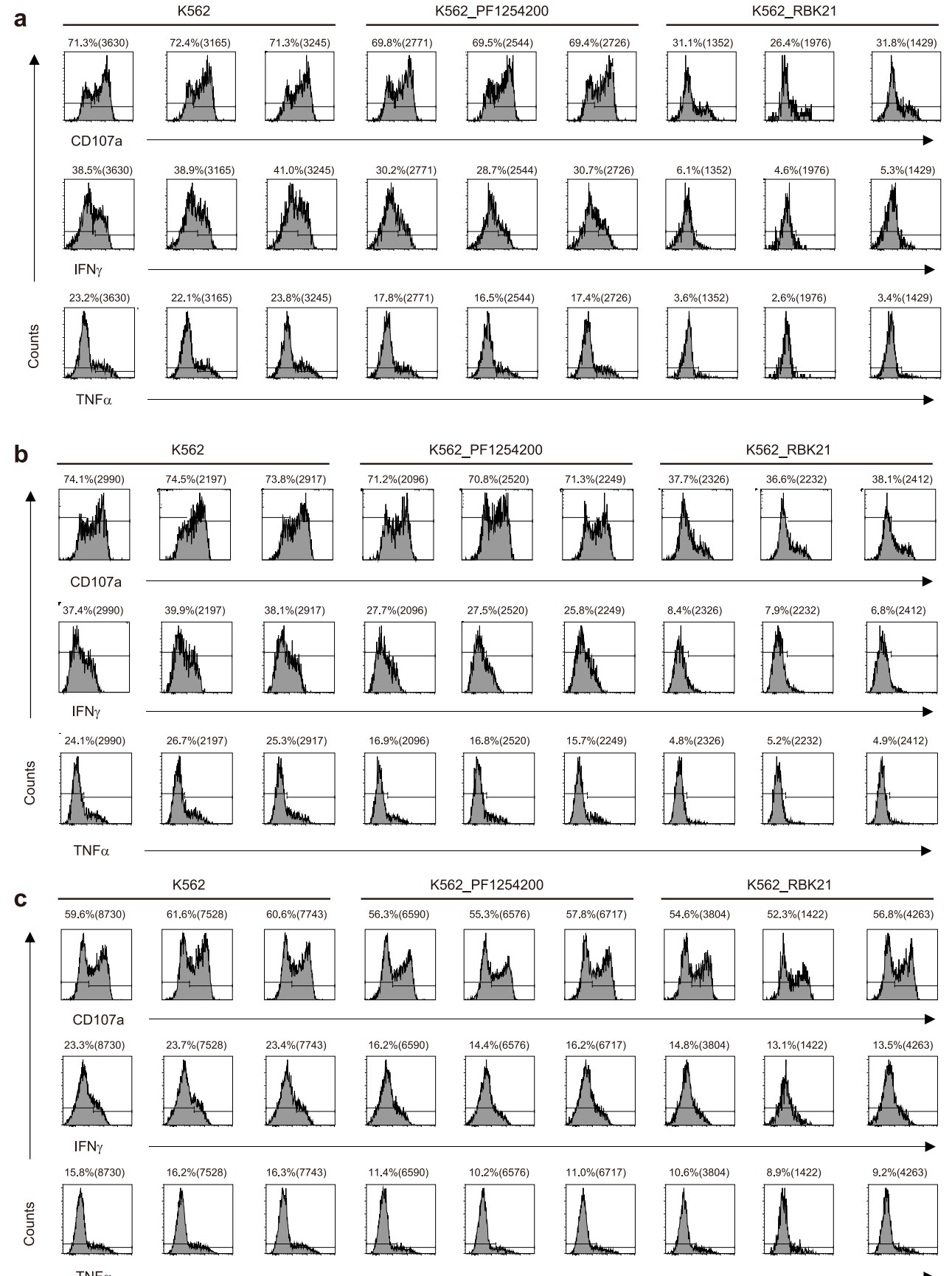

**Extended Data Fig. 5 | Flow sorting data.** Flow-sorting data associated with **a)** Fig. 3c, d, e **b)** Extended Data Fig. 6a and **c)** Extended Data Fig. 6b.

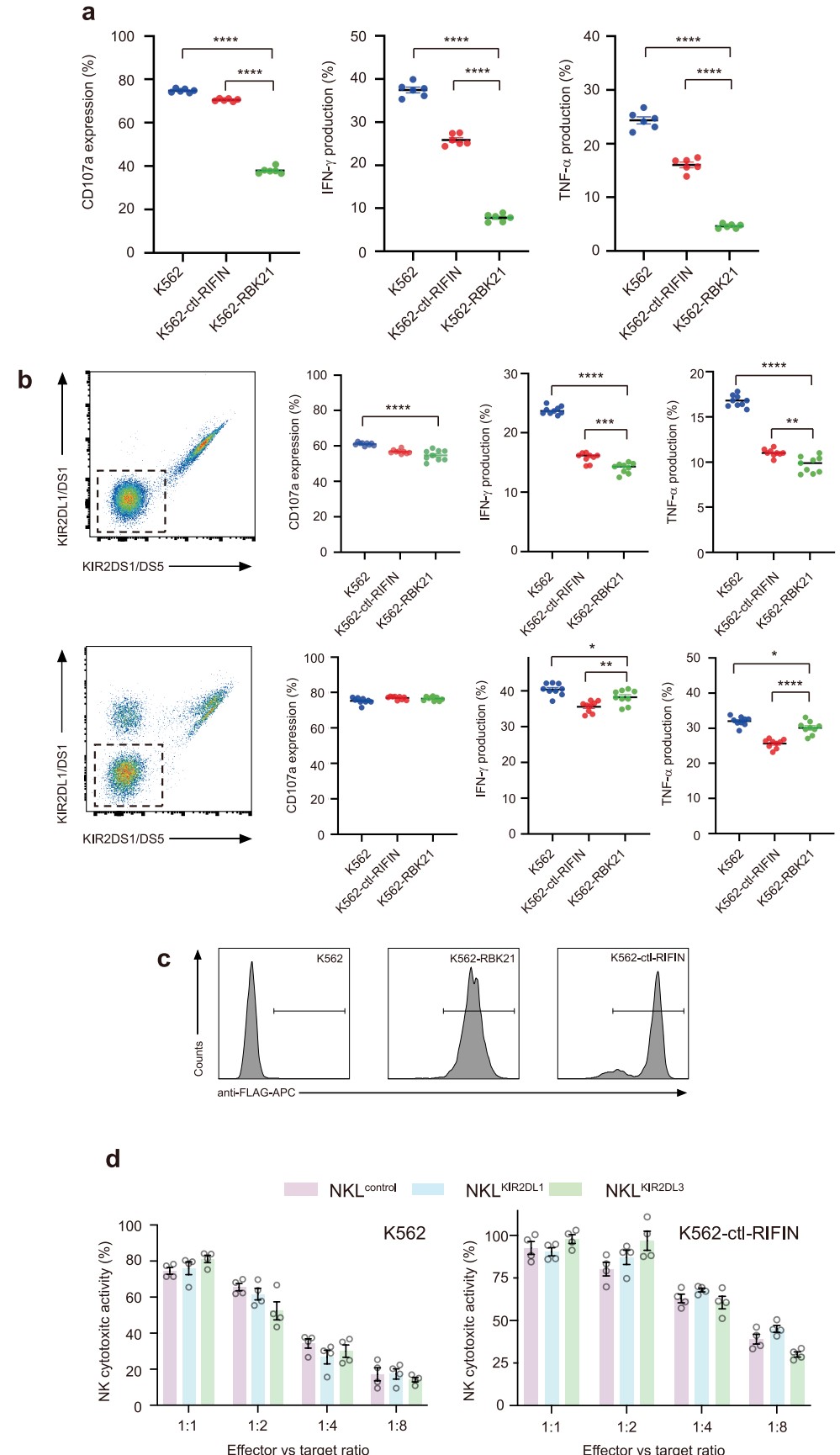

**Extended Data Fig. 6** | See next page for caption.

**Extended Data Fig. 6 | Inhibition of KIR2DL1 activity by RIFINs. a)** NK expressing KIR2DL1 were obtained from a donor different from the one used in Fig. 3. Suppression of CD107a expression (left) and production of IFN-γ (center) and TNF-α (right) in these KIR2DL1-(+)-NK cells by RBK21, expressed on the surface of K562, is shown. K562 or K562 cells expressing ctl-RIFIN (PF3D7_1254200) were used as negative controls. Data represent the mean (n = 6 independent measurements from one donor), ****P < 0.0001 (two-sided Student's t-test). **b)** The effects of RBK21 expressed on K562 on CD107a expression and production of IFN-γ production, and TNF-α in the KIR2DL1-(+) (top-left) and KIR2DS1-(+)- (bottom-left) donor-derived KIR2DL1/DS1-(−) NK cells are shown. Data represent the mean (n = 9 independent measurements from one donor). ****P < 0.0001, ***P < 0.001, **P < 0.01, *P < 0.05 (two-sided Student's t-test). **c)** RBK21 and PF3D71254200 were expressed as FLAG-tagged protein in K562 and their cell surface expression was confirmed using anti-FLAG antibody. **d)** Cytotoxic activities of NKL, NKL-KIR2DL1, and NKL-KIR2DL3 to K562 (left) and K562- expressing control RIFINs (PF3D7_1254200) (right) are shown. Data represent the mean with ± SD (n = 4 biological independent samples). The statistical analyses using two-sided Student's t-test show that there are no significant differences between each sample. Exact P values for all panels in.

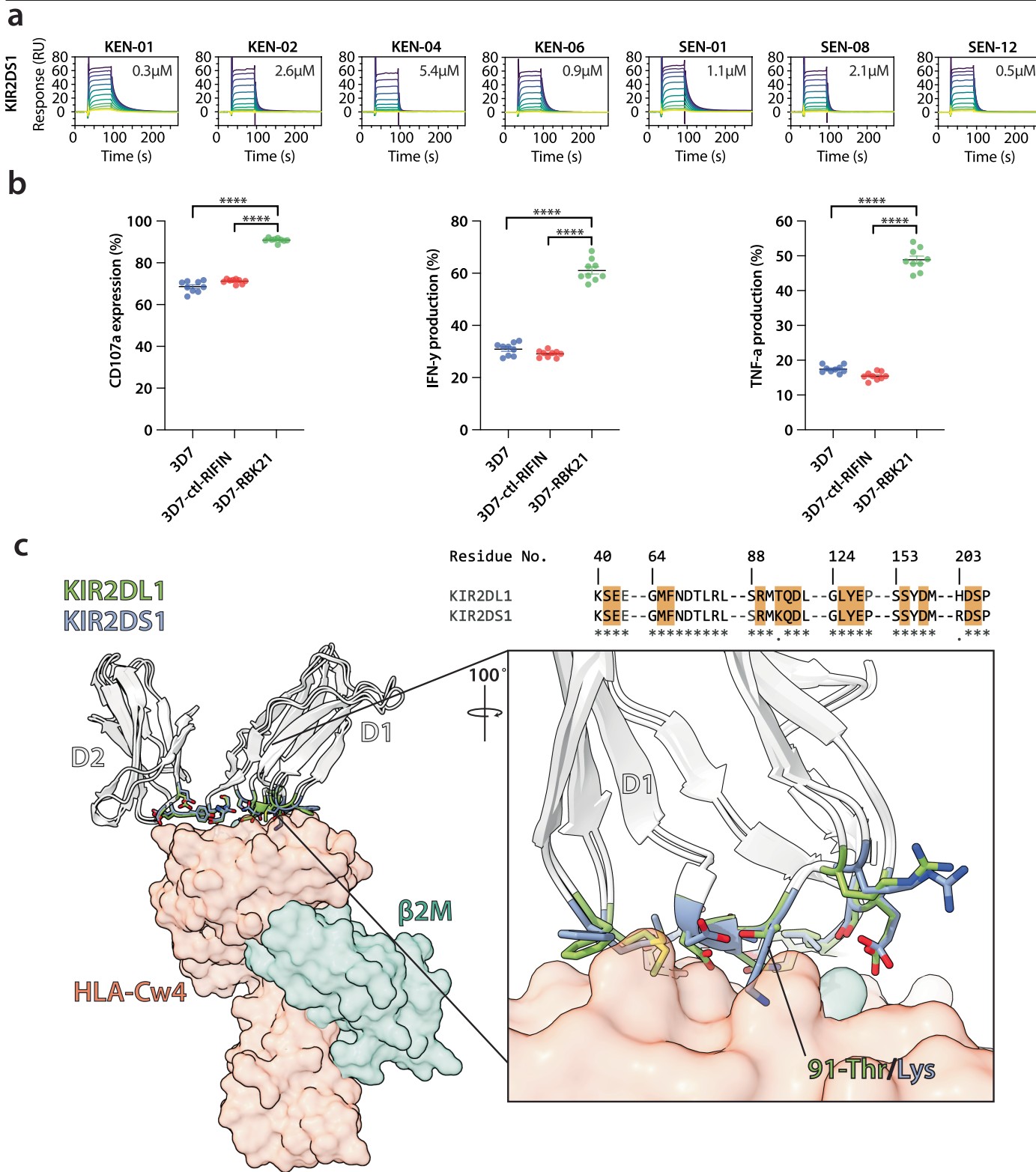

**Extended Data Fig. 7 | Binding of KIR2DL1-binding RIFINs to KIR2DS1 and modulation of KIR2DS1 activity. a)** SPR analysis of the binding of different KIR2DL1-binding RIFINs to immobilised KIR2DS1. In each case, a concentration series was conducted from 40 μM (KEN-02, -04, SEN-01, -08). or 20 μM (KEN-01, -06, SEN-12). Affinities calculated by equilibrium fitting from at least 3 independent experiments (PRISM10) are denoted within each box. **b)** Effect of RBK21 on CD107a expression and production of IFN-γ and TNF-α production in the gated KIR2DS1-(+) subset were assessed for a different donor to that tested in Fig. 4. The 3D7 strain and iRBC expressing PF1254200 (ctl-RIFIN) were used as negative controls. Data represent the mean (n = 9 independent measurements from one donor) with **** P < 0.0001 (two-sided Student's t-test). Exact P values in source data. **c)** The structure of KIR2DL1 bound to HLA-Cw4 (PDB 1IM9) was overlaid with a homology model (SWISS-MODEL) of KIR2DS1. Interfacial residues are shown as sticks for KIR2DL1 (green) and KIR2DS2 (blue). The 91Thr/Lys mutation at this interface is labelled. Also shown is a multiple sequence alignment of the regions responsible for the interaction, with residues involved in interactions highlighted in orange (above inset).

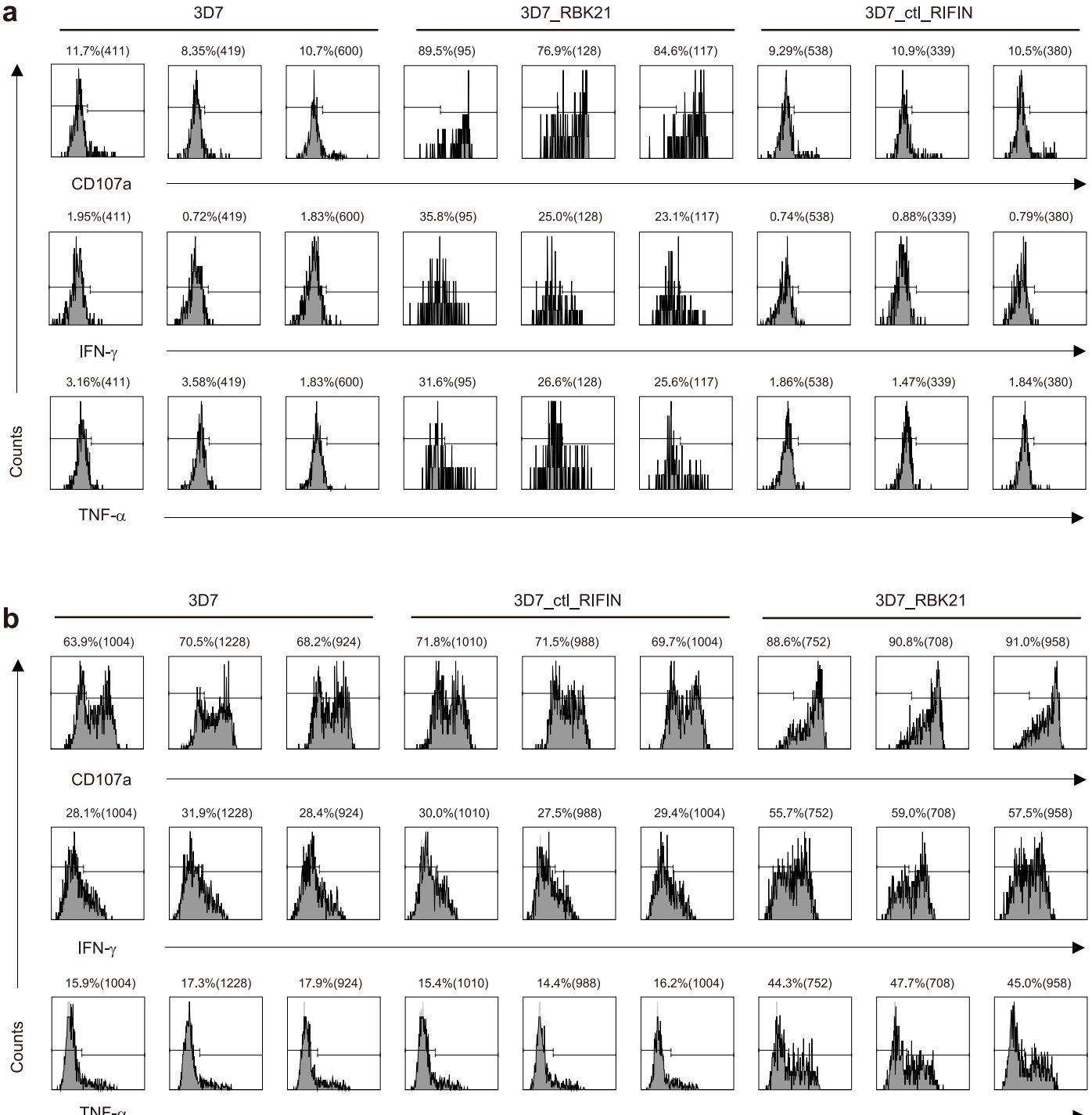

**Extended Data Fig. 8 | Flow sorting data.** Flow-sorting data associated with **a)** Fig. 4g, h, i and **b)** Extended Data Fig. 7b.

**Extended Data Table 1 | The list of KIR2DL1-binding RIFINs identified from rif-lib**

| gene_id | normalization_rif_lib_1-1_before | normalization_rif_lib_1-1_K2DL1 | fold_enrichment | normalization_rif_lib_1-2_before | normalization_rif_lib1_2_K2DL1 | fold_enrichment |
|---|---|---|---|---|---|---|
| PF3D7_10006 | 0 | 1 09E-05 | #DIV/0! | 4 741E-06 | 2 003E-05 | 4 225299 |
| PF3D7_10404 | 0 0024948 | 0 1233378 | 49 438532 | 0 0189662 | 0 4576397 | 24 129174 |
| PF3D7_06318 | 0 0084006 | 0 2615098 | 31 130051 | 0 0013086 | 0 0338963 | 25 90292 |
| PF3D7_04212 | 0 0049497 | 0 1021276 | 20 633056 | 0 000614 | 0 0151451 | 24 66661 |
| PF3D7_06323 | 0 0002241 | 0 0033679 | 15 029908 | 0 0002679 | 0 0087345 | 32 605847 |
| PF3D7_02227 | 0 0005876 | 0 0073898 | 12 576449 | 0 0002951 | 0 0010217 | 3 4616907 |
| PF3D7_13006 | 0 0027089 | 0 0304639 | 11 245875 | 0 0034516 | 0 0323737 | 9 3792351 |
| PF3D7_09005 | 0 0001544 | 0 0016567 | 10 732286 | 0 0003023 | 0 0014224 | 4 7058232 |
| PF3D7_10411 | 0 0049746 | 0 0503662 | 10 124669 | 0 0023327 | 0 0251017 | 10 760772 |
| PF3D7_01153 | 0 0014042 | 0 0094825 | 6 7527473 | 0 0012991 | 0 0040467 | 3 1150014 |
| PF3D7_02232 | 0 0136739 | 0 0721324 | 5 2751715 | 0 0137094 | 0 0588377 | 4 2917874 |
| PF3D7_11012 | 0 0004183 | 0 0021799 | 5 2114799 | 0 000627 | 0 0015225 | 2 4281491 |
| PF3D7_02231 | 3 486E-05 | 0 0001308 | 3 7522655 | 3 674E-05 | 2 003E-05 | 0 5451999 |
| PF3D7_04009 | 8 465E-05 | 0 0002943 | 3 4763636 | 5 215E-05 | 0 0003806 | 7 2982437 |
| PF3D7_10410 | 0 0002988 | 0 0009918 | 3 3197127 | 0 0001268 | 0 001182 | 9 3193511 |
| PF3D7_12003 | 0 0157903 | 0 0497232 | 3 1489763 | 0 0480788 | 0 0224372 | 0 4666767 |
| PF3D7_04133 | 0 0005976 | 0 001733 | 2 9001886 | 0 0005749 | 0 0031052 | 5 4014131 |
| PF3D7_14003 | 0 0110945 | 0 0321751 | 2 9000903 | 0 0174064 | 0 047599 | 2 7345755 |
| PF3D7_12550 | 0 002923 | 0 0060383 | 2 0657703 | 0 003313 | 0 0061702 | 1 8624574 |
| PF3D7_03245 | 0 0060751 | 0 0090574 | 1 4909104 | 0 0048302 | 0 0164673 | 3 4092719 |
| PF3D7_14008 | 0 0002739 | 0 0003379 | 1 2336994 | 0 0002145 | 0 0010417 | 4 8555922 |
| PF3D7_09377 | 9 959E-05 | 6 54E-05 | 0 6566465 | 5 215E-05 | 0 0002003 | 3 8411809 |
| PF3D7_12542 | 0 0005527 | 0 0001962 | 0 354944 | 0 0002252 | 0 000581 | 2 5796562 |
| PF3D7_06327 | 0 0006324 | 0 0001199 | 0 189583 | 0 0002252 | 0 0004808 | 2 1348879 |
| PF3D7_07323 | 0 0043024 | 0 0006104 | 0 1418681 | 0 0053316 | 0 0219765 | 4 121968 |
| PF3D7_06320 | 4 98E-06 | 0 | 0 | 5 927E-06 | 2 003E-05 | 3 3802392 |
| PF3D7_07329 | 4 98E-06 | 0 | 0 | 5 927E-06 | 2 003E-05 | 3 3802392 |

KIR2DL1-binding RIFINs which are enriched by more than 2 are shown.

**Extended Data Table 2 | The list of KIR2DL1-binding RIFIN candidates identified from Lek174-rif-lib and Lek79-rif-lib**

| Geneid | contig_ID | number of mapped reads of rifin from Lek174-rif-lib1 | number of mapped reads of rifin from Lek174-rif-lib2 |
|---|---|---|---|
| 174-rif25 | contig_12_pilon_pilon_pilon | 1515 | 1126 |
| 174-rif82 | contig_65_pilon_pilon_pilon | 450 | 3165 |
| 174-rif29 | contig_14_pilon_pilon_pilon | 268 | 39915 |
| 174-rif100* | contig_9_pilon_pilon_pilon | 192 | 215 |
| 174-rif58 | contig_60_pilon_pilon_pilon | 173 | 161 |
| 174-rif90 | contig_74_pilon_pilon_pilon | 150 | 946 |
| 174-rif42 | contig_5_pilon_pilon_pilon | 144 | 1200 |
| 174-rif80 | contig_65_pilon_pilon_pilon | 133 | 83 |
| 174-rif69 | contig_62_pilon_pilon_pilon | 96 | 100 |
| 174-rif88 | contig_74_pilon_pilon_pilon | 80 | 59 |
| 174-rif30 | contig_14_pilon_pilon_pilon | 78 | 250 |
| 174-rif50 | contig_54_pilon_pilon_pilon | 76 | 1994 |
| 174-rif36* | contig_229_pilon_pilon_pilon | 69 | 252 |
| 174-rif6 | contig_1_pilon_pilon_pilon | 66 | 126 |
| 174-rif47 | contig_51_pilon_pilon_pilon | 63 | 132 |
| 174-rif55 | contig_60_pilon_pilon_pilon | 54 | 45 |
| 174-rif43 | contig_5_pilon_pilon_pilon | 51 | 90 |
| 174-rif53 | contig_54_pilon_pilon_pilon | 40 | 23 |
| 174-rif91 | contig_74_pilon_pilon_pilon | 35 | 107 |
| 174-rif1* | contig_1_pilon_pilon_pilon | 34 | 92 |
| 174-rif7 | contig_10_pilon_pilon_pilon | 33 | 93 |
| 174-rif20* | contig_106_pilon_pilon_pilon | 27 | 92 |
| 174-rif61 | contig_60_pilon_pilon_pilon | 27 | 91 |
| 174-rif21 | contig_11_pilon_pilon_pilon | 26 | 168 |

| Geneid | Chr | number of mapped reads of rifin from Lek79-rif-lib1 | number of mapped reads of rifin from Lek79-rif-lib2 |
|---|---|---|---|
| 79-rif89 | contig_5_pilon_pilon_pilon | 101202 | 62993 |
| 79-rif26 | contig_22_pilon_pilon_pilon | 17420 | 14266 |
| 79-rif35 | contig_25_pilon_pilon_pilon | 7581 | 2108 |
| 79-rif63 | contig_33_pilon_pilon_pilon | 1112 | 30 |
| 79-rif7 | contig_19_pilon_pilon_pilon | 1006 | 951 |
| 79-rif23 | contig_22_pilon_pilon_pilon | 568 | 1539 |
| 79-rif70 | contig_3_pilon_pilon_pilon | 409 | 415 |
| 79-rif17 | contig_20_pilon_pilon_pilon | 288 | 700 |
| 79-rif33 | contig_23_pilon_pilon_pilon | 222 | 17 |
| 79-rif19 | contig_20_pilon_pilon_pilon | 213 | 298 |
| 79-rif3 | contig_19_pilon_pilon_pilon | 211 | 361 |
| 79-rif10 | contig_19_pilon_pilon_pilon | 174 | 173 |
| 79-rif27 | contig_22_pilon_pilon_pilon | 136 | 89 |
| 79-rif60 | contig_33_pilon_pilon_pilon | 119 | 114 |
| 79-rif84 | contig_56_pilon_pilon_pilon | 91 | 64 |
| 79-rif44 | contig_25_pilon_pilon_pilon | 68 | 131 |
| 79-rif80 | contig_55_pilon_pilon_pilon | 66 | 68 |
| 79-rif88 | contig_56_pilon_pilon_pilon | 58 | 87 |
| 79-rif48 | contig_25_pilon_pilon_pilon | 55 | 32 |
| 79-rif45 | contig_25_pilon_pilon_pilon | 54 | 49 |
| 79-rif71* | contig_3_pilon_pilon_pilon | 53 | 88 |
| 79-rif72 | contig_4_pilon_pilon_pilon | 36 | 39 |

The green and blue show the top ranked candidates identified from the two biologically independent libraries.

* Cannot be used for phylogenetic analysis due to incomplete variable region.

**Extended Data Table 3 | Crystallographic data collection and refinement statistics**

| | KIR2DL1:RBK21 | KIR2DL1:KEN-01 |
|---|---|---|
| **Data Collection** | | |
| Space group | P 21 21 21 | P 31 2 1 |
| Cell dimensions | | |
| a, b, c (Å) | 93.40, 99.05, 321.13 | 91.17, 91.17, 109.20 |
| $\alpha, \beta, \gamma$ (°) | 90, 90, 90 | 90, 90, 120 |
| Wavelength (Å) | 0.97625 | 0.97625 |
| Resolution (Å) | 160.57 – 2.89 (3.15– 2.89)* | 84.15 – 2.17 (2.21 – 2.17)* |
| Total Observations | 651743 (33873) | 651518 (26957) |
| Total Unique | 53588 (2679) | 32044 (1596) |
| $R_{merge}$ (%) | 30.2 (192.2) | 12.5 (349.4) |
| $R_{meas}$ (%) | 32.9 (209.0) | 12.8 (360.2) |
| $R_{pim}$ (%) | 13.0 (81.7) | 2.8 (87.1) |
| $CC_{1/2}$ | 0.994 (0.643) | 0.999 (0.404) |
| $I/\sigma(I)$ | 7 (1.5) | 15.3 (0.4) |
| Completeness (%) | 94.70 (58.0) | 100 (100) |
| Multiplicity | 6.4 (6.5) | 20.3 (16.9) |
| Wilson B factor | 96.98 | 64.63 |
| | | |
| **Refinement** | | |
| Reflections | 53578 | 31974 |
| Rwork / Rfree (%) | 25.6/28.6 | 22.2/23.0 |
| Average B factor | | |
| *Protein (all)* | 96.98 | 72.73 |
| *Protein (Chain A)* | 66.92 | 79.25 |
| *Protein (Chain B)* | 57.4 | 76.61 |
| *Protein (Chain C)* | 150.28 | 61.98 |
| *Protein (Chain D)* | 71.18 | |
| *Water* | - | 64.32 |
| *Ligands* | 104.31 | 71.57 |
| Number of residues | | |
| *Protein* | 2283 | 404 |
| *Water* | - | 159 |
| *Ligands* | 25 | 3 |
| RMSDs | | |
| *Bond lengths (Å)* | 0.008 | 0.0117 |
| *Bond angles (°)* | 0.85 | 1.55 |
| Ramachandran plot | | |
| *Favored (%)* | 97.0 | 98.3 |
| *Allowed (%)* | 3.0 | 2.7 |
| *Outliers (%)* | 0.0 | 0.0 |

* Values in parentheses are for highest resolution shell.

**Extended Data Table 4 | Table of contacts between RBK21 or KEN-01 and KIR2DL1**

| Chain | Residue (RBK21) | Residue (KEN-01) | Group | Chain | Residue (KIR2DL1) | Group | Interaction |
|---|---|---|---|---|---|---|---|
| A | - | R212 | SC | B | D93 | SC | Salt bridge/H-bond |
|   |   |   |   |   | F66 |   | Cation-Pi |
| A | - | N213 | SC | B | N67 | MC | H-bond |
| A | - | N219 | SC | B | N67 | G | H-bond |
| A | S221 | S221 | SC | B | T69 | SC | H-bond |
| C | S221 | - | SC | D | N67 | MC | H-bond |
| A | S222 | - | SC | B | N67 | G | H-bond |
| A | - | S222 | MC | B | N67 | G | H-bond |
| A | L224 | L224 | SC | B | L59 | SC | Hydrophobic |
|   |   |   |   |   | H61 |   |   |
|   |   |   |   |   | T69 |   |   |
|   |   |   |   |   | Y101 |   |   |
|   |   |   |   |   | V111 |   |   |
| A | - | L224 | MC | B | Y109 | SC | H-bond |
| A | T225 | - | SC | B | N67 | G | H-bond |
| A | M239 | - | SC | B | Y109 | SC | Hydrophobic |
| A | F242 | F242 | SC | B | L59 | SC | Hydrophobic |
|   |   |   |   |   | T69 |   |   |
|   |   |   |   |   | R71 |   |   |
|   |   |   | MC | B | R71 | SC | H-bond |
| C | F242 | - | MC | D | H57 | SC | H-bond |
| A | - | F242 | SC | B | R71 | SC | H-bond |
| A | F243 | F243 | SC | B | L59 | SC | Hydrophobic |
|   |   |   |   |   | T69 |   |   |
| A | A245 | A245 | MC | B | R71 | SC | H-bond |
| A | - | E255 | MC | B | I73 | MC | H-bond (β-sheet) |
| A | - | A256 | MC | B | R71 | MC | H-bond (β-sheet) |
| A | V257 | V257 | MC | B | R71 | MC | H-bond (β-sheet) |
| A | S258 | S258 | MC | B | T69 | MC | H-bond (β-sheet) |
|   |   |   | SC |   | D68 | SC | H-bond |
| A | M259 | M259 | MC | B | T69 | MC | H-bond (β-sheet) |
| A | - | N260 | SC | B | N67 | MC | H-bond |

SC = sidechain, MC = mainchain, G = glycan.

# Reporting Summary

## Statistics

For all statistical analyses, confirm that the following items are present in the figure legend, table legend, main text, or Methods section.

| n/a | Confirmed | |
|---|---|---|
| ☐ | ☒ | The exact sample size (*n*) for each experimental group/condition, given as a discrete number and unit of measurement |
| ☐ | ☒ | A statement on whether measurements were taken from distinct samples or whether the same sample was measured repeatedly |
| ☐ | ☒ | The statistical test(s) used AND whether they are one- or two-sided *Only common tests should be described solely by name; describe more complex techniques in the Methods section.* |
| ☒ | ☐ | A description of all covariates tested |
| ☒ | ☐ | A description of any assumptions or corrections, such as tests of normality and adjustment for multiple comparisons |
| ☐ | ☒ | A full description of the statistical parameters including central tendency (e.g. means) or other basic estimates (e.g. regression coefficient) AND variation (e.g. standard deviation) or associated estimates of uncertainty (e.g. confidence intervals) |
| ☐ | ☒ | For null hypothesis testing, the test statistic (e.g. *F*, *t*, *r*) with confidence intervals, effect sizes, degrees of freedom and *P* value noted *Give P values as exact values whenever suitable.* |
| ☒ | ☐ | For Bayesian analysis, information on the choice of priors and Markov chain Monte Carlo settings |
| ☒ | ☐ | For hierarchical and complex designs, identification of the appropriate level for tests and full reporting of outcomes |
| ☒ | ☐ | Estimates of effect sizes (e.g. Cohen's *d*, Pearson's *r*), indicating how they were calculated |

*Our web collection on statistics for biologists contains articles on many of the points above.*

## Software and code

Policy information about availability of computer code

| | |
|---|---|
| Data collection | Crystallography data was collected at ESRF ID30-A and Diamond Light Source i03. SPR data were collected using T200 Biacore Software version 2.0 (GE Healthcare). CD data were collected using Spectra Manager Version 2 (Jasco). TIRF microscopy images were acquired using an Olympus cell TIRF-4Line system with a 150x (NA 1.45) oil objective. NGS sequence data of RIFIN expression library were collected using MiSeq (Illumina). Flow cytometry data were collected using Attune NxT (ThermoFisher Scientific). Cell-sorting was performed using SH800 (SONY) or BD LSR II using BD FACSDiva software. In each case, the software is the standard in the field and is available for use by other researchers. |
| Data analysis | Data analysis was performed as described in the methods section using commercially available or openly accessible software. Software used for crystal data processing are standard and freely available to academic users. Model building and refinement was performed with COOT version ccp4-0.8.9.2 and autoBUSTER v2.10 (Global Phasing Ltd). The SWISS-MODEL webserver was used for homology structure prediction. Chimera v1.16 and ChimeraX v1.5 were used for structure visualisation. GraphPad Prism version 10 was used to generate graphs and for statistical tests. T200 Biacore Evaluation software v1.0 is provided with the Biacore SPR machine and is standard in the field. Microscopy images were analysed using ImageJ (v.1.54b, NIH). Fastq data obtained by screening RIFIN expression library were analyzed using bowtie2 v2.3.4 and featureCounts 2.0.1. |

For manuscripts utilizing custom algorithms or software that are central to the research but not yet described in published literature, software must be made available to editors and reviewers. We strongly encourage code deposition in a community repository (e.g. GitHub). See the Nature Portfolio guidelines for submitting code & software for further information.

# Data

Policy information about availability of data

All manuscripts must include a data availability statement. This statement should provide the following information, where applicable:
- Accession codes, unique identifiers, or web links for publicly available datasets
- A description of any restrictions on data availability
- For clinical datasets or third party data, please ensure that the statement adheres to our policy

Data within graphs (source data) and uncropped gel and blot images are included with this submission. Crystallographic data is deposited in the protein data bank with accession codes 9F2D and 9HML. Sequence data related to rif-lib1 and -lib2 were deposited at NCBI Gene expression omnibus with accession number GSE286478. All materials are available from the authors.

# Human research participants

Policy information about studies involving human research participants and Sex and Gender in Research.

| | |
|---|---|
| Reporting on sex and gender | N/A |
| Population characteristics | N/A |
| Recruitment | N/A |
| Ethics oversight | N/A |

Note that full information on the approval of the study protocol must also be provided in the manuscript.

# Field-specific reporting

Please select the one below that is the best fit for your research. If you are not sure, read the appropriate sections before making your selection.

☒ Life sciences  ☐ Behavioural & social sciences  ☐ Ecological, evolutionary & environmental sciences

For a reference copy of the document with all sections, see nature.com/documents/nr-reporting-summary-flat.pdf

# Life sciences study design

All studies must disclose on these points even when the disclosure is negative.

| | |
|---|---|
| Sample size | Sample sizes are described in figure legends and methods. No statistical method was used to predetermine sample size. Instead, experiments were conducted based on experience of similar studies and statistical significance was assessed on the collected data. Quantitative experiments were typically repeated in technical triplicate. Sample sizes for each experiment were chosen to be consistent with the field norms. |
| Data exclusions | For CD measurements, data range between 180 and 190 nm were excluded due to high HT values indicating poor quality data in this low UV range. |
| Replication | The number of repeats for each relevant experiment are given in figure legends and the methods. Typically, experiments were performed in independent technical triplicates. RIFIN-library screening was carried out in biological duplicate. |
| Randomization | No experiments were randomized and there were no covariants to control. The only assay in which subjectivity is a possible confounder, is in selection of which cells to study to extract the data shown in Figure 3d-f. To avoid this, we acquired fields of cells across the sample based on signals in the IRM channel. All cells within these fields were included in the analysis. This avoided 'cherry-picking' of images, and provided an unbiased assessment. None of the other experiments carried a risk of subjective decisions about data inclusion. |
| Blinding | The investigators were not blinded to the group allocation during the experiment and/or when assessing the outcome, as analysis were performed on quantitative endpoints that are not subject to investigator bias. |

# Reporting for specific materials, systems and methods

We require information from authors about some types of materials, experimental systems and methods used in many studies. Here, indicate whether each material, system or method listed is relevant to your study. If you are not sure if a list item applies to your research, read the appropriate section before selecting a response.

## Materials & experimental systems

| n/a | Involved in the study |
|---|---|
| ☐ | ☒ Antibodies |
| ☐ | ☒ Eukaryotic cell lines |
| ☒ | ☐ Palaeontology and archaeology |
| ☒ | ☐ Animals and other organisms |
| ☒ | ☐ Clinical data |
| ☒ | ☐ Dual use research of concern |

## Methods

| n/a | Involved in the study |
|---|---|
| ☒ | ☐ ChIP-seq |
| ☐ | ☒ Flow cytometry |
| ☒ | ☐ MRI-based neuroimaging |

## Antibodies

| | |
|---|---|
| Antibodies used | anti-KIR2DL1/DS1 (Milteny Biotec, 130-118-973, 1:100 dilution)<br>anti-KIR2DL1/DS5 (R&D system, MAB1844-SP, 1:100 dilution)<br>anti-FLAG antibody (Sigma-Aldrich, F1804, 1:200 dilution)<br>APC-conjugated anti-human IgG Fc antibody (Jackson ImmunoResearch,109-136-098, 1:100 dilution)<br>FITC-conjugated CD56 (Biolegend, 318303, 1:100 dilution),<br>PacificBlue-conjugated anti-human CD107a (Biolegend, 328623, 1:100 dilution)<br>PerCP/Cy5.5-conjugated anti-human IFN-g (Biolegend, 506527, 1:100 dilution),<br>APC/Cy7-conjugated anti-human TNF-a (Biolegend, 502943, 1:100 dilution),<br>APC-conjugated anti-mouse-CD45 (Biolegend, 103111, 1:100 dilution)<br>anti-KIR3DL2 antibody (Biolegend, 389602, 1:100 dilution)<br>anti-KIR3DL2 (Biolegend, 389602, 1:100 dilution)<br>anti-KIR3DL3 (R&D Systems, FAB8919P, 1:100 dilution).<br>FITC-conjugated anti-KIR2DL1/DL5 (R&D Systems, FAB1844F, 1:100 dilution)<br>PE-conjugate anti-KIR2DL2/DL3/DS2 (Biolegend, 312605, 1:100 dilution),<br>PE-conjugated anti-KIR2DL5 (Miltenyi Biotec, 130-096-199, 1:100 dilution),<br>FITC-conjugated anti-KIR3DL1 (Biolegend, 312705, 1:100 dilution) |
| Validation | The antibodies used in this study were obtained from commercial vendors. We selected commercially available antibodies based on the validation provided by the manufacturers for their use in flow cytometry. Detailed validation information can be accessed on the manufacturers' websites using the details provided in the "Antibodies Used" section above. |

## Eukaryotic cell lines

Policy information about cell lines and Sex and Gender in Research

| | |
|---|---|
| Cell line source(s) | Commercial FreestyleTM 293 and Expi 293F GNTI-TM cells were purchased from Thermo Fisher.<br>HEK293T was obtained from RIKEN cell Bank.<br>NKL were generously gifted by L.L. Lanier at the University of California San Francisco.<br>The human erythroleukemia cell line, K562, was obtained from the Cell Resource Centre for Biomedical Research, Institute of Development, Ageing and Cancer, Tohoku University.<br>Plasmodium falciparum 3D7 (ID: MRA845) was obtained from BEI Resources. |
| Authentication | FreestyleTM 293 and Expi 293F GNTI-TM cells were authenticated by Thermo Fisher. Example of citation; PMID:14701821.<br>HEK293T cell was authenticated by RIKEN cell Bank. Example of citation: PMID: 39772386.<br>NKL was described in Exp. Hematol. 1996 Feb;24(3):406-15. PMID: 8599969<br>K562, was authenticated by the Cell Resource Centre for Biomedical Research, Institute of Development, Ageing and Cancer, Tohoku University. Example of citation: PMID: 29186116<br>Plasmodium falciparum 3D7 strain was authenticated by BEI Resources. |
| Mycoplasma contamination | Each cell line listed above was regularly tested by mycoplasma contamination by PCR. |
| Commonly misidentified lines<br>(See ICLAC register) | Plasmodium falciparum 3D7 is not listed in ICLAC. |

## Flow Cytometry

### Plots

Confirm that:

☒ The axis labels state the marker and fluorochrome used (e.g. CD4-FITC).

☒ The axis scales are clearly visible. Include numbers along axes only for bottom left plot of group (a 'group' is an analysis of identical markers).

☒ All plots are contour plots with outliers or pseudocolor plots.

☒ A numerical value for number of cells or percentage (with statistics) is provided.

## Methodology

| | |
|---|---|
| Sample preparation | The iRBCs at schizont stage were obtained by 70% - 40% percoll density gradient centrifugation. Human peripheral blood mono-nucleated cells (PBMC) of healthy donors were isolated from fresh blood samples by Ficoll-Paque (Leucosep. |
| Instrument | Sample were analyzed using Attune NxT (ThermoFisher Scientific). The iRBCs , on which expressed KIR2DL1-binding RIFINs, were sorted using SH800 (SONY). KIR-Fc validation assay was analysed using BD LSR II. |
| Software | Data collection: Attune NxT software (for Attune NxT), SH800s software ( for SH800), BD FACSDiva ( for BD LSR II)<br>Data analysisi: FlowJo v10.10.0 |
| Cell population abundance | The sorted iRBC, which was infected with field-isoalated parasites, were > 32.7 % KIR2DL-1 positive (FIgure 1a). The iRBCs expressing KIR2DL1-binding RIFINs were sorted from RIFIN expression libraries and were 2.13- 1.63 % were positive in the iRBC population of libraries (Figure 1d). |
| Gating strategy | All gating strategy are provided in Extended data Figure 12 |

☒ Tick this box to confirm that a figure exemplifying the gating strategy is provided in the Supplementary Information.

