## [Peer Review File · Nature]

RIFINs displayed on malaria-infected erythrocytes bind KIR2DL1 and KIR2DS1

Corresponding Author: Professor Matthew Higgins

Version 0:

Reviewer comments:

Referee #2

(Remarks to the Author)

In "RIFINs displayed on malaria-infected erythrocytes bind both KIR2DL1 and KIR2DS1", Sakoguchi et al. investigate recognition by RIFINs of inhibitory killer immunoglobulin-like receptors (KIRs), assessing seven inhibitor KIRs for recognition by four *P. falciparum* field isolates – finding strain Lek174 to recognize KIR2DL1. They create a transgenic parasite line, expressing a chimera of the KIR-binding RIFIN, and further clone a KIR2DL1-binding RIFIN, which they name RBK21. From a RIFIN-expression library from the 3D7 strain, they identify 16 additional RIFINs that bound KIR2DL1 in two independent screens. 10 of these (including RBK21) fell into the same phylogenetic clade, which included 7 other RIFINs, 5 of which also bound KIR2DL1. They next determine the crystal structure of KIR2DL1 bound to RBK21 variable regions (residues 148-299) at 2.9 Å resolution, analyze and compare the interface with other RIFIN complexes, and identify and confirm S221K as a knockout mutation.

The authors further show RBK21 to inhibit NK function through KIR2DL1-mediated signaling, with RBK21 engagement of KIR2DL1 resulting in inhibitory signaling that suppresses NK cell activation. The authors characterize affinity of RBK21 versus HLA Cw4, finding weak affinity for RBK21 (0.45 μM to KIR2DL1 and 2 μM to KIR2DS1), in both cases about 10-fold tighter than to HLA Cw4. Notably, the authors also show RBK21 to bind the activating KIR, KIR2DS1, and to increase cytotoxic function of KIR2DS1-expressing NK cells.

Overall, the authors add a fascinating chapter to the still developing story of RIFIN interactions, defining interaction and biological impact with both activating and inhibitory KIRs on NK cells. I like the multiple controls used to assess function, and figures and writing are clear.

One issue I have with the paper is that the authors suggest this research highlights the evolutionary battle between pathogen and host. Certainly, there is a battle between pathogen and host. But have activating KIRs evolved to allow detection of red blood cells infected with *P. falciparum* as suggested by the authors? Or is it merely that KIRs exist as pairs, to permit fine tuning of their inhibitory/activating function?

In the third paragraph of the Discussion, the authors argue that the affinity difference with HLA from a threonine to a lysine specifically reduces MHC affinity for KIR2DS1. But I find this data unconvincing in that the relative affinity of MHC is ~10-fold lower than RBK21 to both activating and inhibiting KIRs. That is, the ratios of affinities between HLA and RBK21 for KIR2DL1 (6 μM/0.45 μM = 13.3) and for KIR2DS1 (23 μM/2 μM) = 11.5) are very similar, so the following statements appear to be contradictory: (i) (line 215) "...in the case of KIR2DL1/S1, with the RIFIN-binding interface remaining almost perfectly conserved between the pair."; and (ii) (line 217) "KIR2DS1 contains a threonine to lysine mutation that reduces binding to the MHC class 1 molecule, HLA-Cw4." That is, it doesn't make sense to say that a mutation specifically affects HLA binding, when the affinity ratios to RIFIN are essentially the same (13.3 versus 11.5).

The authors also state that "low KIR2DS1 expression is common in regions with historically low malaria transmission, where it will not be required for malaria prevention (ref. 29)." (lines 224-225). However, examination of ref. 29 indicates that the prevalence data is more complicated, as Tororo with the highest frequency of malaria (28.7%) had a KIR2DS1 a prevalence of 26.7%, intermediate between the that of Kanungu (KIRSD1 prevalence of 19.7%) and of Jinja (KIR2DS1 prevalence of 30.4%), both of which had lower malaria frequencies than Tororo of 7.4% and 9.3%, respectively. Thus, the prevalence of KIR2DS1 was lowest in Kanungu, which has a malaria incidence of only a third that of Tororo, and ref. 29 does not provide

evidence that the prevalence of KIR2DS1 segregates by the frequency of malaria.

Overall, the authors repeatedly highlight the evolutionary battle/race between malaria and human host, but the evolutionary component of "evolutionary battle" and "evolutionary arms race" is not well supported. The affinity data is unclear, and the data from ref. 29 are cited incorrectly. For these reasons, the evolutionary aspect of the battle/race should be removed. "Battle" and "arms race" by themselves are sufficient and clear.

A second issue I have with the paper relates to the prevalence of malaria with RIFINs that bind KIRs. The authors show that only 1 of 4 tested malaria field strains bound KIR2DL1. It would be helpful to show higher prevalence of field strains that recognize KIR2DL1/KIR2DS1, to show that the study relates to an important aspect of malaria biology, versus just a side observation in a minority of field isolates. However, the authors do not determine the prevalence of RIFINs such as RBK21 in malaria field strains, an unfortunate omission. The authors should use the genetic and structural information obtained in their study to create a signature that allows for an estimate of the prevalence of KIR2DL1/KIR2DS1-binding RIFINs based on sequenced malaria field strains.

Other points the authors should address:

1. The authors should clarify whether all 157 3D7 RIFINs were present in the combined rif-lib1 and -lib2, or were any missing in both libraries.
2. In light of only 1 in 4 field strains recognizing KIR2DL1, it is surprising that the 3D7 strain had so many RIFINs that recognized KIR2DL1. The authors should comment/discuss why 3D7 has so many of these types of RIFINs.
3. SPR data with μM affinity is difficult to measure, and the off-rate for HLA measurements appears very high. I like that the authors use equilibrium measurements. However, no error estimates are provided for affinities provided. The authors should provide explicit experimental errors for the affinity measurements shown in Fig. 4c, and also a bit about the concentration series that was used. What are the dilutions in the Fig. 4c experiments? The top curves shown with RBK21 appears closer together than those that are lower down, suggesting experimental problems if a constant dilution between measurements was used.

Referee #4

(Remarks to the Author)

Here the authors present structural and functional characterisation of a *P. falciparum* RIFIN RBK21 binding to human KIR2DL1 and KIR2DS1. The study identifies a novel interacting partner for KIR2DL1 and is the first identified pathogen derived ligand for the activating KIR2DS1. Furthermore, a novel binding site on KIR2DL1/S1 away from the HLA-I interaction surface is revealed, with conservation of the surface between inhibitory and activating receptors. Signal transduction and inhibition/activation of NK cells is shown following engagement with RBK21 indicating a possible impact on NK cell function in vivo. These findings are likely to have broad interest across the fields of infectious disease and immunology, however, there are some concerns over some aspects of data quality and scope of the conclusions drawn given the data presented.

Major Comments

1. Abstract, lines 12-16: "We find that KIR2DL1-binding RIFINs can also bind to KIR2DS1 and that these RIFINs cause activation of KIR2DS1 expressing NK cells. This highlights the evolutionary battle between pathogen and host, suggesting that activating KIRs may have evolved to allow detection of red blood cells infected with *Plasmodium falciparum*, helping the host to clear the parasite."

Please could the authors re-word this section for improved accuracy. While a group of 15 KIR2DL1 binding RIFINs are identified here it seems only RBK21 undergoes structural characterisation and functional evaluation of KIR2DS1 binding in this manuscript.

It has been demonstrated that LILRB1 binding RIFINs bind to at least two distinct binding sites on LILRB1 and that the two LAIR1 binding RIFINs bind overlapping epitopes, but bind in different orientations (manuscript refs 2 and 3). Could it therefore be premature to suggest that this group of KIR2DL1 binding RIFINs possess similar characteristics to RBK21 and would be able to modulate NK signalling and bind KIR2DS1 in a similar manner leading to any significant evolutionary pressure?

Extending this study to evaluate the remaining KIR2DL1 binding RIFINs would be extremely interesting, particularly with regard to their binding site on KIR2DL1 and their ability to impact both inhibitory and activating NK cell function.

2. Please could the authors address some concerns over the structural statistics and model quality presented in extended data table 7 and in the PDB validation report as follows:

a. The completeness of the high resolution shell only reaches 58%. Despite the remaining indicators of data quality falling within acceptable ranges, this is extremely low. Ideally a resolution cutoff should be selected where this reached at least 90-95%.

- b. Please could the authors indicate how their multiplicity values were calculated as there seems to be a mismatch between these values and the reported number of unique and total reflections.
- c. The RSRZ score for the model indicates a large number of residues where there is a poor fit between the model and data. If possible, could the authors attempt to improve this value?

Minor Comments

1. The structural basis for the interaction between RIFINs and KIR2DL1: Line 94. It appears in the Chen et al. Nature 2021 paper (man. ref 5) that the LILRB1 binding RIFINs identified there bind a distinct epitope on D3 of LILRB1 in contrast to the D1D2 binding observed in Harrison et. al. Nature 2020 (man. ref 3). Could the authors please modify this line to reflect the multiple binding sites of RIFINs on LILRB1.
2. The structural basis for the interaction between RIFINs and KIR2DL1: Line 108. In extended data figure 6 it looks like the angle of the longest alpha helix also varies between copies. Could the authors please include a comment on this in the text if this is the case.
3. The structural basis for the interaction between RIFINs and KIR2DL1: Line 110-115. It is somewhat difficult to see the interface being discussed in the text in the current figure 2b. Perhaps the figure could be split into 2-3 panels for clearer visualisation of the hydrophobic pocket, the polar interactions and the stabilising beta sheet.
4. The structural basis for the interaction between RIFINs and KIR2DL1: Line 124-128. Could the authors please additionally comment on sequence diversity in the LAIR1 and LILRB1 binding RIFINs here.
5. Inhibitory signalling mediated by KIR2DL1-binding RIFINs reduces cytotoxic attack of iRBCs by NK cells: lines 143-153. Could the authors please state whether all PBMCs in these assays are KIR2DL1 positive, or just those co-cultured with the K562-RBK21 expressing cells. It is currently worded slightly ambiguously in the results and accompanying methods section and could affect the interpretation of these results.
6. Figure 3.
 - a. Could the authors please improve labelling of the figure axes here with fewer abbreviations and improved descriptors (for example % is used as a sole axis label in fig 3C). The y axis in figure 3F is cutoff.
 - b. There are no text call-outs for figures 3E and 3F, could the authors please include these alongside the text in lines 162-167.
7. Figure 4.
 - a. In Figure 4A it would be helpful to have the mismatching residues between KIR2DL1 and KIR2DS1 highlighted.
 - b. Figure 4D. There seem to be no data on the number of replicates or the mean and SD values for the MFI for this assay.
8. Fig 3b and Extended data fig 9. Fig 4F and Extended data fig 11. Extended data figure 12.
 - a. Could the authors please comment on the lack of any biological replicates for the assays in fig 3b and 4F? It would be extremely useful to show that these data were statistically significant and there were real differences in cytokine signalling.
 - b. In extended data figure 11 it looks like the cell counts for the 3D7-RBK21 data may be considerably lower than for the other two conditions. Please could the authors include cell counts for these datasets. Was a gating strategy selected to ensure consistent cell counts across these assays?
 - c. Unfortunately, I am unable to determine the appropriateness of the gating strategies shown in extended data figure 12 as none of the text or axes are legible. Please could the authors correct this.
 - d. In extended data figure 12G the legend refers to a gating strategy for extended data figure 10. I believe this is an error and should refer to extended data figure 11. Please could the authors correct this if appropriate.

Referee #5

(Remarks to the Author)

Several recent studies (including five published in Nature) have shown that *P. falciparum* RIFINs bind to human inhibitory receptors such as LAIR1 and LILRB1. In this new submission, the Authors extend these observations by identifying a class of RIFINs that bind to another inhibitory receptor, KIR2DL1, and to its homologous activatory receptor, KIR2DS1. However, the binding of RIFINs to KIR was already anticipated in Saito et al. Nature 2017 (Extended Figure 1).

For a limited set of the KIR2D-binding RIFINs, the Authors provide structural and functional data. The structural data show a mode of binding consistent with previous reports on RIFIN binding to the Ig domains of LAIR1 and LILRB1. The functional characterization, performed using a sophisticated system already reported in previous studies, aligns with expectations based on the properties of the engaged receptors. However, the *in vivo* relevance of binding to inhibitory or activatory receptors remains speculative.

In conclusion, this study is generally well executed, and the results are overall convincing. However, the findings are primarily incremental, offering no novel biological insights and addressing a highly specialized audience.

Specific point: why did the Authors test binding to LILRB1 in Figure 2c and what do they conclude from this experiment?

Referee #6

(Remarks to the Author)

The authors appear to have uncovered a really interesting and potentially important set of receptor/ligand interactions that if fully substantiated have implications both for mechanisms of NK cell recognition of malaria infection and the generation of early innate immune responses as well as the selection and diversification of KIR genes across evolution. The novelty is significant as immune evasion strategies involving direct binding of pathogen-encoded molecules to inhibitory KIR have not been described. Moreover, our understanding of the biological significance of activating KIR has remained very limited since their discovery. The observation that a pathogen with sufficient prevalence and disease severity expresses proteins that can be directly recognised by activating KIR provides perhaps a plausible driver for the relatively recent evolution of such KIR, albeit that it is juxtaposed with other observations that might suggest that the primary function of activating KIR is related to regulating the development of spiral arteries in the placenta. However establishing clinical significance may be a step too far at this point in time since it would require detailed genetic analyses of both host-encoded KIR (to identify malaria-infected individuals with KIR2DL1 but not KIR2DS1 as well as those who possess KIR2DS1), detailed analyses of the RIFIN repertoires of infected individuals all linked to clinical data. Regardless, showing that malaria-encoded proteins directly engage KIR2DL1 or KIR2DS1 to impact infection is significant.

Unfortunately, in its current format, I do not think the manuscript sufficiently substantiates the primary observation or indeed use the data at hand to show the observation in its best light. In short, while the generation of transgenic parasites seems almost heroic, the work substantiating KIR binding and functional analyses to address the broader impact of these interactions on NK cell biology is underdone and frequently either lacks appropriate controls or in the case of the structural data, not really highlighting the relevant interactions with sufficient clarity.

I have outlined concerns in more detail below.

The statement "Activating KIR are proposed to have evolved from inhibitory KIRs through mutations in the ITIM domain." (line 36)...is loose. More correctly aKIR were created by 1) truncation of the tail and 2) addition of a charged residue in their transmembrane region to facilitate pairing with ITAM-containing adaptor proteins such as DAP12 and (3) mutations in the extracellular domain that appear designed to attenuate recognition of HLA-encoded ligands.

The initial screening data is interesting but lacking important controls. Negative data from KIR-Ig fusion proteins is not sufficiently robust in the absence of some demonstration of their capacity to bind defined ligands. Supplementary data showing that these reagents bind the expected HLA allotypes and or CD155 in the case of KIR2DL5 should be included. Also included should be their binding to uninfected RBC.

The binding of KIR2DL3 is 2.5% which is not that different from the KIR2DL1 value, yet the allelically related KIR2DL2 has almost no binding. In the absence of data which provides confidence about the fidelity of the reagents, little can be concluded about the selectivity of the RIFIN/KIR2DL1 interaction for KIR2DL1 over other KIR.

The use of library strategies to identify additional KIR2DL1-binding RIFINS is powerful and elegant. The mapping of KIR2DL1-binders to a single clade is nice. The Logo analyses of KIR-binding RIFINS is informative. However it would have also been helpful to identify features that were shared/or not between these the 2 clade members that didn't bind KIR2DL1. The statement that the conservation of KIR2DL1-binding RIFINS suggests there is a major selective advantage to targeting the KIR2DL1 inhibitory receptor seems overstated at this point in time. It is not clear to me how inhibitory KIR modulate NK cell recognition of infected RBC. More context is needed here.

The structural data should be informative but at present is not presented very clearly and perhaps is not used to its full extent. Does it provide insights into the selectivity of these RIFINS for KIR2DL1/S1 as opposed to KIR2DL2/3 or indeed other 2 domain KIR. Some more formal comparison and tests of the key features of KIR and the extent to which they differ across different KIR and even between KIR2DL1 allotypes should be shown. The mutagenesis experiments at the receptor/ligand interface seem pretty minimal and should be extended to test their hypotheses around the key drivers of the RIFIN/KIR interaction and by extension insights into its binding into specific clades or clade members.

The data showing that transfection of RBK21 into K562 protects seem relatively clear. That said, there are additional controls that should be included - responses of KIR2L1-ve NK cells (especially those expressing other similar 2 domain KIR such as KIR2DL2/3). Some form of blocking experiment would be also typical here albeit that the mAbs used for blocking HLA interactions likely will not suffice in this case due to differences in binding sites. It is unclear to me the value of 9 technical replicates or additional clarity on what is meant by technical replicates is required. I would prefer to see data analysing NK cell responses from multiple donors where the number of donors is clearly indicated. It is also not clear the extent to which the inhibition of activation of primary NK cells is in part dependent on the function of other RIFIN-binding inhibitory receptors.

The authors also need to show the level of cell surface expression of both KIR2DL1-binding and non-binding RIFIN's on their K562 cells.

The extent of inhibition observed with NKL cells expressing KIR2L1 with respect to cytotoxicity is not particularly convincing and again additional controls that include NKL cells that do not express KIR2DL1 and/or cells that express other KIR would minimally be required. Additionally, target cells expressing a "non-interacting" RIFIN or the S221R mutant should be included.

Fig 4 is largely focussed on demonstrating that RBK21 also binds to KIR2DS1 and that this can stimulate NK cell activation. Again the data is strongly suggestive of these things but lacks key controls. The SPR experiments should include both non-binding RIFIN as well as control KIR (eg KIR2DL2). The KIR2S1-Ig binding data should also include a non-binding KIR. The functional analyses showing activation following recognition of infected RBC again needs additional analyses on subsets of NK cells that do not express either KIR2DL1 or -2DS1. Additionally, the authors need to be clearer about how many donors have been used in these analyses- what exactly is 9 technically independent samples?).

The authors conclude that RIFINs are the evolutionary drivers for the emergence of activating KIR. This seems premature since there is strong genetic evidence of a role for such receptors in impacting reproductive outcomes independent of infection. Secondly, if this is the evolutionary driver, what then of other activating KIR? Are these also RIFIN specific but the strains assessed simply did not have ligands within their RIFIN-repertoire.

The underlying purpose of having an inhibitory receptor that is specific for a malaria -encoded Ag expressed on the surface of red blood cells is not clear. The data suggest that NK cells can kill RIFIN-expressing rbc as a result of KIR2DS1 engagement. Does KIR2DL1 engagement diminish ADCC responses targeting infected rbc?

Version 1:

Reviewer comments:

Referee #2

(Remarks to the Author)

Sakoguchi and colleagues have responded robustly to the referee comments. I appreciate that the authors in several cases have accepted referee arguments and removed the more speculative discussions/conclusions that were inconsistent with data. I was also impressed with the additional analysis the authors undertook to show the general significance of the RIFIN-KIR interaction, such as in the subsection entitled: "KIR2DL1-binding RIFINs are common in field isolates from both Africa and South-East Asia". The additional structure of KEN-01 bound to KIR2DL1 as well as the additional sequence analysis on the conservation of the recognition motif aid in demonstrating generality.

Referee #4

(Remarks to the Author)

The authors have submitted a substantially revised manuscript and have gone to considerable effort to address the reviewers comments with both improvements to the figures and text as well as a considerable amount of new data.

I am satisfied that the majority of my comments have been addressed in the current version. While it would be exciting to see a slightly broader scope explored with regard to RIFIN interactions with other KIR family members, I agree that this work represents a considerable and novel addition to the field. The additional structural studies on a second KIR2DL2 binding RIFIN KEN-01 help to validate the original RBK21 dataset, which is unfortunately somewhat limited in resolution and the quality of density for the RIFIN as the authors have stated in the rebuttal. It is particularly pleasing to see the inclusion of data showing the presence of KIR2DL2 binding RIFINs in multiple field isolates which helps to build a case for this interaction being of importance for host-parasite biology. Revisions to the manuscript text and figures have helped to improve clarity and reduce overstating the results presented.

Minor Comments

1. Line 141: To develop a non-binding mutant for future studies we produced S221R. This abolished binding to KIR2DL1, as shown through SPR analysis..

As this section is now a comparative discussion of the binding of RBK21 and KEN-01 to KIR2DL1 it should be clearly stated that this mutant. was produced to eliminate binding for RBK21 and abolished only this interaction.

2. Extended data figure 3C.

The mesh used for the KIR2DL1-KEN01 complex in this figure is double that of the adjacent KIR2DL1-RBK21 figure and those in the rest of the manuscript. Could the authors please make this uniform for easier comparison of the structures.

3. Extended Data figure 5.

I believe there is a typo in the figure legend as it reads "KIR2DBL1".

4. Extended data table 3.

Wavelength is given to different decimal places for RBK21 versus KEN-01 structures and units are included only for the RBK21 structure rather than for both or as part of the row heading.

Referee #6

(Remarks to the Author)

The manuscript by Sakoguchi et al provides clear evidence of an interaction between RIFINs and KIR. While there is a significant number of RIFINs in any single parasite and considerable strain to strain variation, the capacity to bind KIR is present in isolates from geographically distinct areas and at a frequency that suggests significance. The authors have used crystallography to illustrate the molecular basis of the interaction which highlights the key elements of both the KIR and RIFIN.

While there are numerous pathogen-encoded immunoevasins that target NK cell responses, this is the first identified example of such a protein directly binding a KIR, with RIFINs selectively binding KIR2DL1 and -2DS1. The reasons for such selective targeting remain unclear. Critically, direct recognition of pathogen-encoded proteins by activating NK cell receptors has been reported previously, this is again the first example of an activating KIR binding such a protein despite such mechanisms being proposed as an evolutionary driver for the generation of such KIR well over a decade earlier.

In my opinion the central observation is well supported by the data which clearly shows an interaction via direct binding assays, structural techniques as well as functional assays and the significance of the observation worthy of publication in Nature. While recognising the magnitude of the changes in functional responses as significant, I am still surprised that the authors have still only analysed NK cells from 2 donors and then used either reporter cells or NKL cells to further substantiate the functional relevance.

Version 2:

Reviewer comments:

Referee #6

(Remarks to the Author)

I appreciate the efforts of the authors to add additional donors to the KIR2DS1 data set. As indicated previously, I believe the data and the observations using primary NK cells is well supported by the other analyses.

Again, as indicated I think the discovery is significant and I like the work and support publication.

Your manuscript, "RIFINs displayed on malaria-infected erythrocytes bind both KIR2DL1 and KIR2DS1", has now been seen by 4 referees, whose comments are attached below. While they find your work of potential interest, as do we, they have raised important concerns that in our view need to be addressed before we can consider publication in Nature.

Should further experiments allow you to address these criticisms, we would be happy to consider a revised manuscript (unless something similar has been accepted at Nature or appeared elsewhere in the meantime). We hope to receive your revised paper within four to six months. If you cannot complete the required revisions within this time frame, please let us know when you would anticipate being able to submit a revised manuscript. We also strongly suggest that your revised manuscript has tracked changes, which is increasingly requested by referees to aid in their re-review.

We thank the reviewers for their comments and have prepared a revised manuscript with substantial new data in response. We have not used track changes as the changes were very substantial. However, we have indicated line numbers in the comments to reviewers as well as highlighting key responses in red text in the manuscript.

I'll share a few thoughts about the Referee comments and what we see as the key priorities, hopefully this will be helpful should you decide to revise your manuscript. Referee #5 feels that the data are an insufficient advance over previous studies showing RIFINs binding to inhibitory host receptors and the RIFIN binding to KIRs could be readily anticipated from earlier publications. Please consider carefully how you can emphasize the new insights and make this aspect clear to the Referees.

We have responded in a detailed way to reviewer 5 below. We do not agree with their view that this is an incremental advance and have addressed directly their claim, with which we disagree, that an earlier publication showed that RIFINs bind to KIRs. No such claim has been made in the literature and their interpretation of a panel in an Extended Data Figure in a previous paper would not, in our view, stand up to peer review. We have outlined in the response to reviewer 5, as well as throughout the revised manuscript, the major advances presented here. We are delighted that reviewer 5 is an outlier and that other reviewers are excited about the novel discoveries presented here.

At least two of the Referees feel it's unclear whether KIR2DL1/S1-binding RIFINs represent central malaria biology, or an unimportant side example of RIFIN recognition. It will be essential to estimate the prevalence of such RIFINs in field isolates. This could help elevate the current finding from perhaps a limited curiosity to something suggesting importance for host-parasite biology.

We now include a substantial new data set in which we identify clades of KIR2DL1-binding RIFINs in four field-isolated strains. We use our library screening approach to find such RIFINs in two South-East Asian isolates and show that many of these cluster in the same clades as our existing 3D7 RIFINs. We then use phylogenetic analysis to predict KIR2DL1-binding RIFINs in two African isolates and confirm that these bind to KIR2DL1 and KIR2DS1 by surface plasmon resonance. We also conduct structural studies and show suppression of NK cell activation by one of these Kenyan RIFINs, KEN-01, showing that it functions in the same way as the South-East Asian RBK21 RIFIN previously studied in depth. We are therefore confident that ~10% of RIFINs across different geographical locations have the KIR2DL1/DS1 binding properties shown in this manuscript and that,

far from a limited curiosity, it is a conserved and retained feature of *Plasmodium falciparum* biology likely to have substantial relevance for immune interaction in malaria.

Many of the Referees feel that the manuscript overclaims in several aspects e.g. the "evolutionary arms race" idea is interesting but speculative- can it be solidified in some way?

We note and accept the reviewer comments that we cannot demonstrate that KIR2DS1 has evolved in response to the presence of KIR2DL1-binding RIFINs in *Plasmodium falciparum*. While this remains an interesting and promising theory, we are not able to demonstrate causation and have therefore changed the way in which we discuss this point in the manuscript. Instead, we highlight that this is the first time a ligand has been found for an activating immune receptor and highlight our data, which provides the first experimental demonstration that activating KIRs can stimulate NK cell function in response to binding of a pathogen-derived protein. It is our view that this is a major advance and we have highlighted this in the discussion.

Referee #2 (Remarks to the Author):

In "RIFINS displayed on malaria-infected erythrocytes bind both KIR2DL1 and KIR2DS1", Sakoguchi et al. investigate recognition by RIFINs of inhibitory killer immunoglobulin-like receptors (KIRs), assessing seven inhibitor KIRs for recognition by four *P. falciparum* field isolates – finding strain Lek174 to recognize KIR2DL1. They create a transgenic parasite line, expressing a chimera of the KIR-binding RIFIN, and further clone a KIR2DL1-binding RIFIN, which they name RBK21. From a RIFIN-expression library from the 3D7 strain, they identify 16 additional RIFINs that bound KIR2DL1 in two independent screens. 10 of these (including RBK21) fell into the same phylogenetic clade, which included 7 other RIFINS, 5 of which also bound KIR2DL1. They next determine the crystal structure of KIR2DL1 bound to RBK21 variable regions (residues 148-299) at 2.9 Å resolution, analyze and compare the interface with other RIFIN complexes, and identify and confirm S221K as a knockout mutation.

The authors further show RBK21 to inhibit NK function through KIR2DL1-mediated signaling, with RBK21 engagement of KIR2DL1 resulting in inhibitory signaling that suppresses NK cell activation. The authors characterize affinity of RBK21 versus HLA Cw4, finding weak affinity for RBK21 (0.45 μM to KIR2DL1 and 2 μM to KIR2DS1), in both cases about 10-fold tighter than to HLA Cw4. Notably, the authors also show RBK21 to bind the activating KIR, KIR2DS1, and to increase cytotoxic function of KIR2DS1-expressing NK cells.

Overall, the authors add a fascinating chapter to the still developing story of RIFIN interactions, defining interaction and biological impact with both activating and inhibitory KIRs on NK cells. I like the multiple controls used to assess function, and figures and writing are clear.

We thank the reviewer for their enthusiastic response to our study.

One issue I have with the paper is that the authors suggest this research highlights the evolutionary battle between pathogen and host. Certainly, there is a battle between pathogen and host. But have activating KIRs evolved to allow detection of red blood cells

infected with *P. falciparum* as suggested by the authors? Or is it merely that KIRs exist as pairs, to permit fine tuning of their inhibitory/activating function?

We thank the reviewer for this comment and, on reflection, agree that we overstated our view that activating RIFINs might have evolved specifically to bind RIFINs, as there are other potential functions for activating KIRs. We have therefore made changes throughout the manuscript to remove this argument. Nevertheless, our demonstration that NK cells which express KIR2DS1 are activated by these RIFINs indicates that individuals with RIFIN-binding activating KIRs will have a survival advantage in malaria-endemic areas. While this is consistent with these KIRs evolving as part of a strategy to limit malaria infection, it is not currently possible to demonstrate evolutionary causation. In the revised manuscript, we have therefore removed the claim that activating KIRs evolved from inhibitory KIRs in response to RIFINs and instead added a discussion of this potential survival advantage in malaria-endemic areas for individuals with activating KIRs that can promote parasite clearance. This includes changes in lines 16-17, 250-252 and 299-301.

In the third paragraph of the Discussion, the authors argue that the affinity difference with HLA from a threonine to a lysine specifically reduces MHC affinity for KIR2DS1. But I find this data unconvincing in that the relative affinity of MHC is ~10-fold lower than RBK21 to both activating and inhibiting KIRs. That is, the ratios of affinities between HLA and RBK21 for KIR2DL1 ($6 \mu\text{M}/0.45 \mu\text{M} = 13.3$) and for KIR2DS1 ($23 \mu\text{M}/2 \mu\text{M} = 11.5$) are very similar, so the following statements appear to be contradictory: (i) (line 215) "...in the case of KIR2DL1/S1, with the RIFIN-binding interface remaining almost perfectly conserved between the pair."; and (ii) (line 217) "KIR2DS1 contains a threonine to lysine mutation that reduces binding to the MHC class 1 molecule, HLA-Cw4." That is, it doesn't make sense to say that a mutation specifically affects HLA binding, when the affinity ratios to RIFIN are essentially the same (13.3 versus 11.5).

On reflection, we agree with the argument made by the reviewer and have removed this section from the discussion.

The authors also state that "low KIR2DS1 expression is common in regions with historically low malaria transmission, where it will not be required for malaria prevention (ref. 29)." (lines 224-225). However, examination of ref. 29 indicates that the prevalence data is more complicated, as Tororo with the highest frequency of malaria (28.7%) had a KIR2DS1 a prevalence of 26.7%, intermediate between the that of Kanungu (KIRSD1 prevalence of 19.7%) and of Jinja (KIR2DS1 prevalence of 30.4%), both of which had lower malaria frequencies than Tororo of 7.4% and 9.3%, respectively. Thus, the prevalence of KIR2DS1 was lowest in Kanungu, which has a malaria incidence of only a third that of Tororo, and ref. 29 does not provide evidence that the prevalence of KIR2DS1 segregates by the frequency of malaria.

We agree with the reviewer and have removed mention of this manuscript.

Overall, the authors repeatedly highlight the evolutionary battle/race between malaria and human host, but the evolutionary component of "evolutionary battle" and "evolutionary arms race" is not well supported. The affinity data is unclear, and the data from ref. 29 are cited incorrectly. For these reasons, the evolutionary aspect of the battle/race should be removed. "Battle" and "arms race" by themselves are sufficient and clear.

As noted above, on reflection, we agree with the reviewer. While our data does support the view that activating KIRs will provide an advantage to human populations, we cannot currently demonstrate causation. We have therefore removed the claim that activating KIRs originated from such an evolutionary battle.

A second issue I have with the paper relates to the prevalence of malaria with RIFINs that bind KIRs. The authors show that only 1 of 4 tested malaria field strains bound KIR2DL1. It would be helpful to show higher prevalence of field strains that recognize KIR2DL1/KIR2DS1, to show that the study relates to an important aspect of malaria biology, versus just a side observation in a minority of field isolates. However, the authors do not determine the prevalence of RIFINs such as RBK21 in malaria field strains, an unfortunate omission. The authors should use the genetic and structural information obtained in their study to create a signature that allows for an estimate of the prevalence of KIR2DL1/KIR2DS1-binding RIFINs based on sequenced malaria field strains.

First, in response to the reviewer's comment that only 1 of 4 field strains bind to KIR2DL1, our view is that this is likely to be due to the lack of expression of KIR2DL1-binding RIFINs in the tested field-isolated strains at the time of testing, not due to their absence in the genome. Each *P. falciparum* genome contains ~150 different RIFINs, most of which show suppressed expression due to heterochromatin, with only two or three RIFINs are expressed. At a different time, a different set of RIFINs will be expressed. A lack of KIR2DL1 binding for 3 of 4 strains tested could therefore be due to low KIR2DL1-binding RIFIN expression rather than lack of such RIFINs in the genome. Indeed, in new data added to the manuscript (lines 87-100 and Figure 2a) we generated RIFIN libraries from both the strain in which we detected KIR2DL1-binding RIFIN (Lek174) and one from which we did not (Lek79) and conducted screening using KIR2DL1-Fc. In both cases, we find that ~10% of RIFINs in these genomes bind KIR2DL1, demonstrating that KIR2DL1-binding RIFINs are present in the at least one of the genomes which didn't bind before selection.

Second, the reviewer makes the good suggestion that we propose a signature based on our structure to predict KIR2DL1-binding RIFINs. We have done this in the past (i.e. for ICAM-1-binding PfEMP1 in Lennartz et al, 2017) but the sequence variability in the KIR2DL1-binding RIFINs has precluded us from proposing a signature. Nevertheless, in new data, we used phylogenetic analysis to predict which RIFINs from two African field-isolated lines might bind to KIR2DL1, selected some of these RIFINs and demonstrated their binding to KIR2DL1 (lines 101-115 and Figure 2). One of these, derived from a Kenyan field-isolate (KEN-01) has then been characterised using structural studies (lines 117-144 and Figure 2d,e) and in our supported lipid bilayer assay (lines 202-215 and Figure 3g-i).

These new data combine to show that (i) KIR2DL1-binding RIFINs are prevalent across all tested field-isolated strains; (ii) that they can be, albeit imperfectly, predicted based on sequence and (iii) that they are similar in structure and ability to signal through KIR2DL1.

Other points the authors should address:

1. The authors should clarify whether all 157 3D7 RIFINs were present in the combined rif-lib1 and -lib2, or were any missing in both libraries.

This information is provided in lines 70-71, as well as Extended Data Tables 1 and 2. These show that rif-lib1 and lib2 cover 95.5% and 97.4% of all RIFINs of the 3D7 strain, respectively. Together they include 153 out of the 157 RIFINs in the genome.

2. In light of only 1 in 4 field strains recognizing KIR2DL1, it is surprising that the 3D7 strain had so many RIFINs that recognized KIR2DL1. The authors should comment/discuss why 3D7 has so many of these types of RIFINs.

As demonstrated for South-East Asian and African isolates in our new data, the field strains have similar numbers of KIR2DL1-binding RIFINs to 3D7 and we presume that they were not detected in our initial experiment due to lack of expression.

3. SPR data with μM affinity is difficult to measure, and the off-rate for HLA measurements appears very high. I like that the authors use equilibrium measurements. However, no error estimates are provided for affinities provided. The authors should provide explicit experimental errors for the affinity measurements shown in Fig. 4c, and also a bit about the concentration series that was used. What are the dilutions in the Fig. 4c experiments? The top curves shown with RBK21 appear closer together than those that are lower down, suggesting experimental problems if a constant dilution between measurements was used.

We have improved our analysis of these SPR data. Each measurement was taken in technical replicate and we have now normalised this data using the R_{MAX} and plotted average and standard deviation to measure affinity and estimate errors. These data are now shown in Figure 4c, which also makes the concentration dependence clearer.

Referee #4 (Remarks to the Author):

Here the authors present structural and functional characterisation of a *P. falciparum* RIFIN RBK21 binding to human KIR2DL1 and KIR2DS1. The study identifies a novel interacting partner for KIR2DL1 and is the first identified pathogen derived ligand for the activating KIR2DS1. Furthermore, a novel binding site on KIR2DL1/S1 away from the HLA-I interaction surface is revealed, with conservation of the surface between inhibitory and activating receptors. Signal transduction and inhibition/activation of NK cells is shown following engagement with RBK21 indicating a possible impact on NK cell function in vivo. These findings are likely to have broad interest across the fields of infectious disease and immunology, however, there are some concerns over some aspects of data quality and scope of the conclusions drawn given the data presented.

We thank the reviewer for their view that these findings are of broad interest. We are confident that we have addressed their specific comments through changes to the manuscript and the inclusion of additional data.

Major Comments

1. Abstract, lines 12-16: "We find that KIR2DL1-binding RIFINs can also bind to KIR2DS1 and that these RIFINs cause activation of KIR2DS1 expressing NK cells. This highlights the evolutionary battle between pathogen and host, suggesting that activating KIRs may have evolved to allow detection of red blood cells infected with *Plasmodium falciparum*, helping the host to clear the parasite."

Please could the authors re-word this section for improved accuracy. While a group of 15 KIR2DL1 binding RIFINs are identified here it seems only RBK21 undergoes structural characterisation and functional evaluation of KIR2DS1 binding in this manuscript.

We accept the reviewer's criticism here and have included substantial new data in the revised manuscript to strengthen these claims. These include:

- The identification of new KIR2DL1-binding RIFINs from Kenyan and Senegalese field isolates and the finding that KIR2DL1 binding RIFINs from 3D7 as well as South-East Asian and African field isolates fall into the same evolutionary clade, suggesting a common ancestor (Lines 87-115 and Figure 2a,b).
- The structure of KEN-01, one of the Kenyan RIFINs, bound to KIR2DL1, allowing structural comparison. While only 57% of contacts between KIR2DL1 and the RIFIN are shared between KEN-01 and RBK21, the binding mode is highly conserved (lines 117-144 and Figure 2c-e).
- The inclusion of KEN-01 in the supported lipid bilayer assay, to show that it signals through KIR2DL1 and suppresses perforin deposition by NK cells (lines 202-215 and Figure 3g-i).
- Competition data showing that the Kenyan and Senegalese isolates all compete with RBK21 for binding to KIR2DL1 (lines 154-159 and Extended Data Figure 4e).

These new data come together to support the view that the identified clade of KIR2DL1 binding RIFINs have similar structure and function.

It has been demonstrated that LILRB1 binding RIFINs bind to at least two distinct binding sites on LILRB1 and that the two LAIR1 binding RIFINs bind overlapping epitopes, but bind in different orientations (manuscript refs 2 and 3). Could it therefore be premature to suggest that this group of KIR2DL1 binding RIFINs possess similar characteristics to RBK21 and would be able to modulate NK signalling and bind KIR2DS1 in a similar manner leading to any significant evolutionary pressure?

We share the reviewer's view that the literature supports two different binding sites of LILRB1 for RIFINs, one identified for LILRB1 and one for LILRB1 domains D3 and D4. We have added this point in lines 154-155 and include our new competition experiment in lines 154-159 and Extended Data Figure 4e, which supports the view that KIR2DL1-binding RIFINs bind very similar, or at least overlapping, binding sites.

Extending this study to evaluate the remaining KIR2DL1 binding RIFINs would be extremely interesting, particularly with regard to their binding site on KIR2DL1 and their ability to impact both inhibitory and activating NK cell function.

We have now done this for KEN-01 as well as RBK21 and observe the same outcomes for KIR2DL1-binding RIFINs from very different origins.

2. Please could the authors address some concerns over the structural statistics and model quality presented in extended data table 7 and in the PDB validation report as follows:

- a. The completeness of the high resolution shell only reaches 58%. Despite the remaining indicators of data quality falling within acceptable ranges, this is extremely low. Ideally a resolution cutoff should be selected where this reached at least 90-95%.
- b. Please could the authors indicate how their multiplicity values were calculated as

there seems to be a mismatch between these values and the reported number of unique and total reflections.

The reviewer is correct to point out these aspects of the experimental data for RBK21. In both cases, this is due to anisotropy in the crystallographic. Diffraction limits across the 3 principal axes were 2.87Å, 3.12Å and 3.24Å, with the diffraction only going to the highest resolution along one of these axes. We have therefore processed the data with 2.89Å resolution, to include the data in the axes with the lowest resolution. This leads to the lower completeness in the outer shell and the mismatch in multiplicity. However, we are confident that this is the correct way to process our data, using a cut-off determined by CC1/2 and $I/\sigma(I)$ and it gives an improved map. The reviewer will also note that the new data for KEN-01, at the higher resolution of 2.17Å, does not display anisotropy, the completeness in the outer shell is 100% and the multiplicity more closely relates the total and unique observations, as expected.

c. The RSRZ score for the model indicates a large number of residues where there is a poor fit between the model and data. If possible, could the authors attempt to improve this value?

The reviewer is correct to note that the RSRZ score is high for the RBK21 structure. However, this is due to the unusual crystal packing in the lattice and can't be improved by further model building and refinement. The underlying issue here is that the crystal is built from a well-packed array of KIR2DL1 molecules, with RBK21 molecules protruding from the KIR2DL1 network into the solvent. As a result, their degree of order is lower than that of KIR2DL1. As a result, the average RSRZ ($\% < 2$) is 4.3% for the KIR2DL1 molecules and 27.3% for the RBK21 molecules, as shown here in a figure for each chain. Thankfully for our ability to generate a high-quality model of RBK21, one of the 8 copies in the asymmetric unit is better ordered and has an RSRZ ($X < 2$) of 5%. It is this molecule which allowed us to build a reliable atomic model and on which we base our conclusions. In the new, higher resolution, KEN-01 structure, the RIFIN is also well ordered with a RSRZ ($X < 2$) of 13%. Maps can be seen in Extended Data Figure 3c and 4b. Therefore, while the RSRZ appears very high in the validation report, this has not come at a cost of being able to build a reliable model or derive experimentally reliable data.

Minor Comments

1. The structural basis for the interaction between RIFINs and KIR2DL1: Line 94. It appears in the Chen et al. Nature 2021 paper (man. ref 5) that the LILRB1 binding RIFINs identified there bind a distinct epitope on D3 of LILRB1 in contrast to the D1D2 binding observed in Harrison et. al. Nature 2020 (man. ref 3). Could the authors please modify this line to reflect the multiple binding sites of RIFINs on LILRB1.

We agree with the reviewer's interpretation of these two papers. While Harrison et al shows binding of a RIFIN to D1D2 of LILRB1, Chen et al show a different RIFIN to bind to LILRB1 D3D4. We have added this in lines 154-55.

2. The structural basis for the interaction between RIFINs and KIR2DL1: Line 108. In extended data figure 6 it looks like the angle of the longest alpha helix also varies between copies. Could the authors please include a comment on this in the text if this is the case.

We have now extensively rewritten the section of the paper which describes the structures (lines 117-144) to focus on a comparison of RBK21 and KEN-01 and as a result, under space constraints, we no longer compare the differences between the 8 copies of RBK21 in the asymmetric unit, as we are not sure of the biological implications of differences. We retain the structural comparison in Extended Data Figure 6, now in Extended Data Figure 4a and describe this in the legend in line 988-9.

3. The structural basis for the interaction between RIFINs and KIR2DL1: Line 110-115. It is somewhat difficult to see the interface being discussed in the text in the current figure 2b. Perhaps the figure could be split into 2-3 panels for clearer visualisation of the hydrophobic pocket, the polar interactions and the stabilising beta sheet.

We have included new panels in Figure 2e, to attempt to show the interaction interface more clearly, and now include a comparison of RBK21 and KEN-01 in the text.

4. The structural basis for the interaction between RIFINs and KIR2DL1: Line 124-128. Could the authors please additionally comment on sequence diversity in the LAIR1 and LILRB1 binding RIFINs here.

Sequence diversity in LILRB1-binding RIFINs was studied in Harrison et al and we have now added this reference in line 153, where we discuss sequence diversity of KIR2DL1-binding RIFINs.

5. Inhibitory signalling mediated by KIR2DL1-binding RIFINs reduces cytotoxic attack of iRBCs by NK cells: lines 143-153. Could the authors please state whether all PBMCs in these assays are KIR2DL1 positive, or just those co-cultured with the K562- RBK21 expressing cells. It is currently worded slightly ambiguously in the results and accompanying methods section and could affect the interpretation of these results.

The NK cells used in this assay were obtained from a KIR2DL1-positive volunteer and cells were sorted using KIR2DL1-binding antibodies to obtain a KIR2DL1-positive NK cell population. These cells were then analyzed in the assay. This is now clarified in lines 181-184 and the sorting is also shown in Figure 3b. The equivalent for KIR2DS1-binding RIFINs is in Figure 4f.

6. Figure 3.

a. Could the authors please improve labelling of the figure axes here with fewer abbreviations and improved descriptors (for example % is used as a sole axis label in fig 3C). The y axis in figure 3F is cutoff.

We amended labelling axis of all figures for clarity.

b. There are no text call-outs for figures 3E and 3F, could the authors please include these alongside the text in lines 162-167.

These have been added.

7. Figure 4.

a. In Figure 4A it would be helpful to have the mismatching residues between KIR2DL1 and KIR2DS1 highlighted.

We have included this information underneath the sequence alignment with the lack of a * symbol indicating a mismatch.

b. Figure 4D. There seem to be no data on the number of replicates or the mean and SD values for the MFI for this assay.

The FACS analyses for binding iRBC with RBK21 to KIR2DL1 (Fig 1b) and KIR2DS1 (Fig. 4d) were each performed in triplicate, which we now state in the legends. The MFI with SD values for both assays are shown in those figures and the original data are shown in Source file. The gating strategy is shown in Supplementary Fig. 4b.

8. Fig 3b and Extended data fig 9. Fig 4F and Extended data fig 11. Extended data figure 12.

a. Could the authors please comment on the lack of any biological replicates for the assays in fig 3b and 4F? It would be extremely useful to show that these data were statistically significant and there were real differences in cytokine signalling.

We have now repeated these studies using NK cells from a different donor. In the assay using K562 expressing RBK21 and KIR2DL1(+) NK cells, the expression of CD107a and production of IFN- γ and TNF- α were suppressed reproducibly with statistical significance for both donors when compared with the control RIFIN (Fig. 3b-e and

Extended data Figure 5a). In the assay using iRBC expressing RBK21 and KIR2DS1(+) NK cells, those cytokine signalling are induced in two biological independent experiment, also with statistically significance (Fig. 4f-i and Extended data Figure 6b). In contrast, RBK21 did not suppress CD107a expression in activated NK-KIR2DL1 (-) cells, providing the relevant negative control (Extended data Figure 5b). The production of IFN- γ and TNF- α in activated NK-KIR2DL1 (-) cells seemed to slightly decrease in the presence of RBK21, however, these decreases were similar to those caused by the control RIFIN, suggesting that this effect is non-specific. Taken these results into consideration, we concluded that RBK21 can modulate NK function thorough specific interaction with KIR2DL1/DS1.

b. In extended data figure 11 it looks like the cell counts for the 3D7-RBK21 data may be considerably lower than for the other two conditions. Please could the authors include cell counts for these datasets. Was a gating strategy selected to ensure consistent cell counts across these assays?

We show cell counts for all datasets (Supplementary Figures 2 and 3). In this analysis, we first gated the constant number of NK cells purified from donors, followed by analysing the cytokine production and CD107a expression in particular subsets, such as KIR2DS1-(+), KIR2DL-(+) and KIR2DL1/DS1(-). As you can see, in the additional experiment, we analysed almost similar number of KIR2DS1-(+)-subset and detected apparent its activation by RBK21.

c. Unfortunately, I am unable to determine the appropriateness of the gating strategies shown in extended data figure 12 as none of the text or axes are legible. Please could the authors correct this.

We have provided an improved figure which is now in supplementary Figure 4.

d. In extended data figure 12G the legend refers to a gating strategy for extended data figure 10. I believe this is an error and should refer to extended data figure 11. Please could the authors correct this if appropriate.

This has been corrected as part of reorganising all of the figures.

Referee #5 (Remarks to the Author):

Several recent studies (including five published in Nature) have shown that *P. falciparum* RIFINs bind to human inhibitory receptors such as LAIR1 and LILRB1. In this new submission, the Authors extend these observations by identifying a class of RIFINs that bind to another inhibitory receptor, KIR2DL1, and to its homologous activatory receptor, KIR2DS1. However, the binding of RIFINs to KIR was already anticipated in Saito et al. Nature 2017 (Extended Figure 1).

For a limited set of the KIR2D-binding RIFINs, the Authors provide structural and functional data. The structural data show a mode of binding consistent with previous reports on RIFIN binding to the Ig domains of LAIR1 and LILRB1. The functional characterization, performed using a sophisticated system already reported in previous studies, aligns with expectations based on the properties of the engaged receptors. However, the in vivo relevance of binding to inhibitory or activatory receptors remains speculative.

In conclusion, this study is generally well executed, and the results are overall convincing. However, the findings are primarily incremental, offering no novel biological insights and addressing a highly specialized audience.

While we thank the reviewer for their opinion that the study is well executed and with convincing results, we strongly disagree with them that the findings are incremental. Indeed, we are supported in this view by all other reviewers, who comment that our manuscript describes a ‘fascinating chapter’ (Reviewer 2), ‘findings likely to have broad interest across the fields of infectious disease and immunology’ (Reviewer 4) and suggest that it is ‘really interesting’ and that the ‘novelty is significant’ (reviewer 6). While disappointed that reviewer 5 did not find our data novel, we are therefore reassured that they are the outlier and that the other reviewers found this to be an important advance.

The reviewer highlights Extended Data Figure 1 of Saito et al as the evidence that our study is incremental. In this seminal study, which first showed that RIFINs bind to any inhibitory immune receptor, LILRB1, parasite lines were screened against other inhibitory immune receptors, including LILRB1 and the KIRs. Both 3D7 and a patent-derived isolate showed strong binding to LILRB1 and weaker binding to LILRB2. There was no binding to KIR2DL1 by the 3D7 strain in this experiment and a very small number of weakly KIR2DL1-binding clones in the patent-derived isolate. This was not followed up with any further experiments in the manuscript and no conclusions were drawn from this observation, which may have been noise, due to the far smaller signal for KIR2DL1 than that observed for LILRB1. We therefore do not agree that Saito et al showed that there are KIR2DL1 binding RIFINs and we are confident that should this claim have been made in a draft of Saito et al, it would not have stood up to peer review.

Instead, our current manuscript provides the first demonstration that there are RIFINs which bind to KIR2DL1, the first demonstration that these are widespread across field-isolated strains from different geographical locations and the first demonstration that these inhibit NK cell function.

In even greater novelty, our manuscript provides the first identification of any pathogen-derived ligand for an activating KIR. We not only show binding but also demonstrate that these RIFINs can activate NK cells which express KIR2DS1, giving the first experimental evidence to support the theory that activating KIRs help human survival in the context of pathogen infection.

We therefore do not agree with the reviewer that these findings are incremental but instead agree with the views of the other reviewers that this study is novel, revealing fascinating insight into the roles of activating receptors, as well as malaria.

Specific point: why did the Authors test binding to LILRB1 in Figure 2c and what do they conclude from this experiment?

We thank the reviewer for pointing out this minor typographic error and have corrected it.

Referee #6 (Remarks to the Author):

The authors appear to have uncovered a really interesting and potentially important set of receptor/ligand interactions that if fully substantiated have implications both for mechanisms of NK cell recognition of malaria infection and the generation of early innate immune responses as well as the selection and diversification of KIR genes across

evolution. The novelty is significant as immune evasion strategies involving direct binding of pathogen-encoded molecules to inhibitory KIR have not been described. Moreover, our understanding of the biological significance of activating KIR has remained very limited since their discovery. The observation that a pathogen with sufficient prevalence and disease severity expresses proteins that can be directly recognised by activating KIR provides perhaps a plausible driver for the relatively recent evolution of such KIR, albeit that it is juxtaposed with other observations that might suggest that the primary function of activating KIR is related to regulating the development of spiral arteries in the placenta. However establishing clinical significance may be a step too far at this point in time since it would require detailed genetic analyses of both host-encoded KIR (to identify malaria-infected individuals with KIR2DL1 but not KIR2DS1 as well as those who possess KIR2DS1), detailed analyses of the RIFIN repertoires of infected individuals all linked to clinical data. Regardless, showing that malaria-encoded proteins directly engage KIR2DL1 or KIR2DS1 to impact infection is significant.

We thank the reviewer for highlighting the importance of this study and its relevance for both malaria and NK cell biology, as the first discovery of a ligand for an activating KIR.

Unfortunately, in its current format, I do not think the manuscript sufficiently substantiates the primary observation or indeed use the data at hand to show the observation in its best light. In short, while the generation of transgenic parasites seems almost heroic, the work substantiating KIR binding and functional analyses to address the broader impact of these interactions on NK cell biology is underdone and frequently either lacks appropriate controls or in the case of the structural data, not really highlighting the relevant interactions with sufficient clarity.

We have addressed the reviewer's comments below, including additional controls and repeats with NK cells from different donors. In our view, this data strongly supports our conclusions, which are the identification of KIR2DL1-binding RIFINs, the demonstration that these are widespread in parasite isolates and that they inhibit KIR2DL1-expressing NK cell function. We are also of the view that our data strongly supports the new finding that these same RIFINs bind to KIR2DS1 and that this leads to the activation of KIR2DS1-expressing NK cells. We are therefore convinced that the conclusions of our study are robustly demonstrated by our data.

Some of the reviewer's comments relate to how RIFINs are specific for KIR2DL1 and whether there are RIFINs which bind to other KIRs. In our view, these comments fall outside the scope of this study. As discussed in the introduction, there are many KIRs, each with many allotypes and these vary in different human populations. Coupled with the many RIFINs in each parasite genome, this creates a large combinatorial problem. Unpicking this will therefore be a major undertaking and will form the subject of future studies, which follow on from the substantial discovery outlined in this manuscript.

I have outlined concerns in more detail below.

The statement "Activating KIR are proposed to have evolved from inhibitory KIRs through mutations in the ITIM domain." (line 36)...is loose. More correctly a KIR were created by 1) truncation of the tail and 2) addition of a charged residue in their transmembrane region to facilitate pairing with ITAM-containing adaptor proteins such as DAP12 and (3) mutations in the extracellular domain that appear designed to attenuate recognition of HLA-encoded ligands.

We agree with the reviewer and have rewritten this section of the introduction (lines 36-39) to more accurately outline the changes involved in evolution of activating KIRs.

The initial screening data is interesting but lacking important controls. Negative data from KIR-Ig fusion proteins is not sufficiently robust in the absence of some demonstration of their capacity to bind defined ligands. Supplementary data showing that these reagents bind the expected HLA allotypes and or CD155 in the case of KIR2DL5 should be included. Also included should be their binding to uninfected RBC.

The binding of KIR2DL3 is 2.5% which is not that different from the KIR2DL1 value, yet the allelically related KIR2DL2 has almost no binding. In the absence of data which provides confidence about the fidelity of the reagents, little can be concluded about the selectivity of the RIFIN/KIR2DL1 interaction for KIR2DL1 over other KIR.

To demonstrate that the KIR-Fc proteins which we used in these screening studies are correctly folded we selected a panel of antibodies with conformational epitopes for each protein. We then tested these for their ability to bind to bead-conjugated KIR-Fc using flow sorting. This assay showed that all KIR-Fc reagents used in our screening bound correctly to antibodies with conformational epitopes. This data is now shown in Extended Data Figure 1a. Together with secretion of these proteins from HEK293 cells and demonstration during purification that they are monodispersed on a size exclusion column, we are confident that they are high quality and folded. Therefore, the negative outcome of the initial screen study is not due to incorrect folding of the KIR-Fc proteins. However, we have now shown in our new data that the genomes of isolates negative in this screening assay may contain RIFINs which bind to KIR2DL1-Fc (as demonstrated for Lek79). Therefore a negative outcome in the screen does not mean that there are not RIFINs encoded in the genome which bind to other KIRs. We made this clearer in the text, for example lines 51-53.

The reviewer also asks for erythrocyte binding studies. However, as the KIR-Fc reagents were negative for binding to infected erythrocytes, we do not agree that this control is required. Similarly, for the KIR2DL1-Fc, the experiment is internally controlled as no binding was seen for erythrocytes infected with three of the lines tested in the initial screen, with binding only observed when KIR2DL1-expressing RIFINs are present. We are therefore confident that the positive signal for KIR2DL1 and Lek174 is meaningful, and indeed this outcome has been validated in many ways in the subsequent data.

The use of library strategies to identify additional KIR2DL1-binding RIFINs is powerful and elegant. The mapping of KIR2DL1- binders to a single clade is nice. The Logo analyses of KIR-binding RIFINS is informative. However it would have also been helpful to identify features that were shared/or not between these the 2 clade members that didn't bind KIR2DL1. The statement that the conservation of KIR2DL1-binding RIFINs suggests there is a major selective advantage to targeting the KIR2DL1 inhibitory receptor seems overstated at this point in time. It is not clear to me how inhibitory KIR modulate NK cell recognition of infected RBC. More context is needed here.

We now include substantial extra data in the manuscript which identifies KIR2DL1 binding RIFINs from four different field-isolated strains from South-East Asia and Africa. In particular, our prediction of KIR2DL1-binding RIFINs from two African strains involved phylogenetic analysis of these RIFINs in conjunction with those from 3D7. These studies show that we can predict KIR2DL1 binding properties, but that the accuracy of prediction

is imperfect. Indeed, we have not been able to use the structural insight to unambiguously explain why our prediction is not fully accurate. However, we do not claim in the manuscript to be able to be fully able to predict KIR2DL1- and KIR2DS1-binders (i.e. line 111 and 285-301). The manuscript outlines our discovery of these interactions, and their consequences for NK cell function, rather than any claim to present a comprehensive mapping. We are currently conducting a global study to experimentally show what inhibitory receptors all RIFINs in a genome bind, and we aim that this will increase our prediction accuracy, but this is a large piece of work for a different study.

The structural data should be informative but at present is not presented very clearly and perhaps is not used to its full extent. Does it provide insights into the selectivity of these RIFINs for KIR2DL1/S1 as opposed to KIR2DL2/3 or indeed other 2 domain KIR. Some more formal comparison and tests of the key features of KIR and the extent to which they differ across different KIR and even between KIR2DL1 allotypes should be shown. The mutagenesis experiments at the receptor/ligand interface seem pretty minimal and should be extended to test their hypotheses around the key drivers of the RIFIN/KIR interaction and by extension insights into its binding into specific clades or clade members.

Our view is that the experiments proposed here are largely out of the scope of this manuscript. Here, we present the discovery of KIR2DL1- and KIR2DS1-binding RIFINs and reveal that the binding of RIFINs to these two receptors have opposite effects on NK cell activity. We also show that these KIR2DL1-binding RIFINs are commonly found in strains from very different geographic locations.

Our S221R mutant was not produced as part of an aim to comprehensively dissect the role of interactions in the binding site, but rather to generate a non-binding mutant which folds correctly as a control for functional studies. Indeed, in a protein family such as the RIFINs, which retain binding function despite extensive sequence variation and without a clear 'motif' for binding, we are not of the view that an extensive mutagenesis campaign will be informative, other than showing how the residues from one RIFIN contribute to binding to KIR2DL1.

It was also not our aim to investigate specificity for KIR2DL1 over other KIRs. As mentioned above, this would involve large and comprehensive studies of all KIRs and KIR allotypes and is outside the scope of this study.

We have, however, followed the reviewer's advice and investigated the location of the changes in the different KIR2DL1 allotypes. We plotted all allotype-associated changes on to the structure and find that only one well-characterised allotype involves a change in the RIFIN binding site. However, we show that this V111F mutation does not affect RBK21 binding. Therefore, we see no evidence that allotypes have evolved to prevent RIFIN binding. We describe this in lines 160-165 and present the data in Extended Data Figure 4g.

The data showing that transfection of RBK21 into K562 protects seem relatively clear. That said, there are additional controls that should be included - responses of KIR2L1-ve NK cells (especially those expressing other similar 2 domain KIR such as KIR2DL2/3). Some form of blocking experiment would be also typical here albeit that the mAbs used for blocking HLA interactions likely will not suffice in this case due to differences in binding sites. It is unclear to me the value of 9 technical replicates or additional clarity on

what is meant by technical replicates is required. I would prefer to see data analysing NK cell responses from multiple donors where the number of donors is clearly indicated. It is also not clear the extent to which the inhibition of activation of primary NK cells is in part dependent on the function of other RIFIN-binding inhibitory receptors. The authors also need to show the level of cell surface expression of both KIR2DL1-binding and non-binding RIFIN's on their K562 cells.

We have now conducted some additional controls to further strengthen the conclusions in the manuscript.

We have now tested the effect of RBK21 on primary NK cells obtained from two genetically different donors. The clear suppression by RBK21 was detected in this additional experiment, which is consistent with previous results (Figure 3b-e and Extended Data Figures 5a). Moreover, suppression of activation primary NK cells by the control-RIFIN (PF3D7_1254200) were not detected, indicating specificity.

We further analyzed the responses of NK cells, which are KIR2DL1/(DS1)-negative, when these cells were co-cultured with K562 expressing RBK21. The results shows that there is no suppression of NK cell activation, indicating that RBK21 specifically inhibit the NK cell function by binding to KIR2DL1 (Extended Data Figure 5b).

We confirmed that the expression level of RBK21 on K562 is comparable to that of the control RIFIN (Extended data Fig, 5c), showing that the specific suppression by RBK21 is not dependent on the expression level.

The extent of inhibition observed with NKL cells expressing KIR2L1 with respect to cytotoxicity is not particularly convincing and again additional controls that include NKL cells that do not express KIR2DL1 and/or cells that express other KIR would minimally be required. Additionally, target cells expressing a “non-interacting” RIFIN or the S221R mutant should be included.

Firstly, we improved the method for this assay: prior to the assay, all cell lines, which included NKL, transgenic NKL, K562 and transgenic K562, were purified by density gradient using Ficoll. Highly viable cells (>95%) can be obtained by this purification and used for the assay. The results now more clearly show the specific inhibition of activation of NKL (Figure 3f).

We have now added additional control experiments. We conducted the inhibition of NKL cells expressing KIR2DL1 and KIR2DL3 by RBK21 (Figure 3f and Extended Data Figure 5d). In addition to RBK21, we used PF3D7_1254200 as non-interacting RIFIN in this additional experiment.

Fig 4 is largely focussed on demonstrating that RBK21 also binds to KIR2DS1 and that this can stimulate NK cell activation. Again the data is strongly suggestive of these things but lacks key controls. The SPR experiments should include both non-binding RIFIN as well as control KIR (eg KIR2DL2). The KIR2S1-Ig binding data should also include a non-binding KIR. The functional analyses showing activation following recognition of infected RBC again needs additional analyses on subsets of NK cells that do not express either KIR2DL1 or -2DS1. Additionally, the authors need to be clearer about how many donors have been used in these analyses- what exactly is 9 technically independent samples?).

We have added different controls. The data in Figure 2b now includes LILRB1 for a set of RIFINs. The KIR2DL1-binding RIFINs all bind to KIR2DS1. Interestingly, one also binds LILRB1, but following up this exception is outside the scope of this study.

In terms of the NK cells used, we have clarified this in the methods and we present data in the manuscript to show the selection and validation of these cells. We purified primary NK cells from two different KIR2DL1(+) and KIR2DS1(+) donors. The outcomes of these repeats are consistent with those using another donor, which is used in the previous manuscript (Extended Data Figures 5a and 6b). In contrast, similar induction of CD107a expression, and IFN- γ and TNF- α production were not detected in KIR2DS1(-) and KIR2DL1(-)-NK subsets (Extended Data Figure 5b), indicating specific activation of NK cells through KIR2DS1 by RBK21.

In this study, we used two donors with KIR2DS1(+)-NK cells. All assays were performed 9 or 6 replicates for each condition and this is stated in the methods and legends.

The authors conclude that RIFINs are the evolutionary drivers for the emergence of activating KIR. This seems premature since there is strong genetic evidence of a role for such receptors in impacting reproductive outcomes independent of infection. Secondly, if this is the evolutionary driver, what then of other activating KIR? Are these also RIFIN specific but the strains assessed simply did not have ligands within their RIFIN-repertoire.

We agree with the reviewer that our argument that RIFINs have been a cause for the development of activating KIR is premature and we have removed the claim from revised manuscript. On the other hand, our data which shows that KIR2DS1-binding RIFINs increase the activation of NK cells expressing KIR2DS1 suggest that individuals with activating KIRs are likely to have a survival advantage in malaria-endemic areas. We increased of discussion of this point in the revised manuscript.

The underlying purpose of having an inhibitory receptor that is specific for a malaria - encoded Ag expressed on the surface of red blood cells is not clear. The data suggest that NK cells can kill RIFIN-expressing rbc as a result of KIR2DS1 engagement. Does KIR2DL1 engagement diminish ADCC responses targeting infected rbc?

It is not one of our conclusions that inhibitory receptors have evolved to bind to RIFINs, as the reviewer correctly points out that this is not in the interest of the infected human. Instead, the hypothesis is that RIFINs evolved to bind to inhibitory receptors, thereby enabling the parasite to activate mechanisms which humans use to prevent immune cell activation in response to self. Substantiated data to support this hypothesis is presented in Saito et al and Harrison et al for LILRB1-binding RIFINs and in Figure 3 for the KIR2DL1-binding RIFINs. Indeed, these data show that the presence of inhibitory immune receptor-binding RIFINs suppress NK cell function. This has been outlined more clearly in the discussion.

Ref. 1

Sakoguchi and colleagues have responded robustly to the referee comments. I appreciate that the authors in several cases have accepted referee arguments and removed the more speculative discussions/conclusions that were inconsistent with data. I was also impressed with the additional analysis the authors undertook to show the general significance of the RIFIN-KIR interaction, such as in the subsection entitled: "KIR2DL1-binding RIFINs are common in field isolates from both Africa and South-East Asia". The additional structure of KEN-01 bound to KIR2DL1 as well as the additional sequence analysis on the conservation of the recognition motif aid in demonstrating generality.

We thank the reviewer for this positive assessment of the revised manuscript.

Ref. 2

The authors have submitted a substantially revised manuscript and have gone to considerable effort to address the reviewers comments with both improvements to the figures and text as well as a considerable amount of new data.

I am satisfied that the majority of my comments have been addressed in the current version. While it would be exciting to see a slightly broader scope explored with regard to RIFIN interactions with other KIR family members, I agree that this work represents a considerable and novel addition to the field. The additional structural studies on a second KIRDL2 binding RIFIN KEN-01 help to validate the original RBK21 dataset, which is unfortunately somewhat limited in resolution and the quality of density for the RIFIN as the authors have stated in the rebuttal. It is particularly pleasing to see the inclusion of data showing the presence of KIRDL2 binding RIFINs in multiple field isolates which helps to build a case for this interaction being of importance for host-parasite biology. Revisions to the manuscript text and figures have helped to improve clarity and reduce overstating the results presented.

We thank the reviewer for their positive comments about the revised manuscript and its significance. We are very happy to be able to provide this additional data to strengthen our conclusions and to demonstrate broader significance.

Minor Comments

1. Line 141: To develop a non-binding mutant for future studies we produced S221R. This abolished binding to KIR2DL1, as shown through SPR analysis.

As this section is now a comparative discussion of the binding of RBK21 and KEN-01 to KIR2DL1 it should be clearly stated that this mutant was produced to eliminate binding for RBK21 and abolish only this interaction.

We agree with the reviewer and have added 'of RBK21' to line 141 to make this clear.

2. Extended data figure 3C.

The mesh used for the KIR2DL1-KEN01 complex in this figure is double that of the adjacent KIR2DL1-RBK21 figure and those in the rest of the manuscript. Could the authors please make this uniform for easier comparison of the structures.

We thank the reviewer for pointing this out. The two panels had been made at different times with different versions of PyMOL, which explained the discrepancy. We have now remade both using the same PyMOL settings and the grid spacing is more similar. The spacing will not be identical as this is influenced by map resolution, but these two maps have been treated in the same way and display in this figure is equivalent.

3. Extended Data figure 5.

I believe there is a typo in the figure legend as it reads "KIR2DBL1".

We have fixed this typo.

4. Extended data table 3.

Wavelength is given to different decimal places for RBK21 versus KEN-01 structures and units are included only for the RBK21 structure rather than for both or as part of the row heading.

We have corrected this error

We thank the reviewer for their careful attention to the manuscript and for helping us by pointing out these slips.

Ref. 3

The manuscript by Sakoguchi et al provides clear evidence of an interaction between RIFINs and KIR. While there is a significant number of RIFINs in any single parasite and considerable strain to strain variation, the capacity to bind KIR is present in isolates from geographically distinct areas and at a frequency that suggests significance. The authors have used crystallography to illustrate the molecular basis of the interaction which highlights the key elements of both the KIR and RIFIN.

While there are numerous pathogen-encoded immunoevasins that target NK cell responses, this is the first identified example of such a protein directly binding a KIR, with RIFINs selectively binding KIR2DL1 and -2DS1. The reasons for such selective targeting remain unclear. Critically, direct recognition of pathogen-encoded proteins by activating NK cell receptors has been reported previously, this is again the first example of an activating KIR binding such a protein despite such mechanisms being proposed as an evolutionary driver for the generation of such KIR well over a decade earlier.

We thank the reviewer for highlighting that this is the first example of an activating KIR binding a pathogen surface protein and for their view that this is a significant discovery.

In my opinion the central observation is well supported by the data which clearly shows an interaction via direct binding assays, structural techniques as well as functional assays and the significance of the observation worthy of publication in Nature. While recognising the magnitude of the changes in functional responses as significant, I am still surprised that the authors have still only analysed NK cells from 2 donors and then used either reporter cells or NKL cells to further substantiate the functional relevance.

We thank the reviewer for their comments and for their view that our conclusions are well supported by the data.

Relating to the reviewer's point about the number of donors from which we studied NK cells, we tested PBMCs purified from multiple donors using KIR2DS1 antibodies prior to the NK suppression assay. However, we identified only two KIR2DS1(+) donors, despite the estimated frequency of KIR2DS1 in the Japanese population being approximately 40%. To increase the number of KIR2DS1(+) donors, we purchased KIR2DS1(+) PBMCs from abroad. However, the viability of NK cells purified from these purchased PBMCs was very low, making them unsuitable for our assays. This is why we used only two KIR2DS1(+) donors in this study. Although the number of donors used was small, the reproducibility of the results led us to conclude that KIR2DS1 recognizes RBK21 and activates NK cells.

On the other hand, the frequency of KIR2DL1 in the Japanese population is estimated to be over 90%. In the manuscript, we included data from two KIR2DL1(+) donors. However, we also collected data using cells from two more donors, bringing the total to four KIR2DL1(+) donors. The datasets using the third and fourth donors showed that RBK21 was also able to suppress the expression of CD107a and the production of IFN γ and TNF α , with a similar outcome for all four donors (please refer the figure 1 and 2 in this response). Due to size and number limitations of extended figures and supplementary information, we chose to present two datasets in the manuscript, rather than showing the data in smaller panels, and the data from the other donors is shown here for completeness.

In view of these independent experiments, which agree with lines of evidence from other types of experiments, we are very confident in the robustness of our conclusions.

Figure 1 a and b. Suppression of CD107a expression and IFN- γ and TNF- α production in KIR2DL1-(+)-NK cells by K562 cells which express RBK21. K562 parental cells or K562 cells expressing a RIFIN which doesn't bind KIR2DL1 (ctl-RIFIN, PF3D7_1254200) were used as negative controls. In **a** and **b**, NK cells from different donors were used. These donors were different from the donors studied in Fig. 3 and Extended Data Figure 5. The line shows the mean (n=6 for a and n=9 for b) with ****P < 0.0001, **P < 0.01, *P < 0.05 (two-sided Student's t-test).

Figure 2. a and b are representative raw data used for Fig. 1a and 1b, respectively. The gating was performed in similar ways as shown in Supplementary information 4f.